# Zero-Shot Whole-Body Humanoid Control via Behavioral Foundation Models

**Andrea Tirinzoni**[1,*], **Ahmed Touati**[1,*], **Jesse Farebrother**[2,†] , **Mateusz Guzek**[1],
**Anssi Kanervisto**[1], **Yingchen Xu**[1,3], **Alessandro Lazaric**[1,‡] **& Matteo Pirotta**[1,‡]
[1] Fundamental AI Research at Meta, [2] Mila, McGill University, [3] UCL
{tirinzoni,atouati,matguzek,anssik,ycxu,lazaric,pirotta}@meta.com

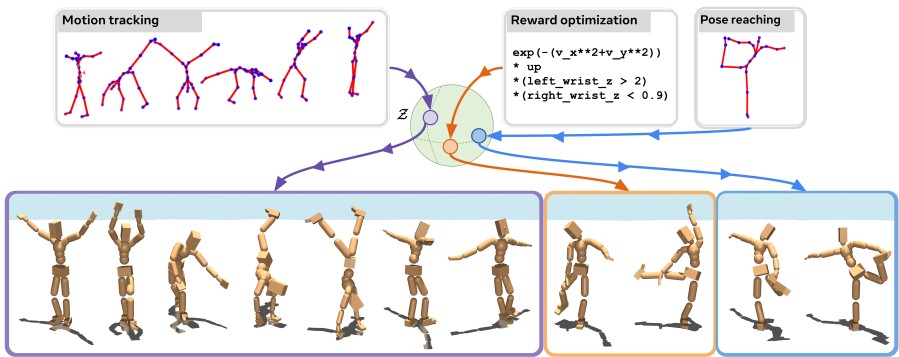

Figure 1: With FB-CPR we pre-train the first behavioral foundation model for humanoid that can solve tracking, goal-reaching, and reward optimization tasks in a zero-shot fashion.

## ABSTRACT

Unsupervised reinforcement learning (RL) aims at pre-training models that can solve a wide range of downstream tasks in complex environments. Despite recent advancements, existing approaches suffer from several limitations: they may require running an RL process on each task to achieve a satisfactory performance, they may need access to datasets with good coverage or well-curated task-specific samples, or they may pre-train policies with unsupervised losses that are poorly correlated with the downstream tasks of interest. In this paper, we introduce FB-CPR, which regularizes unsupervised zero-shot RL based on the forward-backward (FB) method towards imitating trajectories from unlabeled behaviors. The resulting models learn *useful* policies imitating the behaviors in the dataset, while retaining zero-shot generalization capabilities. We demonstrate the effectiveness of FB-CPR in a challenging humanoid control problem. Training FB-CPR online with observation-only motion capture datasets, we obtain the first humanoid behavioral foundation model that can be prompted to solve a variety of whole-body tasks, including motion tracking, goal reaching, and reward optimization. The resulting model is capable of expressing human-like behaviors and it achieves competitive performance with task-specific methods while outperforming state-of-the-art unsupervised RL and model-based baselines.[1]

## 1 INTRODUCTION

Foundation models pre-trained on vast amounts of unlabeled data have emerged as the state-of-the-art approach for developing AI systems that can be applied to a wide range of use cases and solve complex tasks by responding to specific prompts (e.g., Anil et al., 2023; OpenAI et al., 2024; Dubey et al., 2024). A natural step forward is to extend this approach beyond language and visual domains,

---

[*]Joint first author. [†]Work done at Meta. [‡]Joint last author.

[1]Code, models, and an interactive demo are available at https://metamotivo.metademolab.com.

towards *behavioral* foundation models (BFMs) for agents interacting with dynamic environments through actions. In this paper, we aim to develop BFMs for humanoid agents and we focus on whole-body control from proprioceptive observations, a long-standing challenge due to the high-dimensionality and intrinsic instability of the system (Peng et al., 2021; Won et al., 2022; Luo et al., 2024a). Our goal is to learn BFMs that can express a diverse range of behaviors in response to various prompts, including behaviors to imitate, goals to achieve, or rewards to optimize. By doing so, we could significantly simplify the creation of general-purpose humanoid agents for robotics (Cheng et al., 2024), virtual avatars, and non-player characters (Kwiatkowski et al., 2022).

While recent advancements in unsupervised reinforcement learning (RL) have demonstrated the potential of BFMs, several limitations still exist. Pre-trained policies or representations (e.g., Eysenbach et al., 2019; Schwarzer et al., 2021) still require training an RL agent to solve any given downstream task. Unsupervised zero-shot RL (e.g., Touati et al., 2023; Frans et al., 2024) addresses this limitation by pre-training policies that are *promptable* (e.g., by rewards or goals) without additional learning or planning. However, this approach relies on **1)** access to large and diverse datasets of transitions collected through some *unsupervised exploration* strategy, and **2)** optimize *unsupervised losses* that aim at learning as many and diverse policies as possible, but provide limited inductive bias on which ones to favor. As a result, zero-shot RL performs well in simple environments (e.g., low-dimensional continuous control), while struggle in complex scenarios with high-dimensional control and unstable dynamics, where unsupervised exploration is unlikely to yield useful samples and unsupervised learning may lead to policies that are not well aligned with the tasks of interest.

An alternative approach is to train sequence models (e.g., transformer- or diffusion-based) from large demonstration datasets to clone or imitate existing behaviors and rely on their generalization capabilities and prompt conditioning to obtain different behaviors (e.g., Schmidhuber, 2019; Chen et al., 2021; Wu et al., 2023). This approach is particularly effective when high-quality task-oriented data are available, but it tends to generate models that are limited to reproducing the policies demonstrated in the training datasets and struggle to generalize to unseen tasks (Brandfonbrener et al., 2022). Recently, several methods (e.g., Peng et al., 2022; Gehring et al., 2023; Luo et al., 2024b) integrate demonstrations into an RL routine to learn "regularized" policies that preserve RL generalization capabilities while avoiding the issues related to complete unsupervised learning. While the resulting policies can serve as *behavior priors*, a full hierarchical RL process is often needed to solve any specific downstream task. See App. A for a full review of other related works.

In this paper, we aim at leveraging an unlabled dataset of trajectories to ground zero-shot RL algorithms towards BFMs that not only express *useful* behaviors but also retain the capability of solving a wide range of tasks in a *zero-shot fashion*. Our main contributions in this direction are:

- We introduce FB-CPR (Forward-Backward representations with Conditional Policy Regularization) a novel online unsupervised RL algorithm that grounds the unsupervised policy learning of forward-backward (FB) representations (Touati & Ollivier, 2021) towards imitating observation-only unlabeled behaviors. The key technical novelty of FB-CPR is to leverage the FB representation to embed unlabeled trajectories to the same latent space used to represent policies and use a latent-conditional discriminator to encourage policies to "cover" the states in the dataset.

- We demonstrate the effectiveness of FB-CPR by training a BFM for whole-body control of a humanoid that can solve a wide range of tasks (i.e., motion tracking, goal reaching, reward optimization) in zero-shot. We consider a humanoid agent built on the SMPL skeleton (Loper et al., 2015), which is widely used in the virtual character animation community for its human-like structure, and we use the AMASS dataset (Mahmood et al., 2019), a large collection of uncurated motion capture data, for regularization. Through an extensive quantitative and qualitative evaluation, we show that our model expresses behaviors that are "human-like" and it is competitive with ad-hoc methods trained for specific tasks while outperforming unsupervised RL as well as model-based baselines. Finally, we run extensive ablations in the humanoid, walker (App. E), and ant maze (App. F) environments demonstrating the crucial role of the regularization scheme of FB-CPR.

## 2 PRELIMINARIES

We consider a reward-free discounted Markov decision process $\mathcal{M} = (S, A, P, \mu, \gamma)$, where $S$ and $A$ are the state and action space respectively, $P$ is the transition kernel, where $P(\mathrm{d}s'|s, a)$ denotes

the probability measure over next states when executing action $a$ from state $s$, $\mu$ is a distribution over initial states, and $\gamma \in [0, 1)$ is a discount factor. A policy $\pi$ is the probability measure $\pi(\mathrm{d}a|s)$ that maps each state to a distribution over actions. We denote $\Pr(\cdot|s_0, a_0, \pi)$ and $\mathbb{E}[\cdot|s_0, a_0, \pi]$ the probability and expectation operators under state-action sequences $(s_t, a_t)_{t \geq 0}$ starting at $(s_0, a_0)$ and following policy $\pi$ with $s_t \sim P(\mathrm{d}s_t|s_{t-1}, a_{t-1})$ and $a_t \sim \pi(\mathrm{d}a_t|s_t)$.

**Successor measures for zero-shot RL.** For any policy $\pi$, its *successor measure* (Dayan, 1993; Blier et al., 2021) is the (discounted) distribution of future states obtained by taking action $a$ in state $s$ and following policy $\pi$ thereafter. Formally, this is defined as

$$M^\pi(X|s, a) := \sum_{t=0}^\infty \gamma^t \Pr(s_{t+1} \in X \mid s, a, \pi) \qquad \forall X \subset S, \tag{1}$$

and it satisfies a measure-valued Bellman equation (Blier et al., 2021),

$$M^\pi(X|s, a) = P(X \mid s, a) + \gamma \mathbb{E}_{s' \sim P(\cdot|s,a), a' \sim \pi(\cdot|s')} \Big[ M^\pi(X|s', a') \Big], \quad X \subset S. \tag{2}$$

We also define $\rho^\pi(X) := (1 - \gamma)\mathbb{E}_{s \sim \mu, a \sim \pi(\cdot|s)} \big[ M^\pi(X|s, a) \big]$ as the stationary discounted distribution of $\pi$. Given $M^\pi$, the action-value function of $\pi$ for any reward function $r : S \to \mathbb{R}$ is

$$Q_r^\pi(s, a) := \mathbb{E}\left[ \sum_{t=0}^\infty \gamma^t r(s_{t+1}) \mid s, a, \pi \right] = \int_{s' \in S} M^\pi(\mathrm{d}s'|s, a) r(s'). \tag{3}$$

The previous expression conveniently separates the value function into two terms: 1) the successor measure that models the evolution of the policy in the environment, and 2) the reward function that captures task-relevant information. This factorization suggests that learning the successor measure for $\pi$ allows for the evaluation of $Q_r^\pi$ on any reward without further training, i.e., zero-shot policy evaluation. Remarkably, using a low-rank decomposition of the successor measure gives rise to the *Forward-Backward (FB)* representation (Blier et al., 2021; Touati & Ollivier, 2021) enabling not only zero-shot policy evaluation but also the ability to perform zero-shot policy optimization.

**Forward-Backward (FB) representations.** The FB representation aims to learn a finite-rank approximation to the successor measure as $M^\pi(X|s, a) \approx \int_{s' \in X} F^\pi(s, a)^\top B(s') \rho(\mathrm{d}s')$, where $\rho$ is the a state distribution, while $F^\pi : S \times A \to \mathbb{R}^d$ and $B : S \to \mathbb{R}^d$ are the *forward* and *backward* embedding, respectively. With this decomposition, for any given reward function $r$, the action-value function can be expressed as $Q_r^\pi(s, a) = F^\pi(s, a)^\top z$, where $z = \mathbb{E}_{s \sim \rho}[B(s)r(s)]$ is the mapping of the reward onto the backward embedding $B$. An extension of this approach to multiple policies is proposed by Touati & Ollivier (2021), where both $F$ and $\pi$ are parameterized by the same task encoding vector $z$. This results in the following unsupervised learning criteria for pre-training:

$$\begin{cases} M^{\pi_z}(X|s, a) \approx \int_{s' \in X} F(s, a, z)^\top B(s') \, \rho(\mathrm{d}s'), & \forall s \in S, a \in A, X \subset S, z \in Z \\ \pi_z(s) = \arg\max_a F(s, a, z)^\top z, & \forall (s, a) \in S \times A, z \in Z, \end{cases} \tag{4}$$

where $Z \subseteq \mathbb{R}^d$ (e.g., the unit hypersphere of radius $\sqrt{d}$). Given the policies $(\pi_z)$, $F$ and $B$ are trained to minimize the temporal difference loss derived as the Bellman residual of Eq. 2

$$\mathscr{L}_{\mathrm{FB}}(F, B) = \mathbb{E}_{\substack{z \sim \nu, (s,a,s') \sim \rho \\ s^+ \sim \rho, a' \sim \pi_z(s')}} \Big[ \big( F(s, a, z)^\top B(s^+) - \gamma \overline{F}(s', a', z)^\top \overline{B}(s^+) \big)^2 \Big] \tag{5}$$
$$- 2\mathbb{E}_{z \sim \nu, (s,a,s') \sim \rho} \big[ F(s, a, z)^\top B(s') \big],$$

where $\nu$ is a distribution over $Z$, and $\overline{F}, \overline{B}$ denotes stop-gradient. In continuous action spaces, the $\arg\max$ in Eq. 4 is approximated by training an actor network to minimize

$$\mathscr{L}_{\mathrm{actor}}(\pi) = -\mathbb{E}_{z \sim \nu, s \sim \rho, a \sim \pi_z(s)} \Big[ F(s, a, z)^\top z \Big]. \tag{6}$$

In practice, FB models have been trained offline (Touati et al., 2023; Pirotta et al., 2024; Cetin et al., 2024b), where $\rho$ is the distribution of a dataset of transitions collected by unsupervised exploration.

**Zero-shot inference.** Pre-trained FB models can be used to solve different tasks in zero-shot fashion, i.e., without performing additional task-specific learning, planning, or fine-tuning. Given a dataset of reward samples $\{(s_i, r_i)\}_{i=1}^n$, a reward-maximizing policy $\pi_{z_r}$ is inferred by computing $z_r = \frac{1}{n} \sum_{i=1}^n r(s_i)B(s_i)$.[2] Similarly, we can solve zero-shot goal-reaching problems for any state

---

[2]The inferred latent $z$ can also be safely normalized since optimal policies are invariant to reward scaling.

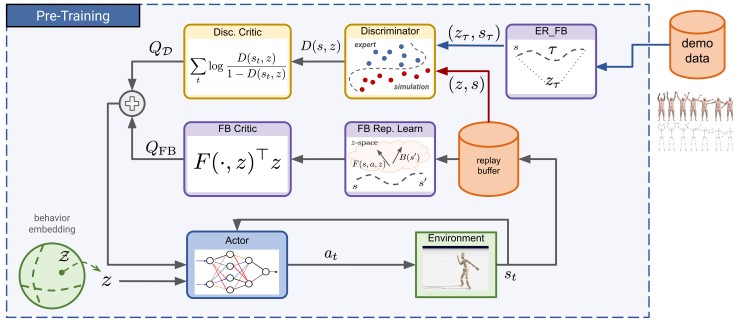

Figure 2: Illustration of the main components of FB-CPR: the discriminator is trained to estimate the ratio between the latent-state distribution induced by policies ($\pi_z$) and the unlabeled behavior dataset $\mathcal{M}$, where trajectories are embedded through $\text{ER}_{\text{FB}}$. The policies are trained with a regularized loss combining a policy improvement objective based on the FB action value function and a critic trained on the discriminator. Finally, the learned policies are rolled out to collect samples that are stored into the replay buffer $\mathcal{D}_{\text{online}}$.

$s \in S$ by executing the policy $\pi_{z_s}$ where $z_s = B(s)$. Finally, Pirotta et al. (2024) showed that FB models can be used to implement different imitation learning criteria. In particular, we recall the *empirical reward via FB* approach where, given a demonstration [3] $\tau = (s_1, \ldots, s_n)$ from an expert policy, the zero-shot inference returns $z_\tau = \text{ER}_{\text{FB}}(\tau) = \frac{1}{n} \sum_{i=1}^{n} B(s_i)$.

In the limit of $d$ and full coverage of $\rho$, FB can learn optimal policies for any reward function and solve any imitation learning problem (Touati & Ollivier, 2021). However, when $d$ is finite, FB training has a limited inductive bias on which policies to favor, except for the low-rank dynamics assumption, and when the dataset has poor coverage, it cannot reliably optimize policies using offline learning. In this case, FB models tend to *collapse* to few policies with poor downstream performance on tasks of interest (see experiments on walker in App. E).

## 3   FB WITH CONDITIONAL POLICY REGULARIZATION

At pre-training, the agent has access to a *dataset of unlabeled behaviors* $\mathcal{M} = \{\tau\}$, which contains observation-only trajectories $\tau = (s_1, \ldots, s_{\ell(\tau)})$ [4] where states are drawn from a generic distribution $\rho^\tau(X), X \subseteq \mathcal{S}$. Furthermore, the agent can directly interact with the environment from initial states $s_0 \sim \mu$ and we denote by $\mathcal{D}_{\text{online}}$ the associated replay buffer of (unsupervised) transitions.

**FB with conditional policy regularization.** We now describe how we steer the unsupervised training of FB towards capturing the diverse behaviors represented in $\mathcal{M}$. We first outline our formalization of the problem, followed by a detailed discussion of the design choices that enable the development of a scalable and effective algorithm.

In FB, we pretrain a continuous set of latent-conditioned policies $\pi(da|s, z)$, where $z$ is drawn from a distribution $\nu$ defined over the latent space $Z$. The space of behaviors represented by FB can be compactly represented by the joint space $(s, z)$ where $z \sim \nu$ and $s \sim \rho^{\pi_z}$. We denote by $p_\pi(s, z) = \nu(z)\rho^{\pi_z}(s)$ the joint distribution induced by FB over this space. We summarize the behaviors represented in the unlabeled dataset in a similar way by assuming that each trajectory can be produced by some FB policy $\pi_z$. Since the dataset only contains states with no latent variables, for each trajectory $\tau$ we must infer a latent $z$ such that the policy $\pi_z$ would visit the same states as $\tau$. Pirotta et al. (2024) proposed several methods for inferring such latent variables from a single trajectory using an FB model. Among these, we choose to encode trajectories using $\text{ER}_{\text{FB}}$, a simple yet empirically effective method, and represent each trajectory $\tau$ in the dataset as

---

[3]While the original method is defined for multiple trajectories, here we report the single-trajectory case for notation convenience and to match the way we will use it later.

[4]In humanoid, we use motion capture datasets where trajectories may contain noise and artifacts and, in general, are not generated by "purposeful" or stationary policies.

$\{(s, z = \mathrm{ER}_{\mathrm{FB}}(\tau))\}_{s \sim \rho^{\tau}}$. We assume a uniform distribution over $\tau \in \mathcal{M}$ and denote by $p_{\mathcal{M}}(s, z)$ the joint distribution of the dataset induced by this process.

To ensure that FB policies encode similar behaviors to the ones represented in the dataset, we regularize the unsupervised training of the FB actor with a distribution-matching objective that minimizes the discrepancy between $p_{\pi}(z, s)$ and $p_{\mathcal{M}}(z, s)$. This results in the following actor training loss:

$$\mathscr{L}_{\mathrm{FB\text{-}CPR}}(\pi) = -\mathbb{E}_{z \sim \nu, s \sim \mathcal{D}_{\mathrm{online}}, a \sim \pi_z(\cdot|s)} \left[ F(s, a, z)^{\top} z \right] + \alpha \mathrm{KL}(p_{\pi}, p_{\mathcal{M}}), \tag{7}$$

where $\alpha$ is hyper-parameter that controls the strength of the regularization.

**Distribution matching objective.** We now explain how to turn Eq. 7 into a tractable RL procedure. The key idea is that we can interpret the KL-divergence as an expected return under the polices $\pi_z$ where the reward is given by the log-ratio $p_{\mathcal{M}}(s, z)/p_{\pi}(s, z)$ of the two distributions,

$$\mathrm{KL}(p_{\pi}, p_{\mathcal{M}}) = \mathbb{E}_{\substack{z \sim \nu, \\ s \sim \rho^{\pi_z}}} \left[ \log \frac{p_{\pi}(s, z)}{p_{\mathcal{M}}(s, z)} \right] = -\mathbb{E}_{z \sim \nu} \mathbb{E} \left[ \sum_{t=0}^{\infty} \gamma^t \log \frac{p_{\mathcal{M}}(s_{t+1}, z)}{p_{\pi}(s_{t+1}, z)} \Big| s_0 \sim \mu, \pi_z \right], \tag{8}$$

To estimate the reward term, we employ a variational representation of the Jensen-Shannon divergence. Specifically, we introduce a discriminator network $D : S \times Z \to [0, 1]$ conditioned on the latent $z$ and train it with a GAN-like objective (Goodfellow et al., 2014),

$$\mathscr{L}_{\mathrm{discriminator}}(D) = -\mathbb{E}_{\tau \sim \mathcal{M}, s \sim \rho^{\tau}} \left[ \log(D(s, \mathrm{ER}_{\mathrm{FB}}(\tau))) \right] - \mathbb{E}_{z \sim \nu, s \sim \rho^{\pi_z}} \left[ \log(1 - D(s, z)) \right]. \tag{9}$$

It is known that the optimal discriminator for the loss in Eq. 9 is $D^{\star} = \frac{p_{\mathcal{M}}}{p_{\pi} + p_{\mathcal{M}}}$ (e.g., Goodfellow et al., 2014; Nowozin et al., 2016), which allows us approximating the log-ratio reward function as $\log \frac{p_{\mathcal{M}}}{p_{\pi}} \approx \log \frac{D}{1-D}$. We can then fit a critic network $Q$ to estimate the action-value of this approximate reward via off-policy TD learning,

$$\mathscr{L}_{\mathrm{critic}}(Q) = \mathbb{E}_{\substack{(s,a,s') \sim \mathcal{D}_{\mathrm{online}} \\ z \sim \nu, a' \sim \pi_z(\cdot|s')}} \left[ \left( Q(s, a, z) - \log \frac{D(s', z)}{1 - D(s', z)} - \gamma \overline{Q}(s', a', z) \right)^2 \right]. \tag{10}$$

This leads us to the final actor loss for FB-CPR,

$$\mathscr{L}_{\mathrm{FB\text{-}CPR}}(\pi) = -\mathbb{E}_{z \sim \nu, s \sim \mathcal{D}_{\mathrm{online}}, a \sim \pi_z(\cdot|s)} \left[ F(s, a, z)^{\top} z + \alpha Q(s, a, z) \right]. \tag{11}$$

**Latent space distribution.** So far, we have not specified the distribution $\nu$ over the latent space $Z$. According to the FB optimality criteria (Touati & Ollivier, 2021), it is sufficient to choose a distribution that has support over the entire hypersphere. However, in practice, we can leverage $\nu$ to allocate more model capacity to meaningful latent tasks and to enhance the training signal provided by and to the discriminator, while ensuring generalization over a variety of tasks. In particular, we choose $\nu$ as a mixture of three components: **1)** $z = \mathrm{ER}_{\mathrm{FB}}(\tau)$ where $\tau \sim \mathcal{M}$, which encourages FB to accurately reproduce each trajectory in the unlabeled dataset, thus generating challenging samples for the discriminator and boosting its training signal; **2)** $z = B(s)$ where $s \in \mathcal{D}_{\mathrm{online}}$, which focuses on goal-reaching tasks for states observed during the training process; and **3)** uniform over the hypersphere, which allocates capacity for broader tasks and covers the latent space exhaustively.

**Online training and off-policy implementation.** FB-CPR is pre-trained online, interleaving environment interactions with model updates. During interaction, we sample $N$ policies with $z \sim \nu$ and rollout each for a fixed number steps. All the collected (unsupervised) transitions are added to a finite capacity replay buffer $\mathcal{D}_{\mathrm{online}}$. We then use an off-policy procedure to update all components of FB-CPR: $F$ and $B$ using Eq. 5, the discriminator $D$ using Eq. 9, the critic $Q$ using Eq. 10, and the actor $\pi$ using equation 11. The full pseudo-code of the algorithm is reported in App. B.

**Discussion.** While the distribution matching term in Eq. 8 is closely related to existing imitation learning schemes, it has crucial differences that makes it more suitable for our problem. Peng et al. (2022) and Vlastelica et al. (2024) focus on the state marginal version of $p_{\pi}$ and $p_{\mathcal{M}}$, thus regularizing towards policies that globally cover the same states as the behaviors in $\mathcal{M}$. In general, this may lead to situations where no policy can accurately reproduce the trajectories in $\mathcal{M}$. Tessler et al. (2023) address this problem by employing a conditional objective similar to Eq. 8, where a trajectory encoder is learned end-to-end together with the policy space $(\pi_z)$. In our case, distribution matching is used to regularize the FB unsupervised learning process and we directly use $\mathrm{ER}_{\mathrm{FB}}$ to embed trajectories into the latent policy space. Not only this simplifies the learning process by removing an ad-hoc trajectory encoding, but it also binds FB and policy training together, thus ensuring a more stable and consistent learning algorithm.

## 4  EXPERIMENTS ON HUMANOID

We propose a novel suite of whole-body humanoid control tasks based on the SMPL humanoid (Loper et al., 2015), which is widely adopted in virtual character animation (e.g., Luo et al., 2021; 2024a). The SMPL skeleton contains 24 rigid bodies, of which 23 are actuated. The body proportion can vary based on a body shape parameter, but in this work we use a neutral body shape. The state consists of proprioceptive observations containing body pose (70D), body rotation (144D), and linear and angular velocities (144D), resulting in a state space $S \subseteq \mathbb{R}^{358}$. All the components of the state are normalized w.r.t. the current facing direction and root position (e.g., Won et al., 2022; Luo et al., 2023). We use a proportional derivative (PD) controller and the action space $A \subseteq [-1, 1]^{69}$ thus specifies the "normalized" PD target. Unlike previous work, which considered an under-constrained skeleton and over-actuated controllers, we define joint ranges and torque limits to create "physically plausible" movements. The simulation is performed using MuJoCo (Todorov et al., 2012) at 450 Hz, while the control frequency is 30 Hz. More details in App. C.1.

**Motion datasets.** For the behavior dataset we use a subset of the popular AMASS motion-capture dataset (Mahmood et al., 2019), which contains a combination of short, task-specific motions (e.g., few seconds of running or walking), long mixed behaviors (e.g., more than 3 minutes of dancing or daily house activities) and almost static motions (e.g., greeting, throwing). Following previous approaches (e.g., Luo et al., 2021; 2023; 2024b), we removed motions involving interactions with objects (e.g., stepping on boxes). After a $10\%$ train-test split, we obtained a train dataset $\mathcal{M}$ of 8902 motions and a test dataset $\mathcal{M}_{\text{TEST}}$ of 990 motions, with a total duration of approximately 29 hours and 3 hours, respectively (see Tab. 2 in App. C.2). Motions are action-free, comprising only body position and orientation information, which we supplement with estimated velocities using a finite difference method. Some motions may exhibit variations in frequency, discontinuities such as joint flickering, or artifacts like body penetration, making exact reproduction impossible in simulation, thereby increasing the realism and complexity of our experimental setting.

**Downstream tasks and metrics.** The evaluation suite comprises three categories (see App. C.3 for details): **1)** *reward optimization*, which involves 45 rewards designed to elicit a range of behaviors, including static/slow and dynamic/fast movements that require control of different body parts and movement at various heights. The performance is evaluated based on the average return over episodes of 300 steps, with some reward functions yielding policies similar to motions in the dataset and others resulting in distinct behaviors. **2)** *goal reaching*, where the model's ability to reach a goal from an arbitrary initial condition is assessed using 50 manually selected "stable" poses. Two metrics are employed: success rate, indicating whether the goal position has been attained at any point, and proximity, calculated as the normalized distance to the goal position averaged over time. **3)** *tracking*, which assesses the model's capacity to reproduce a target motion when starting from its initial pose. A motion is considered successfully tracked if the agent remains within a specified distance (in joint position and rotation) to the motion along its entire length (Luo et al., 2021). Additionally, the earth mover's distance (Rubner et al., 2000, EMD) is used as a less-restrictive metric that does not require perfect time-alignment between the agent's trajectory and the target motion.

**Protocol and baselines.** We first define single-task baselines for each category. We use TD3 (Fujimoto et al., 2018) trained from scratch for each reward-maximization and goal-reaching task. We also train Goal-GAIL (Ding et al., 2019) and PHC (Luo et al., 2023) on each individual motion to have strong baselines for motion tracking. All the algorithms are trained online.[5] We then consider "multi-task" unsupervised RL algorithms. Goal-GAIL and Goal-TD3 are state-of-the-art goal-conditioned RL algorithms. PHC is a goal-conditioned algorithm specialized for motion tracking and CALM (Tessler et al., 2023) is an algorithm for behavior-conditioned imitation learning. All these baselines are trained online and leverage $\mathcal{M}$ in the process. ASE (Peng et al., 2022) is the closest BFM approach to ours as it allows for zero-shot learning and leverages motions for regularization. We train ASE online with $\mathcal{M}$ using an off-policy routine. An extensive comparison to other unsupervised skill discovery methods is reported in App. H. We also test planning-based approaches such as MPPI (Williams et al., 2017), DIFFUSER (Janner et al., 2022) and H-GAP (Jiang et al., 2024). All these methods are offline and require action-labeled datasets. For this purpose, we first create an action-labeled version of the AMASS dataset by replaying policies from single-motion

---

[5]We pick the best performance over 5 seeds for reward and goal-based tasks, and run only one seed for single-motion tracking due to the high volume of motions. Standard deviations are thus omitted in Tab. 1.

| Algorithm | Reward (↑) | Goal | | Tracking - EMD (↓) | | Tracking - Success (↑) | |
|---|---|---|---|---|---|---|---|
| | | Proximity (↑) | Success (↑) | Train | Test | Train | Test |
| TD3† | 249.74 | 0.98 | 0.98 | | | | |
| GOAL-GAIL† | | | | 1.08 | 1.09 | 0.22 | 0.23 |
| PHC† | | | | 1.14 | 1.14 | 0.94 | 0.94 |
| ORACLE MPPI† | 178.50 | 0.47 | 0.73 | | | | |
| GOAL-TD3 | | 0.67 (0.34) | 0.44 (0.47) | 1.39 (0.08) | 1.41 (0.09) | 0.90 (0.01) | 0.91 (0.01) |
| GOAL-GAIL | | 0.61 (0.35) | 0.35 (0.44) | 1.68 (0.02) | 1.70 (0.02) | 0.25 (0.01) | 0.25 (0.02) |
| PHC | | 0.07 (0.11) | 0.05 (0.11) | 1.66 (0.06) | 1.65 (0.07) | 0.82 (0.01) | 0.83 (0.02) |
| CALM | | 0.18 (0.27) | 0.04 (0.17) | 1.67 (0.02) | 1.70 (0.03) | 0.71 (0.02) | 0.73 (0.02) |
| ASE | 105.73 (3.82) | 0.46 (0.37) | 0.22 (0.37) | 2.00 (0.02) | 1.99 (0.02) | 0.37 (0.02) | 0.40 (0.03) |
| DIFFUSER | 85.27 (0.99) | 0.20 (0.03) | 0.14 (0.01) | | | | |
| FB-CPR | 151.68 (7.53) | 0.68 (0.35) | 0.48 (0.46) | 1.37 (0.00) | 1.39 (0.01) | 0.83 (0.01) | 0.83 (0.01) |
| SCORE$_{norm}$ | 0.61 | 0.69 | 0.48 | 0.80 | 0.80 | 0.88 | 0.88 |

Table 1: Summary results comparing FB-CPR to different single-task baselines (i.e., retrained for each task) and "multi-task" unsupervised baselines across three different evaluation categories. We report mean and standard deviation across 5 seeds. For FB-CPR we report the normalized performance against the best algorithm, i.e., $\text{SCORE}_{norm} = \mathbb{E}_{task}[\text{FB-CPR(task)}/\text{BEST(task)}]$. Note that the best algorithm may vary depending on the metric being evaluated (TD3 for reward and goal, Goal-GAIL for tracking EMD and PHC for tracking success). For each metric, we highlight the best "multi-task" baseline and the second best "multi-task" baseline. † are top-liner runs on individual tasks, goals or motions (we use the best performance over seeds).

Goal-GAIL and then combine it with the replay buffer generated by FB-CPR to obtain a diverse dataset with good coverage that can be used for offline training (more details in App. C.1).

We use a comparable architecture and hyperparameter search for all models. Online algorithms are trained for 3M gradient steps corresponding to 30M interaction steps. Evaluation is done by averaging results over 100 episodes for reward and goal, and with a single episode for tracking, as the initial state is fixed. Due to the high computational cost, we were able to compute metrics over only 20 episodes for MPPI and DIFFUSER. We provide further implementation details in App. C.5.

## 4.1 MAIN RESULTS

Table 1 presents the aggregate performance of each algorithm for each evaluation category. MPPI with a learned model and H-GAP exhibit poor performance in all tasks, thus their results are not included in the table (see App. D.1); instead, an oracle version of MPPI serves as a planning-based top-line. On average, FB-CPR achieves 73.4% of the top-line algorithms' performance across all categories, a remarkable result given its lack of explicit training for downstream tasks and ability to perform zero-shot inference without additional learning or planning. Furthermore, FB-CPR outperforms ASE by more than 1.4 times in each task category and matches or surpasses specialized unsupervised RL algorithms. We now provide an in-depth analysis of each category, while a finer breakdown of the results is available in App. D.1.

**Reward-maximization.** In reward-based tasks FB-CPR achieves 61% of the performance of TD3, which is re-trained from scratch for each reward. Compared to unsupervised baselines, FB-CPR outperforms all the baselines that requires planning on a learned model. For example, FB-CPR achieves 177% of the performance of DIFFUSER that relies on a larger and more complex model to perform reward optimization. ORACLEMPPI performs better than FB-CPR, while still lagging behind model-free TD3. This improvement (+17.8% w.r.t. FB-CPR) comes at the cost of a significant increase in computational cost. ORACLEMPPI requires at least 30 minutes to complete a 300 step episode compared to the 12 seconds needed by FB-CPR to perform inference and execute the policy (about 7, 3 and 2 seconds for reward relabeling, inference, and policy rollout). DIFFUSER takes even more, about 5 hours for a single episode. While this comparison is subject to specific implementation details, it provides an interesting comparison between pre-training zero-shot policies and using test-time compute for planning. Finally, ASE, which has the same zero-shot properties as FB-CPR, only achieves 70% of its performance across all tasks.

**Goal-reaching.** Table 1 shows that FB-CPR performs similarly to specialized goal-based baselines (i.e., Goal-GAIL and Goal-TD3) and outperforms the zero-shot baseline (48% and 118% performance increase w.r.t. ASE on proximity and success). When compared with planning-based ap-

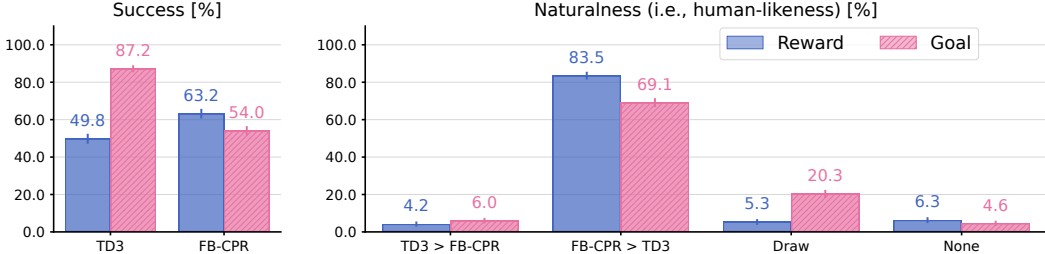

Figure 3: Human-evaluation. Left figure reports the percentage of times a behavior solved a reward-based (blue) or a goal-reaching (pink) task (tasks are independently evaluated). Right figure reports the score for human-likeness by direct comparison of the two algorithms.

proaches, FB-CPR achieves a higher proximity but lower success rate. This means that FB-CPR is able to spend more time close to the goal, whereas ORACLEMPPI is able to reach the goal but not keeping a stable pose thereafter. We believe this is due to the fact that ORACLEMPPI aims to minimize only the distance w.r.t. position at planning without considering velocities.[6] Finally, similarly to the reward case, all other algorithms under-perform w.r.t. TD3 trained to reach each individual goal independently.[7] Since Goal-TD3 is trained using the same reward signal, the conjecture is that the unsupervised algorithm learns behaviors that are biased by the demonstrations. Indeed, by visually inspecting the motions, we noticed that TD3 tends to reach the goal in a faster way, while sacrificing the "quality" of the behaviors (further details below).

**Tracking.** We first notice that the same algorithm may have quite different success and EMD metrics. This is the case for Goal-GAIL, which achieves low EMD but quite poor success rate. This is due to the fact that Goal-GAIL is trained to reach the goal in a few steps, rather than in a single step. On the other hand, Goal-TD3 is trained to reach the goal in the shortest time possible and obtain good scores in both EMD and success metrics. We thus used two different algorithms trained on single motions for the top-line performance in EMD (Goal-GAIL) and success (PHC). The performance of FB-CPR is about $80\%$ and $88\%$ of the top-line scorer for EMD and success, and it achieves an overall 83% success rate on the test dataset. Similarly to previous categories, FB-CPR outperforms both zero-shot and planning-based baselines. Among "multi-task" baselines, only Goal-TD3 is able to do better than FB-CPR on average (about $9\%$ improvement in success and a $1\%$ drop in EMD). Interestingly, PHC achieves the same performance of FB-CPR despite being an algorithm designed specifically for tracking[8]. Due to the high computation cost, we were not able to test MPPI and DIFFUSER on tracking.

**Qualitative Evaluation.** A qualitative evaluation was conducted to assess the quality of learned behaviors, as quantitative metrics alone do not capture this aspect. In line with previous work (Hansen et al., 2024a), we employed 50 human evaluators to compare clips generated by TD3 and FB-CPR for episodes of the same task. The evaluation involved rating whether the model solved the task or achieved the goal, and which model exhibited more natural behavior (see App. D.4 for details). This study encompassed all 45 rewards and 50 goals, with results indicating that despite TD3 achieving higher rewards, both algorithms demonstrated similar success rates in reward-based tasks, producing intended behaviors such as jumping and moving forward (cf. Fig. 3). Notably, FB-CPR was perceived as more human-like in 83% of cases, whereas TD3 was considered more natural in only 4% of cases. This disparity highlights the issue of underspecified reward functions and how motion regularization in FB-CPR compensates for it by capturing human-like biases. In App. D.4.2, we provide further examples of this "human bias" in underspecified and composed rewards. In goal-reaching tasks, human evaluators' assessments of success aligned with our qualitative analysis, showing that FB-CPR exhibited a 6% improvement while TD3 experienced an 11% drop. Furthermore, FB-CPR was deemed more human-like in 69% of cases, even though TD3 had a higher success rate. In the remaining cases, evaluators considered TD3 and FB-CPR equally good for 20% of the goals, while

---

[6]We tried to train with a full distance (i.e., position and velocities) but we did not get any significant result.

[7]TD3 is trained using the full distance to the goal as reward function.

[8]The original PPO-based implementation of PHC (Luo et al., 2024b) achieves 0.95 tracking accuracy on both the train and test set, but leverages information not available to FB-CPR (e.g., global positions).

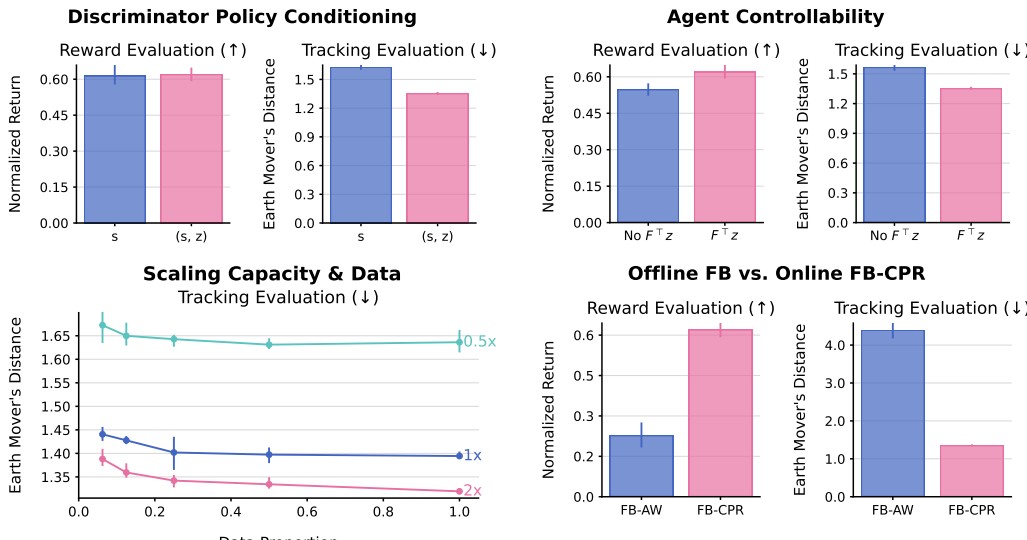

Figure 4: **FB-CPR Ablations.** (TOP LEFT) Ablating the FB-CPR discriminator's policy conditioning. (TOP RIGHT) Ablating the contribution of $F(z)^\top z$ in the FB-CPR actor loss (Eq. 11). (BOTTOM LEFT) The effect of increasing model capacity along with the number of motions in the dataset $\mathcal{M}$. (BOTTOM RIGHT) Contrasting Advantage-Weighed FB (FB-AW) trained from a large diverse offline dataset versus FB-CPR trained fully online with policy regularization. All ablations are averaged over 5 seeds with ranges representing bootstrapped 95% confidence intervals.

TD3 was better in only 6% of the goals. Finally, we report additional qualitative investigation on the embedding and the space of policies in App. G.

## 4.2 ABLATIONS

Various design decisions have gone into FB-CPR that deserves further analysis. In the following, we seek to answer key questions surrounding the necessity of online interaction and how components of our algorithm affect different axes of performance. Additionally, Appendix D.2 provides further ablations on design decisions regarding the FB-CPR discriminator, sampling distribution $\nu$, and other forms of policy regularization when provided action labels.

**Is online policy regularization necessary given a large diverse dataset?** Prior works on unsupervised RL have relied on large and diverse datasets that contain sufficient coverage of any downstream task. If such a dataset exists is there anything to be gained from the guided approach of online FB-CPR outlined herein? In order to test this hypothesis, we evaluate training offline FB with an advantage weighted actor update (Nair et al., 2020) (FB-AW) which compensates for overestimation when performing policy optimization with an offline dataset (Cetin et al., 2024b). As no dataset with our criterion exists, we curate a dataset by collating all 30M transition from an online FB-CPR agent. The offline agent is trained for the same total number of gradients steps as the online agent and all hypereparameters shared between the two methods remain fixed. In the bottom right quadrant of Figure 4, we can see that FB-AW perform substantially worse than FB-CPR highlighting the difficulty of offline policy optimization and the efficacy of guiding online interactions through the conditional policy regularization of FB-CPR.

**How important is maximizing the unsupervised RL term $F(z)^\top z$?** The primary mechanism by which FB-CPR regularizes its policy is through the discriminator's critic (Eq. 10). This begs the question to what extent is maximizing the unsupervised value-function $F(s, a, z)^\top z$ contributes to the overall performance of FB-CPR. To answer this question, we train FB-CPR while omitting this unsupervised term when updating the actor. This has the effect of reducing FB-CPR to be more akin to CALM (Tessler et al., 2023), except that our motions are encoded with FB through $ER_{FB}$. These results are presented in top right quadrant of Figure 4 for both reward and tracking-based performance measures. We can see that including the unsupervised value-function from FB results

in improved performance in both reward and tracking evaluation emphasizing that FB is providing much more than just a motion encoder through $\text{ER}_{\text{FB}}$.

**How important is policy conditioning for the discriminator?** FB-CPR relies on a latent-conditional discriminator to evaluate the distance between a specific motion and a policy selected through the trajectory embedding of $\text{ER}_{\text{FB}}$. We hypothesize that this policy-conditioned discriminator should provide a stronger signal to the agent and lead to better overall performance. We test this hypothesis by comparing FB-CPR with a discriminator that solely depends on state, thus converting the regularization term into a marginal state distribution matching. The top left quadrant of Figure 4 shows that the latent-conditioned discriminator outperforms the state-only configuration in tracking tasks while performing similarly in reward tasks. These findings demonstrate the importance of the $\text{ER}_{\text{FB}}$ embedding in enabling FB-CPR to more accurately reproduce motions.

**How does network capacity and expert dataset size impact FB-CPR performance?** Many recent works in RL have shown vast performance improvements when scaling the capacity of neural networks (Schwarzer et al., 2023; Obando-Ceron et al., 2024; Nauman et al., 2024) along with dataset size (Brohan et al., 2023; Zitkovich et al., 2023) or task diversity (Kumar et al., 2023; Ali Taïga et al., 2023). Given these findings, we seek to understand the capabilities of FB-CPR when scaling both the network capacity and the number of expert demonstrations. To this end, we perform a grid sweep over three configurations of model sizes that alters the amount of compute by roughly $\{0.5\times, 1\times, 2\times\}$ of the base models; as well as datasets that are $\{6.25\%, 12.5\%, 25\%, 50\%, 100\%\}$ the size of our largest motion dataset via subsampling. For each of these combinations we report the tracking performance on all motions and present these results in the bottom left quadrant of Figure 4 with additional evaluation metrics in Appendix D.2. Consistent with prior results we can see that larger capacity models are better able to leverage larger motion datasets resulting in significantly improved performance for our $2\times$ larger model over the results of the $1\times$ model reported in Table 1.

**Scaling FB-CPR to very deep architectures**. To scale further and avoid vanishing/exploding gradients, we replace MLP layers with blocks akin to those of transformer architectures (Vaswani, 2017), involving residual connections, layer normalization, and Mish activation functions (Misra, 2019). With this simple modification, we could train our largest and most capable model, outperforming our base model both in size (from 25M to 288M parameters) and performance (see table below).

| Algorithm | Reward (↑) | Goal | | Tracking - EMD (↓) | | Tracking - Success (↑) | |
|---|---|---|---|---|---|---|---|
| | | Proximity (↑) | Success (↑) | Train | Test | Train | Test |
| FB-CPR | 179.94 | 0.82 | 0.66 | 1.11 | 1.13 | 0.84 | 0.84 |
| $\text{SCORE}_{\text{norm}}$ | 0.72 | 0.84 | 0.67 | 0.97 | 0.96 | 0.89 | 0.89 |

## 5 Conclusions

We introduced FB-CPR, a novel algorithm combining the zero-shot properties of FB models with a regularization grounding online training and policy learning on a dataset of unlabeled behaviors. We demonstrated the effectiveness of FB-CPR by training the first BFM for zero-shot control of a complex humanoid agent with state-of-the-art performance across a variety of tasks.

While FB-CPR effectively grounds unsupervised RL with behavior trajectories, a theoretical understanding of its components is still lacking and alternative formulations may be possible. In practice, FB-CPR struggles with problems far from motion-capture datasets, such as tracking motions or solving reward-based tasks involving ground movements. Although FB-CPR produces more human-like behaviors than pure reward-optimization algorithms and achieves good tracking performance, it sometimes generates imperfect and unnatural movements, particularly for behaviors like falling or standing. The BFM trained with FB-CPR is limited to proprioceptive observations and cannot solve tasks requiring environmental navigation or object interaction. Integrating additional state variables, including complex perception, could allow models to tackle harder tasks, but this might necessitate test-time planning or fast online adaptation. Currently, FB-CPR relies on expensive motion capture datasets; extending it to leverage videos of various human activities could refine and expand its capabilities. Finally, while language prompting could be added by leveraging text-to-motion models to set tracking targets, an interesting research direction is to align language and policies more directly.

ACKNOWLEDGMENTS

The authors thank Zhengyi Luo for providing the tracking evaluation of the original PHC implementation on our humanoid environment. The authors thank Claire Roberts, Dominic Burt, Jiemin Zhang, Leonel Sentana, Maria Ruiz, Matt Hanson, Morteza Behrooz, Ryan Winstead, Spaso Ilievski, Vincent Moens, Vlad Bodurov, William Ngan for significant contributions to the project management and the development of the interactive demo and website at https://metamotivo.metademolab.com.

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

# Appendices

## A   RELATED WORK

**RL for Humanoid Control.** Controlling a humanoid agent is considered a major objective for both in robotic (UniTree, 2024; Dynamics, 2024) and simulated (Peng et al., 2021; Won et al., 2022; Luo et al., 2024a) domains and it has emerged as a major challenge for reinforcement learning due to its high dimensionality and intrinsic instability. In robotics, a predominant approach is to perform direct behavior cloning of task-specific demonstrations (e.g., Seo et al., 2023) or combing imitation and reinforcement learning (RL) to regularize task-driven policies by using human-like priors (e.g., Cheng et al., 2024). In virtual domains, RL is often used for physics-based character animation by leveraging motion-capture datasets to perform motion tracking (Luo et al., 2023; Merel et al., 2019; Wagener et al., 2022; Reda et al., 2023) or to learn policies solving specific tasks, such as locomotion or manipulation (Luo et al., 2024c; Wang et al., 2023; Hansen et al., 2024a). Despite its popularity across different research communities, no well-established platform, data, or benchmark for multi-task whole-body humanoid control is available. Standard simulation platforms such as `dm_control` (Tunyasuvunakool et al., 2020) or IsaacGym (Makoviychuk et al., 2021) employ different humanoid skeletons and propose only a handful of reward-based tasks. Luo et al.

(2024c) and Sferrazza et al. (2024) recently introduced a broader suite of humanoid tasks, but they all require task-specific observations to include object interaction and world navigation. Regarding datasets, MoCapAct Wagener et al. (2022) relies on CMU motion capture data mapped onto a CMU humanoid skeleton, Peng et al. (2022) uses a well curated animation dataset related to a few specific movements mapped onto the IsaacGym humanoid, and Luo et al. (2023) use the AMASS dataset mapped to an SMPL skeleton.

**Unsupervised RL.** Pre-trained unsupervised representations from interaction data (Yarats et al., 2021; Schwarzer et al., 2021; Farebrother et al., 2023) or passive data (Baker et al., 2022; Ma et al., 2023; Brandfonbrener et al., 2023; Ghosh et al., 2023), such as unlabeled videos, significantly reduce the sample complexity and improve performance in solving downstream tasks such as goal-based, reward-based, or imitation learning by providing effective state embeddings that simplify observations (e.g., image-based RL) and capture the dynamical features of the dynamics. Another option is to pre-train a set of policies through skill diversity metrics (e.g. Gregor et al., 2016; Eysenbach et al., 2019; Sharma et al., 2020; Laskin et al., 2022; Klissarov & Machado, 2023; Park et al., 2024c) or exploration-driven metrics (e.g. Pathak et al., 2017; Machado et al., 2020; Mendonca et al., 2021; Rajeswar et al., 2023) that can serve as behavior priors. While both pre-trained representations and policies can greatly reduce sample complexity and improve performance, a full RL model still needs to be trained from scratch to solve any downstream task.

**Zero-shot RL.** Goal-conditioned methods (Andrychowicz et al., 2017; Pong et al., 2020; Warde-Farley et al., 2019; Mezghani et al., 2022; Ma et al., 2022; Park et al., 2023) train goal-conditioned policies to reach any goal state from any other state. While they are the most classical form of zero-shot RL, they are limited to learn goal-reaching behaviors. Successor features based methods are the most related to our approach. They achieve zero-shot capabilities by modeling a discounted sum of state features learned via low-rank decomposition (Touati & Ollivier, 2021; Touati et al., 2023; Pirotta et al., 2024; Jeen et al., 2024) or Hilbert representation (Park et al., 2024b). One of the key advantages of these methods is their low inference complexity, as they can infer a near-optimal policy for a given task through a simple regression problem. Generalized occupancy models (Zhu et al., 2024) learn a distribution of successor features but requires planning for solving novel downstream tasks. Building general world models is another popular technique (Yu et al., 2023; Ding et al., 2024; Jiang et al., 2024) for zero-shot RL when combined with search/planning algorithms (e.g. Williams et al., 2017; Howell et al., 2022). While this category hold the promise of being zero-shot, several successful world-modeling algorithms uses a task-aware training to obtain the best downstream task performance (Hansen et al., 2024b;a; Hafner et al., 2024; Sikchi et al., 2022). Finally, recent works (Frans et al., 2024; Ingebrand et al., 2024) have achieved zero-shot capabilities by learning an encoding of reward function at pre-train time by generating random unsupervised rewards.

**Integrating demonstrations.** Our method is related to the vast literature of learning from demonstrations. Transformer-based approaches have became a popular solution for integrating expert demonstrations in the learning process. The simplest solution is to pre-train a model through conditioned or masked behavioral cloning (Cui et al., 2023; Shafiullah et al., 2022; Schmidhuber, 2019; Chen et al., 2021; Liu et al., 2022; Wu et al., 2023; Jiang et al., 2023). If provided with sufficiently curated expert datasets at pre-training, these models can be prompted with different information (e.g., state, reward, etc) to solve various downstream tasks. While these models are used in a purely generative way, H-GAP (Jiang et al., 2024) combines them with model predictive control to optimize policies that solve downstream tasks. Similar works leverage diffusion models as an alternative to transformer architectures for conditioned trajectory generation (e.g., Pearce et al., 2023; He et al., 2023) or to solve downstream tasks via planning (Janner et al., 2022). Another popular approach is to rely on discriminator-based techniques to integrate demonstrations into an RL model either for imitation (e.g., Ho & Ermon, 2016; Ding et al., 2019; Tessler et al., 2023), reward-driven (hierarchical) tasks (Peng et al., 2021; Gehring et al., 2021; 2023; Vlastelica et al., 2024) or zero-shot (Peng et al., 2022)[9]. When the demonstrations are of "good" quality, the demonstrated behaviors can be distilled into the learned policies by constructing a one-step tracking problem (e.g., Luo et al., 2023; 2024b; Qian et al., 2024). These skills can be then used as behavior priors to train task-oriented controllers using hierarchical RL. Finally, recent papers leverage internet-scale data to learn general controllers for video games or robotic control. These methods leverage curated data with action

---

[9]While the original ASE algorithm is designed to create behavior priors that are then used in a hierarchical RL routine, we show in our experiments that it is possible to leverage the learned discriminator to solve downstream tasks in a zero-shot manner.

labeling (Wang et al., 2024; Team et al., 2024; Zitkovich et al., 2023) or the existence of high-level API for low-level control (Zitkovich et al., 2023).

---

**Algorithm 1** FB-CPR

---

1: **Inputs**: unlabeled dataset $\mathcal{M}$, Polyak coefficient $\zeta$, number of parallel networks $m$, randomly initialized networks $\{F_{\theta_k}\}_{k\in[m]}$, $B_\omega$, $\pi_\phi$, $\{Q_{\eta_k}\}_{k\in[m]}$, $D_\psi$, learning rate $\xi$, batch size $n$, B regularization coefficient $\lambda$, Fz-regularization coefficient $\beta$, actor regularization coefficient $\alpha$, number of rollouts per update $N_{\text{rollouts}}$, rollout length $T_{\text{rollout}}$, z sampling distribution $\nu = (\nu_{\text{online}}, \nu_{\text{unlabeled}})$, sequence length $T_{\text{seq}}$, z relabeling probability $p_{\text{relabel}}$

2: Initialize empty train buffer: $\mathcal{D}_{\text{online}} \leftarrow \emptyset$
3: **for** $t = 1, \ldots$ **do**
4:     **/* Rollout**
5:     **for** $i = 1, \ldots, N_{\text{rollouts}}$ **do**
6:         Sample $z = \begin{cases} B(s) & \text{where } s \sim \mathcal{D}_{\text{online}}, & \text{with prob } \nu_{\text{online}} \\ \frac{1}{T_{\text{seq}}}\sum_{t=1}^{T_{\text{seq}}} B(s_t) & \text{where } \{s_1, \ldots, s_{T_{\text{seq}}}\} \sim \mathcal{M}, & \text{with prob } \tau_{\text{unlabeled}} \\ \sim \mathcal{N}(0, I_d) & & \text{with prob } 1 - \tau_{\text{online}} - \tau_{\text{unlabeled}} \end{cases}$
7:         $z \leftarrow \sqrt{d}\frac{z}{\|z\|_2}$
8:         Rollout $\pi_\phi(\cdot, z)$ for $T_{\text{rollout}}$ steps, and store data into $\mathcal{D}_{\text{train}}$
9:     **end for**
10:     **/* Sampling**
11:     Sample a mini-batch of $n$ transitions $\{(s_i, a_i, s_i', z_i)\}_{i=1}^n$ from $\mathcal{D}_{\text{online}}$
12:     Sample a mini-batch of $\frac{n}{T_{\text{seq}}}$ sequences $\{(s_{j,1}, s_{j,2} \ldots, s_{j,T_{\text{seq}}})\}_{j=1}^{\frac{n}{T_{\text{seq}}}}$ from $\mathcal{M}$
13:     **/* Encode Expert sequences**
14:     $z_j \leftarrow \frac{1}{T_{\text{seq}}}\sum_{t=1}^{T_{\text{seq}}} B(s_{j,t})$ ; $z_j \leftarrow \sqrt{d}\frac{z_j}{\|z_j\|_2}$
15:     **/* Compute discriminator loss**
16:     $\mathscr{L}_{\text{discriminator}}(\psi) = -\frac{1}{n}\sum_{j=1}^{\frac{n}{T_{\text{seq}}}}\sum_{t=1}^{T_{\text{seq}}}\log D_\psi(s_{j,t}, z_j) - \frac{1}{n}\sum_{i=1}^n \log(1 - D_\psi(s_i, z_i))$
17:     **/* Sampling and Relabeling latent variables z**
18:     Set $\forall i \in [i], z_i = \begin{cases} z_i & \text{(no relabel)} & \text{with prob } 1 - p_{\text{relabel}} \\ B(s_k) & \text{where } k \sim \mathcal{U}([n]), & \text{with prob } p_{\text{relabel}} * \tau_{\text{online}} \\ \frac{1}{T_{\text{seq}}}\sum_{t=1}^{T_{\text{seq}}} B(s_{j,t}) & \text{where } j \sim \mathcal{U}([\frac{n}{T_{\text{seq}}}]), & \text{with prob } p_{\text{relabel}} * \tau_{\text{unlabeled}} \\ \sim \mathcal{N}(0, I_d) & & \text{with prob } p_{\text{relabel}} * (1 - \tau_{\text{online}} - \tau_{\text{unlabeled}}) \end{cases}$
19:     **/* Compute FB loss**
20:     Sample $a_i' \sim \pi_\phi(s_i', z_i)$ for all $i \in [n]$
21:     $\mathscr{L}_{\text{FB}}(\theta_k, \omega) = \frac{1}{2n(n-1)}\sum_{i\neq j}\left(F_{\theta_k}(s_i, a_i, z_i)^\top B_\omega(s_j') - \gamma\frac{1}{m}\sum_{l\in[m]}\overline{F_{\theta_l}}(s_i', a_i', z_i)^\top\overline{B_\omega}(s_j')\right)^2$
22:         $-\frac{1}{n}\sum_i F_{\theta_k}(s_i, a_i, z_i)^\top B_\omega(s_i') \forall k \in [m]$
23:     **/* Compute orthonormality regularization loss**
24:     $\mathscr{L}_{\text{ortho}}(\omega) = \frac{1}{2n(n-1)}\sum_{i\neq j}(B_\omega(s_i')^\top B_\omega(s_j'))^2 - \frac{1}{n}\sum_i B_\omega(s_i')^\top B_\omega(s_i')$
25:     **/* Compute Fz-regularization loss**
26:     $\mathscr{L}_{\text{Fz}}(\theta_k) = \frac{1}{n}\sum_{i\in[n]}\left(F_{\theta_k}(s_i, a_i, z_i)^\top z_i - \overline{B_\omega(s_i')^\top \Sigma_B^{-1} z_i} - \gamma\min_{l\in[m]}\overline{F_{\theta_l}}(s_i', a_i', z_i)^\top z_i\right)^2, \forall k$
27:     **/* Compute critic loss**
28:     Compute discriminator reward: $r_i \leftarrow \log(D_\psi(s_i, z_i)) - \log(1 - D_\psi(s_i, z_i)), \quad \forall i \in [n]$
29:     $\mathscr{L}_{\text{critic}}(\eta_k) = \frac{1}{n}\sum_{i\in[n]}\left(Q_{\eta_k}(s_i, a_i, z_i) - r_i - \gamma\min_{l\in[m]}\overline{Q_{\eta_l}}(s_i', a_i', z_i)\right)^2, \quad \forall k \in [m]$
30:     **/* Compute actor loss**
31:     Sample $a_i^\phi \sim \pi_\phi(s_i, z_i)$ for all $i \in [n]$
32:     Let $\overline{F} \leftarrow \text{stopgrad}\left(\frac{1}{n}\sum_{i=1}^n |\min_{l\in[m]} F_{\theta_l}(s_i, a_i^\phi, z_i)^T z_i|\right)$
33:     $\mathscr{L}_{\text{actor}}(\phi) = -\frac{1}{n}\sum_{i=1}^n\left(\min_{l\in[m]} F_{\theta_l}(s_i, a_i^\phi, z_i)^T z_i + \alpha\overline{F}\min_{l\in[m]} J_{\theta_l}(s_i, a_i^\phi, z_i)\right)$
34:     **/* Update all networks**
35:     $\psi \leftarrow \psi - \xi\nabla_\psi\mathscr{L}_{\text{discriminator}}(\psi)$
36:     $\theta_k \leftarrow \theta_k - \xi\nabla_{\theta_k}(\mathscr{L}_{\text{FB}}(\theta_k, \omega) + \beta\mathscr{L}_{\text{Fz}}(\theta_k))$ for all $k \in [m]$
37:     $\omega \leftarrow \omega - \xi\nabla_\omega(\sum_{l\in[m]}\mathscr{L}_{\text{FB}}(\theta_l, \omega) + \lambda \cdot \mathscr{L}_{\text{ortho}}(\omega))$
38:     $\eta_k \leftarrow \eta_k - \xi\nabla_{\eta_k}\mathscr{L}_{\text{critic}}(\eta_k) \forall k \in [m]$
39:     $\phi \leftarrow \phi - \xi\nabla_\phi\mathscr{L}_{\text{actor}}(\phi)$
40: **end for**

---

## B  ALGORITHMIC DETAILS

In Alg. 1 we provide a detailed pseudo-code of FB-CPR including how all losses are computed. Following Touati et al. (2023), we add two regularization losses to improve FB training: an orthonormality loss pushing the covariance $\Sigma_B = \mathbb{E}[B(s)B(s)^\top]$ of $B$ towards the identity, and a temporal difference loss pushing $F(s, a, z)^\top z$ toward the action-value function of the corresponding reward $B(s)^\top \Sigma_B^{-1} z$. The former is helpful to make sure that $B$ is well-conditioned and does not collapse, while the latter makes $F$ spend more capacity on the directions in $z$ space that matter for policy optimization.

## C  EXPERIMENTAL DETAILS FOR THE HUMANOID ENVIRONMENT

### C.1  THE SMPL MUJOCO MODEL

Our implementation of the humanoid agent is build on the MuJoCo model for SMPL humanoid by Luo (2023). Previous work in this domain considers unconstrained joint and over-actuated controllers with the objective of perfectly matching any behavior in motion datasets and then use the learned policies as frozen behavioral priors to perform hierarchical RL (e.g., Luo et al., 2024b). Unfortunately, this approach strongly relies on motion tracking as the only modality to extract behaviors and it often leads to simulation instabilities during training. Instead, we refined the agent specification and designed more natural joint ranges and PD controllers by building on the `dm_control` (Tunyasuvunakool et al., 2020) CMU humanoid definition and successive iterations based on qualitative evaluation. While this does not prevent the agent to express non-natural behaviors (see e.g., policies optimized purely by reward maximization), it does provide more stability and defines a more reasonable control space. We will release the full agent specification and environment code at a later time for full reproducibility.

### C.2  DATA

The AMASS dataset (Mahmood et al., 2019) unifies 15 different motion capture datasets into a single SMPL-based dataset (Loper et al., 2015). For our purposes, we only consider the kinematic aspects of the dataset and ignore the full meshed body reconstruction. In order to enable the comparison to algorithms that require action-labeled demonstration datasets, we follow a similar procedure to (Wagener et al., 2022) and train a single instance of Goal-GAIL to accurately match each motion in the dataset and then roll out the learned policies to generate a dataset of trajectories with actions. The resulting dataset, named AMASS-Act, contains as many motions as the original AMASS dataset.

As mentioned in the main paper, we select only a subset of the AMASS (AMASS-Act) dataset. Following previous approaches (e.g., Luo et al., 2021; 2023; 2024b), we removed motions involving interactions with objects (e.g., stepping on boxes). We also sub-sampled the BMLhandball dataset to just 50 motions since it contains many redundant behaviors. Finally, we removed two dataset `SSM_synced` and `TCD`. We report several statistics about the datasets in Tab. 2.

### C.3  TASKS AND METRICS

In this section we provide a complete description of the tasks and metrics.

#### C.3.1  REWARD-BASED EVALUATION

Similarly to (Tunyasuvunakool et al., 2020), rewards are defined as a function of next state and optionally action and are normalized, i.e., the reward range is $[0, 1]$. Here we provide a high level description of the 8 categories of rewards, we refer the reader to the code (that we aim to release after the submission) for details.

| Dataset | Train dataset $\mathcal{M}$ | | | | Test dataset $\mathcal{M}_{\text{test}}$ | | | |
|---|---|---|---|---|---|---|---|---|
| | Motion count | Average length | Total Steps | Total Time (s) | Motion count | Average length | Total Steps | Total Time (s) |
| ACCAD | 223 | 189.00 | 42146 | 1404.87 | 25 | 174.48 | 4362 | 145.40 |
| BMLhandball | 45 | 291.18 | 13103 | 436.77 | 5 | 292.40 | 1462 | 48.73 |
| BMLmovi | 1456 | 167.36 | 243683 | 8122.77 | 162 | 165.98 | 26888 | 896.27 |
| BioMotionLab | 1445 | 348.88 | 504134 | 16804.47 | 161 | 266.89 | 42969 | 1432.30 |
| CMU | 1638 | 445.85 | 730307 | 24343.57 | 182 | 485.52 | 88364 | 2945.47 |
| DFaust | 80 | 179.39 | 14351 | 478.37 | 9 | 134.67 | 1212 | 40.40 |
| DanceDB | 23 | 1768.91 | 40685 | 1356.17 | 2 | 855.00 | 1710 | 57.00 |
| EKUT | 124 | 157.49 | 19529 | 650.97 | 14 | 153.00 | 2142 | 71.40 |
| Eyes | 562 | 862.41 | 484677 | 16155.90 | 62 | 872.95 | 54123 | 1804.10 |
| HumanEva | 25 | 540.68 | 13517 | 450.57 | 3 | 582.33 | 1747 | 58.23 |
| KIT | 2858 | 235.56 | 673239 | 22441.30 | 318 | 232.09 | 73806 | 2460.20 |
| MPI | 264 | 974.24 | 257199 | 8573.30 | 29 | 908.59 | 26349 | 878.30 |
| SFU | 30 | 569.37 | 17081 | 569.37 | 3 | 849.67 | 2549 | 84.97 |
| TotalCapture | 33 | 2034.06 | 67124 | 2237.47 | 4 | 1715.50 | 6862 | 228.73 |
| Transitions | 96 | 247.86 | 23795 | 793.17 | 11 | 228.82 | 2517 | 83.90 |
| Total | 8,902 | | 3,144,570 | 29h6m59s | 990 | | 337,062 | 3h7m15s |

Table 2: AMASS statistics split into $\mathcal{M}$ (train) and $\mathcal{M}_{\text{test}}$ (test) datasets.

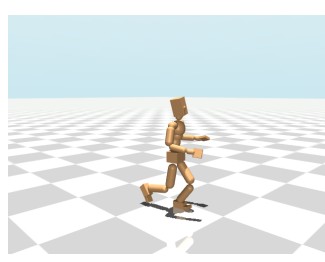

**Locomotion.** This category includes all the reward functions that require the agent to move at a certain speed, in a certain direction and at a certain height. The speed is the xy-linear velocity of the center of mass of the kinematic subtree rooted at the chest. We require the velocity to lie in a small band around the target velocity. The direction defined as angular displacement w.r.t. the robot facing direction, that is computed w.r.t. the chest body. We defined high and low tasks. In high locomotion tasks, we constrain the head z-coordinate to be above a threshold, while in low tasks the agent is encouraged to keep the pelvis z-coordinate inside a predefined range. Finally, we also includes a term penalizing high control actions.[10] We use the following name structure for tasks in this category: `smpl_move-ego-[low-]-{angle}-{speed}`.

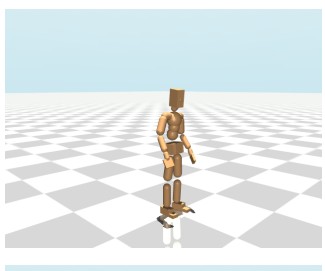

**Standing.** This category includes tasks that require a vertical stable position. Similarly to locomotion we defined standing "high" and "low". These two tasks are obtained from locomotion tasks by setting the speed to 0 (i.e., `smpl_move-ego-[low-]-0-0`).

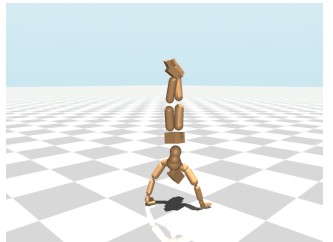

**Handstand.** This is a reverse standing position on the hands (i.e., `smpl_handstand`). To achieve this, the robot must place its feet and head above specific thresholds, with the feet being the highest point and the head being the lowest. Additionally, the robot's velocities and rotations should be zero, and control inputs should be minimal.

---

[10]This is a common penalization used to avoid RL agents to learn rapid unnatural movements. Nonetheless, notice that FB-CPR leverages only state-based information for reward inference through $B(s)$. This means that we entirely rely on the regularized pre-training to learn to avoid high-speed movements.

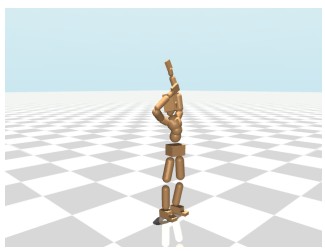

**Arm raising.** Similar to the previous category, this task requires the robot to maintain a standing position while reaching specific vertical positions with its hands, measured at the wrist joints. We define three hand positions: Low (z-range: 0-0.8), Medium (z-range: 1.4-1.6), and High (z-range: 1.8 and above). The left and right hands are controlled independently, resulting in nine distinct tasks. Additionally, we incorporate a penalty component for unnecessary movements and high actions. These tasks are denoted as `smpl_raisearms-{left_pos}-{right_pos}`.

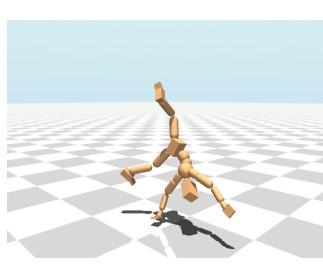

**Rotation.** The tasks in this category require the robot to achieve a specific angular velocity around one of the cardinal axes (x, y, or z) while maintaining proper body alignment. This alignment component is crucial to prevent unwanted movement in other directions. Similar to locomotion tasks, the robot must keep its angular velocity within a narrow range of the target velocity, use minimal control inputs, and maintain a minimum height above the ground, as measured by the pelvis z-coordinate. The tasks in this category are denoted as `smpl_rotate-{axis}-{speed}-{height}`.

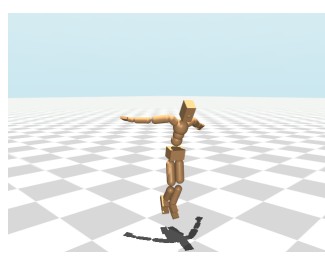

**Jump.** The jump task is defined as reaching a target height with the head while maintaining a sufficiently high vertical velocity. These tasks are named `smpl_jump-{height}`.

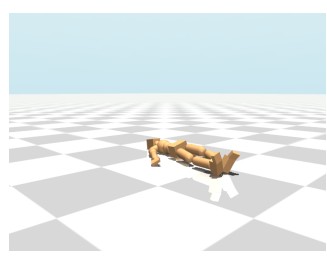

**Ground poses.** This category includes tasks that require the robot to achieve a stable position on the ground, such as sitting, crouching, lying down, and splitting. The sitting task (`smpl_sitonground`) requires the robot's knees to touch the ground, whereas crouching does not have this constraint. The lie-down task has two variants: facing upward (`smpl_lieonground-up`) and facing downward (`smpl_lieonground-down`). Additionally, we define the split task, which is similar to sitting on the ground but requires the robot to spread its feet apart by a certain distance (`smpl_split-{distance}`).

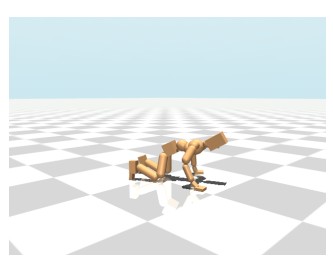

**Crawl.** The crawl task requires the agent to move across the floor in a crawling position, maintaining a specific target height at the spine link. Similar to locomotion tasks, the agent must move in its facing direction at a desired speed. The crawl tasks are denoted as `smpl_crawl-{height}-{speed}-{facing}`. We provide two options for the agent's orientation: crawling while facing downwards (towards the floor) or upwards (towards the sky), with the latter being significantly more challenging.

While our suite allows to generate virtually infinite tasks, we extracted 55 representative tasks for evaluation. See Tab. 18 and Tab. 19 for the complete list. We evaluate the performance of a policy in solving the task via the cumulative return over episodes of $H = 300$ steps: $\mathbb{E}_{s_0 \sim \mu_{\text{test}}, \pi} \left[ \sum_{t=1}^{H} r(a_t, s_{t+1}) \right]$. The initial distribution used in test is a mixture between a random falling position and a subset of the whole AMASS dataset, this is different from the distribution used in training (see App. C.4).

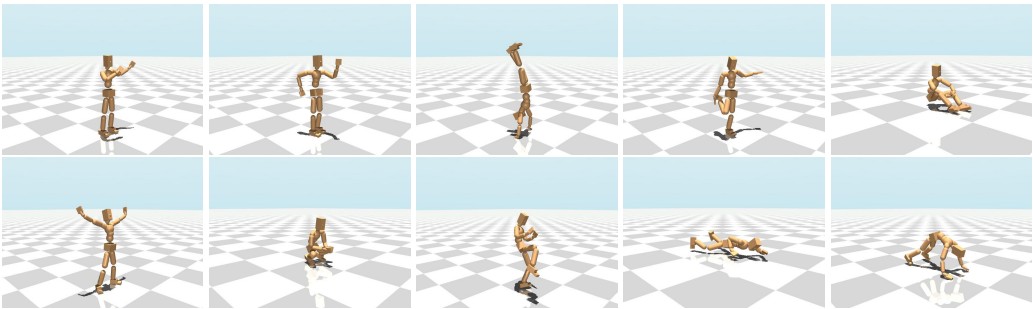

Figure 5: Examples of the poses used for goal-based evaluation.

### C.3.2 MOTION TRACKING EVALUATION

This evaluation aims to assess the ability of the model to accurately replicate a motion, ideally by exactly matching the sequence of motion states. At the beginning of each episode, we initialize the agent in the first state of the motion and simulate as many steps as the motion length. Similarly to (Luo et al., 2021; 2023), we use success to evaluate the ability of the agent to replicate a set of motions. Let $\mathcal{M} = \{\tau_i\}_{i=1}^M$ the set of motions to track and denote by $\tau_i^{\mathfrak{A}}$ the trajectory generated by agent $\mathfrak{A}$ when asked to track $\tau_i$. Then, given a threshold $\xi = 0.5$, we define

$$\text{success}(\mathcal{M}) = \frac{1}{M} \sum_{i=1}^M \mathbb{I}\Big\{\forall t \leq \text{len}(\tau_i) : d_{\text{smpl}}\big(s_t^{\tau_i}, s_t^{\tau_i^{\mathfrak{A}}}\big) \leq \xi\Big\}$$

where $s_t^\tau$ is the state of trajectory $\tau$ at step $t$, $d_{\text{smpl}}(s, s') = \|[X, \theta] - [X', \theta']\|_2$ and $[X, \theta]$ is the subset of the state containing joint positions and rotations. This metric is very restrictive since it requires accurate alignment at each step. Unfortunately, exactly matching the motion at each time step may not be possible due discontinuities (the motion may flicker, i.e., joint position changes abruptly in a way that is not physical), physical constraints (the motion is not physically realizable by our robot), object interaction[11], etc. We thus consider the Earth Mover's Distance (Rubner et al., 2000, EMD) with $d_{\text{smpl}}$ as an additional metric. EMD measures the cost of transforming one distribution into another. In our case, two trajectories that are slightly misaligned in time may still be similar in EMD because the alignment cost is small, while the success metric may still be zero. While these metrics capture different dimensions, if motions are accurately tracked on average, we expect low EMD and high success rate.

### C.3.3 GOAL-BASED EVALUATION

The main challenge in defining goal-based problems for humanoid is to generate target poses that are attainable and (mostly) stable. For this reason, we have manually extracted 50 poses from the motion dataset, 38 from motions in the training dataset and 12 from motions in the test dataset, trying to cover poses involving different heights and different positions for the body parts. In Fig. 5 we report a sample of 10 poses.

In order to assess how close the agent is to the target pose, we use $d_{\text{smpl}}(s, s')$ as in tracking, where the distance is only measured between position and rotation variables, while velocity variables are ignored. Let $g$ be the goal state obtained by setting positions and rotations to the desired pose and velocities to 0, $\beta = 2$ be a threshold parameter, and $\sigma = 2$ be a margin parameter, we then define

---

[11]We curated our datasets but we cannot exclude we missed some non-realizable motion given that this process was hand made.

two evaluation metrics

$$\text{success} = \mathbb{E}_{s_0 \sim \mu_{\text{test}}} \Big[ \mathbb{I}\big\{ \exists t \leq 300 : d_{\text{smpl}}(s_t, g) \leq \beta \big\} \Big];$$

$$\text{proximity} = \mathbb{E}_{s_0 \sim \mu_{\text{test}}} \Bigg[ \frac{1}{300} \sum_{t=1}^{300} \bigg( \mathbb{I}\big\{ d_{\text{smpl}}(s_t, g) \leq \beta \big\}$$
$$+ \mathbb{I}\big\{ d_{\text{smpl}}(s_t, g) > \beta \wedge d_{\text{smpl}}(s_t, g) \leq \beta + \sigma \big\} \Big( \frac{\beta + \sigma - d_{\text{smpl}}(s_t, g)}{\sigma} \Big) \big\} \bigg) \Bigg].$$

The *success* metric matches the standard shortest-path metric, where the problem is solved as soon as the agent reaches a state that is close enough to the goal. The *proximity* metric is computing a "soft" average distance across the full episode of 300 steps. The "score" for each step is 1 if the distance is within the threshold $\beta$, while it decreases linearly down to 0 when the current state is further than $\beta + \sigma$ from the goal. Finally, the metrics are averaged over multiple episodes when starting from initial states randomly sampled from $\mu_{\text{test}}$.

When evaluating FB-CPR, CALM, ASE, and GOAL-GAIL, we need to pass a full goal state $g$, which includes the zero-velocity variables. On the other hand, PHC and GOAL-TD3 are directly trained to match only the position and rotation part of the goal state. Finally, for both MPPI and TD3 directly optimizing for the distance to the pose (i.e., no velocity) led to the better results.

## C.4 Training Protocols

In this section we provide a description of the training protocol, you can refer to the next section for algorithm dependent details. We have two train protocols depending on whether the algorithm is trained online or offline.

**Online training.** The agent interacts with the environment via episodes of fix length $H = 300$ steps. We simulate 50 parallel (and independent) environments at each step. The algorithm has also access to the dataset $\mathcal{M}$ containing observation-only motions. The initial state distribution of an episode is a mixture between randomly generated falling positions (named "Fall" initialization) and states in $\mathcal{M}$ (named "MoCap" initialization[12]). We select the "Fall" modality with probability 0.2. For "MoCap", we use prioritization to sample motions from $\mathcal{M}$ and, inside a motion, the state is uniformly sampled. We change the prioritization during training based on the ability of the agent to track motions. Every 1M interaction steps, we evaluate the tracking performance of the agent on all the motions in $\mathcal{M}$ and update the priorities based on the following scheme. We clip the EMD in $[0.5, 5]$ and construct bins of length 0.5. This leads to 10 bins. Let $b(m)$ the bin to which motion $m$ is mapped to and $|b(m)|$ the cardinality of the bin. Then,

$$\forall m \in \mathcal{D}_{\text{train}}, \quad \text{priority}(m) = \frac{1}{|b(m)|}.$$

We train all the agents for 3M gradient steps corresponding to 30M environment steps. The only exception is PHC where we had to change the update/step ratio and run 300M steps to achieve 3M gradient steps (we also updated the priorities every 10M steps instead of 1M).

**Offline training.** Offline algorithms (i.e., Diffuser and H-GAP) require a dataset label with actions and sufficiently diverse. We thus decided to use a combination of the in-house generated AMASS-Act and the replay buffer of a trained FB-CPR agent. We selected the same motions in $\mathcal{M}$ from the AMASS-Act dataset. The FB-CPR replay buffer corresponds to the buffer of the agent after being trained for 30M environment steps. The resulting dataset contains about 8.1M transitions.

## C.5 Algorithms Implementation and Parameters

In this section, we describe how each considered algorithm was implemented and the hyperparameters used to obtain the results of Tab. 1.

---

[12]We use both velocity and position information for the initialization.

### C.5.1 SHARED CONFIGURATIONS

We first report some configurations shared across multiple algorithms, unless otherwise stated in each section below.

**General training parameters.** We use a replay buffer of capacity 5M transitions and update agents by sampling mini-batches of 1024 transitions. Algorithms that need trajectories from the unlabeled dataset sample segments of these of length 8 steps. During online training, we interleave a rollout phase, where we collect 500 transitions across 50 parallel environments, with a model update phase, where we update each network 50 times. During rollouts of latent- or goal-conditioned agents, we store into the online buffer transitions $(s, a, s', z)$, where $z$ is the latent parameter of the policy that generated the corresponding trajectory. To make off-policy training of all networks (except for discriminators) more efficient, we sample mini-batches containing $(s, a, s', z)$ from the online buffer but relabel each $z$ with a randomly-generated one from the corresponding distribution $\nu$ with some "relabeling probability" (reported in the tables below).

All algorithms keep the running mean and standard deviation of states in batches sampled from the online buffer and the unlabeled dataset at each update. These are used to normalize states before feeding them into each network. Unless otherwise stated we use the Adam optimizer (Kingma & Ba, 2015) with $(\beta_1, \beta_2) = (0.9, 0.999)$ and $\epsilon = 10^{-8}$.

Table 3: Summary of general traning parameters.

| Hyperparameter | Value |
|---|---|
| Number of environment steps | 30M |
| Number of parallel environments | 50 |
| Number of rollout steps between each agent update | 500 |
| Number of gradient steps per agent update | 50 |
| Number of initial steps with random actions | 50000 |
| Replay buffer size | 5M |
| Batch size | 1024 |
| Discount factor | 0.98 |

We report also the parameters used for motion prioritization.

Table 4: Summary of prioritization parameters.

| Hyperparameter | Value |
|---|---|
| Update priorities every N environment steps | 1M |
| EMD clip | [0.5, 5] |
| Bin width | 0.5 |

**Network architectures.** All networks are MLPs with ReLU activations, except for the first hidden layer which uses a layernorm followed by tanh. Each z-conditioned network has two initial "embedding layers", one processing $(s, z)$, and the other processing $s$ alone (or $s$ and $a$). The second embedding layer has half the hidden units of the first layer, and their outputs are concatenated and fed into the main MLP. On the other hand, networks that do not depend on $z$ directly concatenate all inputs and feed them into a simple MLP. The shared parameters used for these two architectures are reported in the table below. Each actor network outputs the mean of a Gaussian distribution with fixed standard deviation of 0.2.

Table 5: Hyperparameters used for the "simple MLP" architectures.

| Hyperparameter | critics | actors | state embeddings |
|---|---|---|---|
| Input variables | $(s, a)$ | $s$ | $s$ |
| Hidden layers | 4 | 4 | 1 |
| Hidden units | 1024 | 1024 | 256 |
| Activations | ReLU | ReLU | ReLU |
| First-layer activation | layernorm + tanh | layernorm + tanh | layernorm + tanh |
| Output activation | linear | tanh | l2-normalization |
| Number of parallel networks | 2 | 1 | 1 |

Table 6: Hyperparameters used for the architectures with embedding layers.

| Hyperparameter | critics (e.g., $F$, $Q$) | actors |
|---|---|---|
| Input variables | $(s, a, z)$ | $(s, z)$ |
| Embeddings | one over $(s, a)$ and one over $(s, z)$ | one over $(s)$ and one over $(s, z)$ |
| Embedding hidden layers | 2 | 2 |
| Embedding hidden units | 1024 | 1024 |
| Embedding output dim | 512 | 512 |
| Hidden layers | 2 | 2 |
| Hidden units | 1024 | 1024 |
| Activations | ReLU | ReLU |
| First-layer activation | layernorm + tanh | layernorm + tanh |
| Output activation | linear | tanh |
| Number of parallel networks | 2 | 1 |

**Discriminator.** The discriminator is an MLP with 3 hidden layers of 1024 hidden units, each with ReLU activations except for the first hidden layer which uses a layernorm followed by tanh. It takes as input a state observation $s$ and a latent variable $z$, and has a sigmoidal unit at the output. It is trained by minimizing the standard cross-entropy loss with a learning rate of $10^{-5}$ regularized by the gradient penalty used in Wasserstein GANs (Gulrajani et al., 2017) with coefficient 10. Note that this is a different gradient penalty than the one used by Peng et al. (2022); Tessler et al. (2023). We provide an in depth ablation into the choice of gradient penalty in App. D.2.

Table 7: Hyperparameters used for the discriminator.

| Hyperparameter | FB-CPR | CALM | ASE | Goal-GAIL |
|---|---|---|---|---|
| Input variables | $(s, z)$ | $(s, z)$ | $s$ | $(s, g)$ |
| Hidden layers | 3 | 3 | 3 | 3 |
| Hidden units | 1024 | 1024 | 1024 | 1024 |
| Activations | ReLU | ReLU | ReLU | ReLU |
| Output activation | sigmoid | sigmoid | sigmoid | sigmoid |
| WGAN gradient penalty coefficient | 10 | 10 | 10 | 10 |
| Learning rate | $10^{-5}$ | $10^{-5}$ | $10^{-5}$ | $10^{-5}$ |

### C.5.2 TD3

We follow the original implementation of algorithm by Fujimoto et al. (2018), except that we replace the minimum operator over target networks to compute the TD targets and the actor loss by a penalization wrt the absolute difference between the Q functions in the ensemble, as proposed by Cetin et al. (2024a). This penalty is used in the actor and the critic of all TD3-based algorithms, with the coefficients reported in the tables below. Note that we will report only the values 0, for which the target is the average of the Q networks in the ensemble, and 0.5, for which the target is the minimum of these networks.

Table 8: Hyperparameters used for TD3 training.

| Hyperparameter | Value |
|---|---|
| General training parameters | See Tab. 3 |
| General prioritization parameters | See Tab. 4 |
| actor network | third column of Tab. 5, output dim = action dim |
| critic network | second column of Tab. 5, output dim 1 |
| Learning rate for actor | $10^{-4}$ |
| Learning rate for critic | $10^{-4}$ |
| Polyak coefficient for target network update | 0.005 |
| Actor penalty coefficient | 0 |
| Critic penalty coefficient | 0 |

### C.5.3 FB-CPR

The algorithm is implemented following the pseudocode App. B. The values of its hyperparameters are reported in the table below.

**Inference methods.** For reward-based inference, we use a weighted regression method $z_r \propto \mathbb{E}_{s' \sim \mathcal{D}_{\text{online}}}[\exp(10r(s'))B(s')r(s')]$, where we estimate the expectation with 100k samples from the online buffer. We found this to work better than standard regression, likely due to the high

diversity of behaviors present in the data. For goal-based inference, we use the original method $z_g = B(g)$, while for motion tracking of a motion $\tau$ we infer one $z$ for each time step t in the motion as $z_t \propto \sum_{j=t+1}^{t+L+1} B(s_j)$, where $s_j$ is the $j$-th state in the motion and $L$ is the same encoding sequence length used during pre-training.

Table 9: Hyperparameters used for FB-CPR pretraining.

| Hyperparameter | Value |
|---|---|
| General training parameters | See Tab. 3 |
| General prioritization parameters | See Tab. 4 |
| Sequence length for trajectory sampling from $\mathcal{D}$ | 8 |
| $z$ update frequency during rollouts | once every 150 steps |
| $z$ dimension $d$ | 256 |
| Regularization coefficient $\alpha$ | 0.01 |
| $F$ network | second column of Tab. 6, output dim 256 |
| actor network | third column of Tab. 6, output dim = action dim |
| critic network | second column of Tab. 6, output dim 1 |
| $B$ network | fourth column of Tab. 5, output dim 256 |
| Discriminator | Tab. 7 |
| Learning rate for F | $10^{-4}$ |
| Learning rate for actor | $10^{-4}$ |
| Learning rate for critic | $10^{-4}$ |
| Learning rate for B | $10^{-5}$ |
| Coefficient for orthonormality loss | 100 |
| $z$ distribution $\nu$ | |
|    -encoding of unlabeled trajectories | 60% |
|    -goals from the online buffer | 20% |
|    -uniform on unit sphere | 20% |
| Probability of relabeling zs | 0.8 |
| Polyak coefficient for target network update | 0.005 |
| FB penalty coefficient | 0 |
| Actor penalty coefficient | 0.5 |
| Critic penalty coefficient | 0.5 |
| Coefficient for Fz-regularization loss | 0.1 |

### C.5.4 ASE

We implemented an off-policy version of ASE to be consistent with the training protocol of FB-CPR. In particular, we use a TD3-based scheme to optimize all networks instead of PPO as in the original implementation of Peng et al. (2022). As for FB-CPR, we fit a critic to predict the expected discounted sum of rewards from the discriminator by temporal difference (see Eq. 10), and another critic to predict $\mathbb{E}[\sum_{t=0}^{\infty} \gamma^t \phi(s_{t+1})^\top z | s, a, \pi_z]$, where $\phi$ is the representation learned by the DIAYN-based (Eysenbach et al., 2019) skill discovery part of the algorithm. We train such representation by an off-policy version of Eq. 13 in (Peng et al., 2022), where we sample couples $(s', z)$ from the online buffer and maximize $\mathbb{E}_{(s',z) \sim \mathcal{D}_{\text{online}}}[\phi(s')^T z]$. Note that this is consistent with the original off-policy implementation of DIAYN (Eysenbach et al., 2019). The output of $\phi$ is normalized on the hypersphere of radius $\sqrt{d}$. We also add an othornormality loss (same as the one used by FB) as we found this to be essential for preventing collapse of the encoder.

**Inference methods**. For reward-based and goal-based inference we use the same methods as FB-CPR, with B replaced with $\phi$. For tracking we use $z_t \propto B(s_{t+1})$ for each timestep $t$ in the target motion.

Table 10: Hyperparameters used for ASE pretraining.

| Hyperparameter | Value |
| --- | --- |
| General training parameters | See Tab. 3 |
| General prioritization parameters | See Tab. 4 |
| $z$ update frequency during rollouts | once every 150 steps |
| $z$ dimension $d$ | 64 |
| Regularization coefficient $\alpha$ | 0.01 |
| actor network | third column of Tab. 6, output dim = action dim |
| critic networks | second column of Tab. 6, output dim 1 |
| $\phi$ encoder network | fourth column of Tab. 5, output dim 64 |
| Discriminator | Tab. 7 |
| Learning rate for actor | $10^{-4}$ |
| Learning rate for critic | $10^{-4}$ |
| Learning rate for $\phi$ | $10^{-8}$ |
| Coefficient for orthonormality loss | 100 |
| $z$ distribution $\nu$ | |
|     -goals from unlabeled dataset | 60% |
|     -goals from the online buffer | 20% |
|     -uniform on unit sphere | 20% |
| Probability of relabeling zs | 0.8 |
| Polyak coefficient for target network update | 0.005 |
| Coefficient for diversity loss (Eq. 15 in (Peng et al., 2022)) | 0 |
| Actor penalty coefficient | 0.5 |
| Critic penalty coefficient | 0.5 |

### C.5.5 CALM

As for ASE, we implemented an off-policy TD3-based version of CALM to be consistent with the training protocol of FB-CPR. We fit a critic $Q(s, a, z)$ to predict the expected discounted sum of rewards from the discriminator by temporal difference (see Eq. 10). We also train a sequence encoder $\phi(\tau)$ which embeds a sub-trajectory $\tau$ from the unlabeled dataset into $z$ space through a transformer. The encoder and the actor are trained end-to-end by maximizing $Q(s, \pi(s, z = \phi(\tau)), z = \phi(\tau))$, plus the constrastive regularization loss designed to prevent the encoder from collapsing (Eq. 5,6 in (Tessler et al., 2023)). The transformer interleaves attention and feed-forward blocks. The former uses a layernorm followed by multi-head self-attention plus a residual connection, while the latter uses a layernorm followed by two linear layers interleaved by a GELU activation. Its output is normalized on the hypersphere of radius $\sqrt{d}$.

**Inference methods.** We use the same methods as FB-CPR for goal-based and tracking inference.

Table 11: Hyperparameters used for CALM pretraining.

| Hyperparameter | Value |
| --- | --- |
| General training parameters | See Tab. 3 |
| General prioritization parameters | See Tab. 4 |
| Sequence length for trajectory sampling from $\mathcal{D}$ | 8 |
| $z$ update frequency during rollouts | once every 150 steps |
| $z$ dimension $d$ | 256 |
| actor network | third column of Tab. 6, output dim = action dim |
| critic network | second column of Tab. 6, output dim 1 |
| $\phi$ encoder network | transformer (see text above) |
|     -attention blocks | 2 |
|     -embedding dim | 256 |
|     -MLP first linear layer | 256x1024 |
|     -MLP second linear layer | 1024x256 |
| Discriminator | Tab. 7 |
| Learning rate for actor | $10^{-4}$ |
| Learning rate for critic | $10^{-4}$ |
| Learning rate for $\phi$ | $10^{-7}$ |
| Coefficient for constrastive loss | 0.1 |
| $z$ distribution $\nu$ | |
|     -encoding of unlabeled trajectories | 100% |
|     -goals from the online buffer | 0% |
|     -uniform on unit sphere | 0% |
| Probability of relabeling zs | 1 |
| Polyak coefficient for target network update | 0.005 |
| Actor penalty coefficient | 0.5 |
| Critic penalty coefficient | 0.5 |

### C.5.6 PHC

PHC is similar to a goal-conditioned algorithm except that the goal is "forced" to be the next state in the motion. This makes PHC an algorithm specifically designed for one-step tracking. We use a TD3-based variant of the original implementation (Luo et al., 2023). Concretely the implementation is exactly the same of TD3 but we changed the underlying environment. In this tracking environment the state is defined as the concatenation of the current state $s$ and the state $g$ to track. The resulting state space is $\mathbb{R}^{716}$. At the beginning of an episode, we sample a motion $m$ from the motion set (either $\mathcal{M}$ or $\mathcal{D}_{\text{test}}$) and we initialize the agent to a randomly selected state of the motion. Let $\bar{t}$ being the randomly selected initial step of the motion, then at any episode step $t \in [1, \text{len}(m) - \bar{t} - 1]$ the target state $g_t$ correspond to the motion state $m_{\bar{t}+t+1}$. We use the negative distance in position/orientation as reward function, i.e., $r((s,g), a, (s', g')) = -d_{\text{smpl}}(g, s')$.

**Inference methods.** By being a goal-conditioned algorithm we just need to pass the desired goal as target reference and can be evaluated for goal and tracking tasks.

Table 12: Hyperparameters used for PHC pretraining.

| Hyperparameter | Value |
|---|---|
| General training parameters | See Tab. 3 |
| General prioritization parameters | See Tab. 4 |
| Update priorities every N environment steps | 10M |
| Number of environment steps | $300M$ |
| Number of gradient steps per agent update | 5 |
| TD3 configuration | See Tab. 8 |

### C.5.7 GOAL-GAIL

We use a TD3-based variant of the original implementation (Ding et al., 2019). Concretely, the implementation is very similar to the one of CALM, except that there is no trajectory encoder and the discriminator directly receives couples $(s, g)$, where $g$ is a goal state sampled from the online buffer or the unlabeled dataset. In particular, the negative pairs $(s, g)$ for updating the discriminator are sampled uniformly from the online buffer (where $g$ is the goal that was targeted when rolling out the policy that generated $s$), while the positive pairs are obtained by sampling a sub-trajectory $\tau$ of length 8 from the unlabeled dataset and taking $g$ as the last state and $s$ as another random state. Similarly to CALM, we train a goal-conditioned critic $Q(s, a, g)$ to predict the expected discounted sum of discriminator rewards, and an goal-conditioned actor $\pi(s, g)$ to maximize the predictions of such a critic.

**Inference methods.** We use the same methods as ASE for goal-based and tracking inference.

Table 13: Hyperparameters used for GOAL-GAIL pretraining.

| Hyperparameter | Value |
|---|---|
| General training parameters | See Tab. 3 |
| General prioritization parameters | See Tab. 4 |
| Sequence length for trajectory sampling from $\mathcal{D}$ | 8 |
| goal update frequency during rollouts | once every 150 steps |
| actor network | third column of Tab. 6, output dim = action dim |
| critic network | second column of Tab. 6, output dim 1 |
| Discriminator | Tab. 7 |
| Learning rate for actor | $10^{-4}$ |
| Learning rate for critic | $10^{-4}$ |
| goal sampling distribution | |
|    -goals from the unlabeled dataset | 50% |
|    -goals from the online buffer | 50% |
| Probability of relabeling zs | 0.8 |
| Polyak coefficient for target network update | 0.005 |
| Actor penalty coefficient | 0.5 |
| Critic penalty coefficient | 0.5 |

### C.5.8 GOAL-TD3

We closely follow the implementation of Pirotta et al. (2024). For reaching each goal $g$, we use the reward function $r(s', g) = -\|\text{pos}(s') - \text{pos}(g)\|_2$, where $\text{pos}(\cdot)$ extracts only the position

of each joint, ignoring their velocities. We then train a goal-conditioned TD3 agent to optimize such a reward for all $g$. We sample a percentage of training goals from the unlabeled dataset, and a percentage using hindsight experience replay (HER, Andrychowicz et al., 2017) on trajectories from the online buffer.

**Inference methods.** We use the same methods as ASE for goal-based and tracking inference.

Table 14: Hyperparameters used for GOAL-TD3 pretraining.

| Hyperparameter | Value |
|---|---|
| General training parameters | See Tab. 3 |
| General prioritization parameters | See Tab. 4 |
| Sequence length for HER sampling | 8 |
| goal update frequency during rollouts | once every 150 steps |
| actor network | third column of Tab. 6, output dim = action dim |
| critic network | second column of Tab. 6, output dim 1 |
| Learning rate for actor | $10^{-4}$ |
| Learning rate for critic | $10^{-4}$ |
| goal sampling distribution | |
|     -goals from the unlabeled dataset | 100% |
|     -goals from the online buffer (HER) | 0% |
| Probability of relabeling zs | 0.5 |
| Polyak coefficient for target network update | 0.005 |
| Actor penalty coefficient | 0.5 |
| Critic penalty coefficient | 0.5 |

### C.5.9 MPPI

We use MPPI with the real dynamic and real reward function for each task. For each evaluation state, action plans are sampled according to a factorized Gaussian distribution. Initially, mean and standard variation of the Gaussian are set with 0 and 1, respectively. actions plans are evaluated by deploying them in the real dynamics and computed the cumulative return over some planning horizon. Subsequently, the Gaussian parameters are updated using the top-$k$ most rewarding plans. For goal-reaching tasks, we use the reward $r(s', g) = -\|\mathrm{pos}(s') - \mathrm{pos}(g)\|_2$

Table 15: Hyperparameters used for MPPI planning.

| Hyperparameter | Value |
|---|---|
| Number of plans | 256 |
| Planning horizon | 32 for reward-based tasks, 8 for goals |
| $k$ for the top-$k$ | 64 |
| Maximum of standard deviation | 2 |
| Minimum of standard deviation | 0.2 |
| Temperature | 1 |
| Number of optimization steps | 10 |

### C.5.10 DIFFUSER

We train Diffuser offline on FB-CPR replay buffer and AMASS-Act dataset as described in C.4. We follow the original implementation in Janner et al. (2022). We use diffusion probabilistic model to learn a generative model over sequence of state-action pairs. Diffusion employs a forward diffusion process $q(\tau^i|\tau^{i-1})$ (typically pre-specified) to slowly corrupt the data by adding noise and learn a parametric reverse denoising process $p_\theta(\tau^{i-1}|\tau^i), \forall i \in [0, n]$ which induces the following data distribution:

$$p_\theta(\tau^0) = \int p(\tau^n) \prod_{i=1}^{n} p_\theta(\tau^{i-1} \mid \tau^i) \mathrm{d}\tau^1 \dots \mathrm{d}\tau^n \qquad (12)$$

where $\tau^0$ denotes the real data and $\tau^n$ is sampled from a standard Gaussian prior. The parametric models is trained using a variationnal bound on the log-likelihood objective $\mathbb{E}_{\tau^0 \sim \mathcal{D}}[\log p_\theta(\tau^0)]$. We use Temporal U-net architecture as in Janner et al. (2022) for $p_\theta$.

At test time, we learn a value function to predict the cumulative sum of reward given a sequence $\tau$: $R_\psi(\tau) \approx \sum_{t=1}^{l(\tau)} \gamma^{t-1} r(s_t)$. To do that, we relabel the offline dataset according to the task's reward

and we train $R_\psi$ by regression on the same noise distribution used in the diffusion training:

$$\mathbb{E}_{\tau^0 \sim \mathcal{D}} \mathbb{E}_{i \in \mathcal{U}[n]} \mathbb{E}_{\tau^i \sim q(\tau^i | \tau^0)} \left[ \left( R_\psi(\tau^i) - \sum_{t=1}^{l(\tau^0)} \gamma^{t-1} r(s_t) \right)^2 \right] \tag{13}$$

We use then guiding sampling to solve the task by following the gradient of the value function $\nabla_{\tau^i} R_\psi(\tau^i)$ at each denoising step. For goal-reaching tasks, we condition the diffuser sampling by replacing the last state of the sampled sequence $\tau^i$ by the goal state after each diffusion steps. We sample several sequences and we select the one that maximizes the cumulative sum of the reward $r(s', g) = -\|\text{pos}(s') - \text{pos}(g)\|_2$.

Table 16: Hyperparameters used for Diffuser pretraining and planning.

| Hyperparameter | Value |
|---|---|
| Learning rate | $4 \times 10^{-5}$ |
| Number of gradient steps | $3 \times 10^6$ |
| Sequence length | 32 |
| U-Net hidden dimension | 1024 |
| Number of diffusion steps | 50 |
| Weight of the action loss | 10 |
| Planning horizon | 32 |
| Gradient scale | 0.1 |
| Number of plans | 128 |
| Number of guided steps | 2 |
| Number of guided-free denoising steps | 4 |

### C.5.11 H-GAP

We train the H-GAP model on the FB-CPR replay buffer and the AMASS-Act dataset as outlined in C.4. Following the methodology described in Jiang et al. (2024), we first train a VQ-VAE on the dataset to discretize the state-action trajectories. Subsequently, we train a decoder-only Prior Transformer to model the latent codes autoregressively. In line with the procedures detailed in Jiang et al. (2024), we integrate H-GAP within a Model Predictive Control (MPC) framework. This integration involves employing top-p sampling to generate a set of probable latent trajectories, which were then decoded back into the original state-action space. At test time, we selected the most optimal trajectory based on the task-specific reward functions, assuming access to these functions.

Table 17: Hyperparameters used for H-GAP.

| Hyperparameter | Value |
|---|---|
| batch size | 128 |
| training steps | $10^8$ |
| Modeling horizon | 32 |
| VQ-VAE chunk size | 4 |
| VQ-VAE code per chunk | 32 |
| VQ-VAE number of code | 512 |
| VQ-VAE learning rate | $3 \times 10^{-4}$ |
| VQ-VAE number of heads | 4 |
| VQ-VAE number of layers | 4 |
| Prior Transformer number of heads | 10 |
| Prior Transformer number of layers | 10 |
| Prior Transformer learning rate | $3 \times 10^{-4}$ |

| Task | TD3 | MPPI | Norm. | Diffuser | Normalized | ASE | Normalized | FB-CPR | Normalized |
|---|---|---|---|---|---|---|---|---|---|
| move-ego-0-0 | 275.08 | 203.33 | 0.74 | 227.27 (3.09) | 0.83 (0.01) | 266.03 (1.41) | 0.97 (0.01) | 274.68 (1.48) | 1.00 (0.01) |
| move-ego-low-0-0 | 273.67 | 249.12 | 0.91 | 118.50 (15.56) | 0.43 (0.06) | 222.14 (19.48) | 0.81 (0.07) | 215.61 (27.63) | 0.79 (0.10) |
| handstand | 251.30 | 3.58 | 0.01 | 5.21 (3.76) | 0.02 (0.01) | 0.04 (0.08) | 0.00 (0.00) | 41.27 (10.20) | 0.16 (0.04) |
| move-ego-0-2 | 255.57 | 263.67 | 1.03 | 238.99 (5.79) | 0.94 (0.02) | 224.29 (50.58) | 0.88 (0.20) | 260.93 (5.21) | 1.02 (0.02) |
| move-ego-0-4 | 242.66 | 251.13 | 1.03 | 179.82 (19.33) | 0.74 (0.04) | 211.65 (32.39) | 0.87 (0.13) | 235.44 (29.42) | 0.97 (0.12) |
| move-ego--90-2 | 255.45 | 260.71 | 1.02 | 206.48 (7.00) | 0.81 (0.03) | 230.46 (9.72) | 0.90 (0.04) | 210.99 (6.55) | 0.83 (0.03) |
| move-ego--90-4 | 245.76 | 250.29 | 1.02 | 137.80 (9.33) | 0.56 (0.04) | 143.12 (26.14) | 0.58 (0.11) | 202.99 (9.33) | 0.83 (0.04) |
| move-ego-90-2 | 253.77 | 262.62 | 1.02 | 207.27 (4.74) | 0.82 (0.02) | 194.18 (64.48) | 0.77 (0.25) | 224.68 (9.15) | 0.89 (0.04) |
| move-ego-90-4 | 247.49 | 251.61 | 1.02 | 132.93 (10.93) | 0.54 (0.04) | 134.14 (12.22) | 0.54 (0.05) | 185.60 (14.42) | 0.75 (0.06) |
| move-ego-180-2 | 258.28 | 251.46 | 0.97 | 195.45 (7.26) | 0.76 (0.03) | 237.73 (21.51) | 0.92 (0.08) | 227.34 (27.01) | 0.88 (0.10) |
| move-ego-180-4 | 249.81 | 252.28 | 1.01 | 132.89 (9.70) | 0.53 (0.04) | 134.54 (13.34) | 0.54 (0.05) | 205.54 (14.40) | 0.82 (0.06) |
| move-ego-low-0-2 | 274.71 | 273.65 | 1.00 | 100.64 (8.61) | 0.37 (0.03) | 56.46 (10.91) | 0.21 (0.04) | 207.27 (58.01) | 0.75 (0.21) |
| move-ego-low--90-2 | 270.69 | 266.74 | 0.99 | 80.33 (4.51) | 0.30 (0.02) | 65.01 (44.17) | 0.24 (0.16) | 221.37 (35.35) | 0.82 (0.13) |
| move-ego-low-90-2 | 259.97 | 267.52 | 1.03 | 96.12 (6.79) | 0.37 (0.03) | 58.71 (47.10) | 0.23 (0.18) | 222.81 (21.94) | 0.86 (0.08) |
| move-ego-low-180-2 | 280.15 | 273.37 | 0.98 | 65.61 (7.73) | 0.23 (0.03) | 13.77 (16.25) | 0.05 (0.06) | 65.20 (32.64) | 0.23 (0.12) |
| jump-2 | 90.66 | 67.45 | 0.74 | 15.85 (0.64) | 0.17 (0.01) | 8.73 (6.86) | 0.10 (0.08) | 34.88 (3.52) | 0.38 (0.04) |
| rotate-x--5-0.8 | 222.60 | 163.35 | 0.73 | 8.31 (1.82) | 0.04 (0.01) | 0.04 (0.05) | 0.00 (0.00) | 7.42 (5.69) | 0.03 (0.03) |
| rotate-x-5-0.8 | 219.28 | 176.23 | 0.80 | 13.04 (3.12) | 0.06 (0.01) | 0.04 (0.01) | 0.00 (0.00) | 2.29 (1.78) | 0.01 (0.01) |
| rotate-y--5-0.8 | 272.15 | 270.84 | 1.00 | 107.14 (14.51) | 0.39 (0.05) | 124.52 (32.52) | 0.46 (0.12) | 217.70 (43.67) | 0.80 (0.16) |
| rotate-y-5-0.8 | 273.74 | 272.66 | 1.00 | 97.70 (10.05) | 0.36 (0.04) | 149.48 (36.92) | 0.55 (0.13) | 199.08 (51.78) | 0.73 (0.19) |
| rotate-z--5-0.8 | 257.30 | 208.39 | 0.81 | 6.67 (1.50) | 0.03 (0.01) | 0.39 (0.77) | 0.00 (0.00) | 95.23 (15.75) | 0.37 (0.06) |
| rotate-z-5-0.8 | 266.16 | 206.59 | 0.78 | 5.83 (2.46) | 0.02 (0.01) | 0.01 (0.00) | 0.00 (0.00) | 124.95 (17.61) | 0.47 (0.07) |
| raisearms-l-l | 264.61 | 194.60 | 0.74 | 221.11 (5.14) | 0.84 (0.02) | 265.15 (1.35) | 1.00 (0.01) | 270.43 (0.37) | 1.02 (0.00) |
| raisearms-l-m | 266.03 | 187.43 | 0.70 | 133.55 (8.85) | 0.50 (0.03) | 63.67 (18.97) | 0.24 (0.07) | 97.66 (81.17) | 0.37 (0.31) |
| raisearms-l-h | 268.30 | 41.05 | 0.15 | 87.44 (13.21) | 0.33 (0.05) | 258.00 (1.36) | 0.96 (0.01) | 243.16 (19.18) | 0.91 (0.07) |
| raisearms-m-l | 269.36 | 178.85 | 0.66 | 116.25 (13.75) | 0.43 (0.05) | 70.66 (36.32) | 0.26 (0.13) | 134.83 (70.28) | 0.50 (0.26) |
| raisearms-m-m | 267.55 | 137.62 | 0.51 | 139.84 (12.04) | 0.52 (0.04) | 11.52 (0.14) | 0.04 (0.00) | 87.25 (98.42) | 0.33 (0.37) |
| raisearms-m-h | 264.12 | 34.64 | 0.13 | 91.54 (8.02) | 0.35 (0.03) | 52.79 (1.61) | 0.20 (0.01) | 75.05 (69.32) | 0.28 (0.26) |
| raisearms-h-l | 273.91 | 40.19 | 0.15 | 62.35 (9.37) | 0.23 (0.03) | 240.23 (22.36) | 0.88 (0.08) | 167.98 (82.03) | 0.61 (0.30) |
| raisearms-h-m | 264.67 | 36.41 | 0.14 | 78.29 (16.38) | 0.30 (0.06) | 54.58 (3.27) | 0.21 (0.01) | 104.26 (81.69) | 0.39 (0.31) |
| raisearms-h-h | 265.17 | 8.23 | 0.03 | 69.31 (19.10) | 0.26 (0.07) | 255.83 (0.69) | 0.96 (0.00) | 199.88 (42.03) | 0.75 (0.16) |
| crouch-0 | 268.83 | 222.66 | 0.83 | 82.36 (12.78) | 0.31 (0.05) | 181.96 (58.21) | 0.68 (0.22) | 226.28 (28.17) | 0.84 (0.10) |
| sitonground | 271.76 | 243.64 | 0.90 | 61.18 (9.02) | 0.23 (0.03) | 114.03 (57.40) | 0.42 (0.21) | 199.44 (22.15) | 0.73 (0.08) |
| lieonground-up | 278.66 | 249.31 | 0.89 | 29.05 (7.71) | 0.10 (0.03) | 204.26 (18.93) | 0.73 (0.07) | 193.66 (33.18) | 0.69 (0.12) |
| lieonground-down | 277.51 | 242.08 | 0.87 | 73.70 (10.52) | 0.27 (0.04) | 158.10 (68.06) | 0.57 (0.25) | 193.50 (18.89) | 0.70 (0.07) |
| split-0.5 | 276.13 | 250.66 | 0.91 | 104.29 (12.85) | 0.38 (0.05) | 112.46 (71.92) | 0.41 (0.26) | 232.18 (20.26) | 0.84 (0.07) |
| split-1 | 279.25 | 253.28 | 0.91 | 27.28 (5.74) | 0.10 (0.02) | 13.92 (20.72) | 0.05 (0.07) | 117.67 (61.27) | 0.42 (0.22) |
| crawl-0.4-0-u | 145.11 | 124.76 | 0.86 | 10.47 (6.81) | 0.07 (0.05) | 77.46 (36.91) | 0.53 (0.25) | 101.76 (15.97) | 0.70 (0.11) |
| crawl-0.4-2-u | 287.01 | 60.50 | 0.21 | 1.81 (1.25) | 0.01 (0.01) | 4.03 (4.03) | 0.01 (0.01) | 15.02 (6.03) | 0.05 (0.02) |
| crawl-0.5-0-u | 146.02 | 124.75 | 0.85 | 4.84 (3.67) | 0.03 (0.03) | 77.72 (37.07) | 0.53 (0.25) | 101.92 (16.39) | 0.70 (0.11) |
| crawl-0.5-2-u | 234.51 | 60.16 | 0.26 | 1.77 (1.27) | 0.01 (0.01) | 3.97 (4.04) | 0.02 (0.02) | 15.81 (6.10) | 0.07 (0.03) |
| crawl-0.4-0-d | 145.79 | 112.27 | 0.77 | 27.44 (9.15) | 0.19 (0.06) | 20.32 (14.02) | 0.14 (0.10) | 191.75 (43.60) | 1.32 (0.30) |
| crawl-0.4-2-d | 289.55 | 105.70 | 0.37 | 4.00 (0.78) | 0.01 (0.00) | 15.50 (3.19) | 0.05 (0.01) | 19.00 (4.07) | 0.07 (0.01) |
| crawl-0.5-0-d | 146.46 | 112.00 | 0.76 | 24.68 (3.74) | 0.17 (0.03) | 7.03 (2.07) | 0.05 (0.01) | 131.13 (64.97) | 0.90 (0.44) |
| crawl-0.5-2-d | 291.74 | 64.94 | 0.22 | 4.64 (2.01) | 0.02 (0.01) | 19.41 (9.51) | 0.07 (0.03) | 22.93 (5.31) | 0.08 (0.02) |
| Average | 249.74 | 178.50 | 0.72 | 85.27 | 0.33 | 105.73 | 0.41 | 151.68 | 0.61 |
| Median | 265.17 | 206.59 | 0.83 | 80.33 | 0.30 | 77.46 | 0.41 | 191.75 | 0.73 |

Table 18: Humanoid Environment. Average return per task for reward-optimization evaluation.

| Group | Num. Tasks | TD3 | MPPI | Normalized | Diffuser | Normalized | ASE | Normalized | FB-CPR | Normalized |
|---|---|---|---|---|---|---|---|---|---|---|
| Stand | 2 | 274.38 (0.71) | 226.22 (22.89) | 0.82 (0.09) | 172.89 (54.38) | 0.63 (0.20) | 244.09 (21.94) | 0.89 (0.08) | 245.14 (29.53) | 0.89 (0.11) |
| Handstand | 1 | 251.30 (0.00) | 3.58 (0.00) | 0.01 (0.00) | 5.21 (0.00) | 0.02 (0.00) | 0.04 (0.00) | 0.00 (0.00) | 41.27 (0.00) | 0.16 (0.00) |
| Locomotion | 8 | 251.10 (5.15) | 255.47 (5.39) | 1.02 (0.02) | 178.95 (37.70) | 0.71 (0.14) | 188.76 (41.77) | 0.75 (0.16) | 219.19 (21.64) | 0.87 (0.08) |
| Locom.-Low | 4 | 271.38 (7.39) | 270.32 (3.20) | 1.00 (0.02) | 85.67 (13.83) | 0.32 (0.06) | 48.49 (20.28) | 0.18 (0.08) | 179.16 (66.08) | 0.67 (0.25) |
| Jump | 1 | 90.66 (0.00) | 67.45 (0.00) | 0.74 (0.00) | 15.85 (0.00) | 0.17 (0.00) | 8.73 (0.00) | 0.10 (0.00) | 34.88 (0.00) | 0.38 (0.00) |
| Rotation | 6 | 251.87 (22.52) | 216.34 (42.26) | 0.85 (0.16) | 39.78 (44.43) | 0.15 (0.16) | 45.75 (64.93) | 0.17 (0.24) | 107.78 (83.74) | 0.40 (0.31) |
| RaiseArms | 9 | 267.08 (2.96) | 95.45 (72.90) | 0.36 (0.27) | 111.08 (46.67) | 0.42 (0.18) | 141.38 (102.78) | 0.53 (0.38) | 153.39 (67.09) | 0.57 (0.25) |
| On-Ground | 6 | 275.36 (3.80) | 243.61 (10.14) | 0.88 (0.03) | 62.98 (27.77) | 0.23 (0.10) | 130.79 (61.96) | 0.48 (0.23) | 193.79 (37.32) | 0.71 (0.14) |
| Crawl | 8 | 210.77 (67.08) | 95.63 (26.87) | 0.54 (0.28) | 9.96 (9.66) | 0.06 (0.07) | 28.18 (29.15) | 0.18 (0.21) | 74.91 (62.42) | 0.48 (0.45) |

Table 19: Humanoid Environment. Average return per category for reward-optimization evaluation.

# D    ADDITIONAL EXPERIMENTAL RESULTS

In this section we report a more detailed analysis of the experiments.

## D.1    DETAILED RESULTS

In this section we report detailed results split across tasks.

- Table 18 shows the average return for each reward-based task and Table 19 groups the results per task category.

- Table 20 shows the proximity metric for each goal pose, while Table 21 shows the success rate.

- Table 22 shows the train and test tracking performance for both EMD and success rate grouped over the AMASS datasets.

| Goal | TD3 | MPPI | Diffuser | Goal-GAIL | Goal-TD3 | PHC | CALM | ASE | FB-CPR |
|---|---|---|---|---|---|---|---|---|---|
| Proximity | | | | | | | | | |
| t_pose | 0.99 | 0.21 | 0.60 (0.07) | 0.98 (0.00) | 0.99 (0.00) | 0.24 (0.03) | 0.53 (0.34) | 0.98 (0.01) | 0.99 (0.00) |
| t_pose_lower_arms | 0.99 | 0.28 | 0.52 (0.04) | 0.96 (0.05) | 0.99 (0.00) | 0.44 (0.04) | 0.81 (0.17) | 0.95 (0.06) | 0.99 (0.00) |
| t_pose_bow_head | 0.99 | 0.23 | 0.60 (0.13) | 0.98 (0.00) | 0.99 (0.00) | 0.21 (0.06) | 0.63 (0.27) | 0.82 (0.12) | 0.99 (0.00) |
| u_stretch_y_right | 0.99 | 0.19 | 0.12 (0.12) | 0.79 (0.17) | 0.87 (0.07) | 0.02 (0.01) | 0.16 (0.14) | 0.55 (0.20) | 0.70 (0.21) |
| u_stretch_y_left | 0.98 | 0.20 | 0.01 (0.01) | 0.55 (0.11) | 0.77 (0.06) | 0.02 (0.01) | 0.10 (0.20) | 0.37 (0.23) | 0.73 (0.18) |
| u_stretch_z_right | 0.99 | 0.28 | 0.02 (0.01) | 0.66 (0.28) | 0.81 (0.14) | 0.04 (0.00) | 0.09 (0.14) | 0.31 (0.23) | 0.83 (0.10) |
| u_stretch_z_left | 0.99 | 0.16 | 0.25 (0.09) | 0.95 (0.04) | 0.95 (0.07) | 0.06 (0.01) | 0.09 (0.15) | 0.45 (0.25) | 0.97 (0.03) |
| u_stretch_x_back | 0.98 | 0.07 | 0.10 (0.11) | 0.81 (0.14) | 0.72 (0.17) | 0.02 (0.01) | 0.01 (0.01) | 0.76 (0.22) | 0.93 (0.04) |
| u_stretch_x_front_part | 0.99 | 0.63 | 0.55 (0.04) | 0.94 (0.07) | 0.99 (0.00) | 0.14 (0.02) | 0.34 (0.20) | 0.74 (0.16) | 0.99 (0.00) |
| u_stretch_x_front_full | 0.98 | 0.98 | 0.06 (0.03) | 0.84 (0.09) | 0.90 (0.07) | 0.01 (0.00) | 0.34 (0.29) | 0.60 (0.22) | 0.95 (0.02) |
| crossed_arms | 0.98 | 0.20 | 0.26 (0.10) | 0.80 (0.06) | 0.86 (0.08) | 0.02 (0.01) | 0.14 (0.17) | 0.56 (0.07) | 0.89 (0.05) |
| scratching_head | 0.99 | 0.24 | 0.29 (0.14) | 0.98 (0.00) | 0.99 (0.01) | 0.06 (0.02) | 0.15 (0.25) | 0.97 (0.01) | 0.99 (0.00) |
| right_hand_wave | 0.99 | 0.23 | 0.42 (0.17) | 0.92 (0.01) | 0.98 (0.00) | 0.12 (0.01) | 0.32 (0.20) | 0.94 (0.02) | 0.95 (0.00) |
| x_strech | 0.98 | 0.11 | 0.42 (0.13) | 0.90 (0.08) | 0.93 (0.05) | 0.06 (0.02) | 0.12 (0.14) | 0.82 (0.13) | 0.94 (0.05) |
| i_strech | 0.86 | 0.07 | 0.20 (0.15) | 0.71 (0.07) | 0.74 (0.09) | 0.01 (0.00) | 0.02 (0.03) | 0.69 (0.08) | 0.88 (0.08) |
| arms_stretch | 0.98 | 0.08 | 0.22 (0.13) | 0.58 (0.08) | 0.72 (0.14) | 0.07 (0.01) | 0.05 (0.10) | 0.39 (0.13) | 0.68 (0.06) |
| drinking_from_bottle | 0.98 | 0.23 | 0.17 (0.07) | 0.69 (0.09) | 0.88 (0.08) | 0.04 (0.02) | 0.07 (0.10) | 0.80 (0.08) | 0.97 (0.04) |
| arm_on_chest | 0.98 | 0.15 | 0.17 (0.07) | 0.92 (0.05) | 0.99 (0.00) | 0.04 (0.01) | 0.16 (0.17) | 0.95 (0.02) | 0.98 (0.00) |
| pre_throw | 0.56 | 0.03 | 0.00 (0.00) | 0.08 (0.07) | 0.23 (0.13) | 0.04 (0.01) | 0.00 (0.00) | 0.02 (0.03) | 0.08 (0.10) |
| egyptian | 0.99 | 0.18 | 0.18 (0.08) | 0.80 (0.10) | 0.94 (0.06) | 0.12 (0.03) | 0.28 (0.28) | 0.60 (0.27) | 0.98 (0.00) |
| zombie | 0.98 | 0.14 | 0.47 (0.09) | 0.96 (0.03) | 0.99 (0.00) | 0.15 (0.04) | 0.33 (0.30) | 0.92 (0.05) | 0.98 (0.00) |
| stand_martial_arts | 0.99 | 0.41 | 0.41 (0.17) | 0.94 (0.05) | 0.99 (0.01) | 0.05 (0.03) | 0.34 (0.23) | 0.94 (0.02) | 0.98 (0.00) |
| peekaboo | 0.90 | 0.25 | 0.27 (0.12) | 0.91 (0.10) | 0.75 (0.20) | 0.06 (0.03) | 0.18 (0.23) | 0.87 (0.15) | 0.95 (0.04) |
| dance | 0.98 | 0.17 | 0.31 (0.06) | 0.97 (0.02) | 0.99 (0.00) | 0.07 (0.04) | 0.34 (0.24) | 0.86 (0.16) | 0.99 (0.00) |
| kneel_left | 0.99 | 0.97 | 0.10 (0.07) | 0.79 (0.12) | 0.94 (0.05) | 0.04 (0.00) | 0.23 (0.30) | 0.34 (0.19) | 0.95 (0.02) |
| crouch_high | 0.99 | 0.89 | 0.39 (0.05) | 0.98 (0.00) | 0.99 (0.00) | 0.46 (0.08) | 0.76 (0.18) | 0.85 (0.12) | 0.99 (0.00) |
| crouch_medium | 0.99 | 0.95 | 0.47 (0.06) | 0.99 (0.00) | 1.00 (0.00) | 0.38 (0.07) | 0.81 (0.12) | 0.86 (0.12) | 0.99 (0.00) |
| crouch_low | 0.99 | 0.63 | 0.08 (0.03) | 0.73 (0.02) | 0.85 (0.09) | 0.07 (0.03) | 0.16 (0.15) | 0.47 (0.11) | 0.85 (0.06) |
| squat_pre_jump | 0.98 | 0.97 | 0.03 (0.01) | 0.17 (0.13) | 0.22 (0.07) | 0.02 (0.01) | 0.03 (0.05) | 0.31 (0.20) | 0.56 (0.04) |
| squat_hands_on_ground | 0.98 | 0.77 | 0.21 (0.07) | 0.72 (0.08) | 0.93 (0.04) | 0.02 (0.01) | 0.21 (0.25) | 0.30 (0.19) | 0.74 (0.10) |
| side_high_kick | 0.98 | 0.38 | 0.00 (0.00) | 0.02 (0.02) | 0.02 (0.01) | 0.01 (0.01) | 0.00 (0.00) | 0.01 (0.01) | 0.03 (0.03) |
| pre_front_kick | 0.99 | 0.33 | 0.01 (0.00) | 0.54 (0.22) | 0.75 (0.09) | 0.06 (0.03) | 0.08 (0.06) | 0.20 (0.16) | 0.69 (0.21) |
| arabesque_hold_foot | 0.85 | 0.17 | 0.03 (0.03) | 0.11 (0.06) | 0.30 (0.13) | 0.01 (0.00) | 0.02 (0.04) | 0.02 (0.02) | 0.11 (0.05) |
| hold_right_foot | 0.99 | 0.17 | 0.04 (0.03) | 0.28 (0.11) | 0.56 (0.20) | 0.03 (0.01) | 0.01 (0.03) | 0.10 (0.07) | 0.64 (0.12) |
| hold_left_foot | 0.99 | 0.44 | 0.04 (0.01) | 0.51 (0.09) | 0.76 (0.08) | 0.20 (0.02) | 0.29 (0.10) | 0.17 (0.17) | 0.72 (0.07) |
| bend_on_left_leg | 0.98 | 0.69 | 0.01 (0.00) | 0.09 (0.10) | 0.40 (0.08) | 0.02 (0.01) | 0.04 (0.08) | 0.09 (0.08) | 0.57 (0.12) |
| lie_front | 0.97 | 0.87 | 0.16 (0.16) | 0.67 (0.11) | 0.52 (0.08) | 0.01 (0.00) | 0.05 (0.04) | 0.46 (0.14) | 0.61 (0.10) |
| crawl_backward | 0.98 | 0.92 | 0.13 (0.13) | 0.36 (0.19) | 0.37 (0.15) | 0.00 (0.00) | 0.01 (0.02) | 0.03 (0.04) | 0.13 (0.13) |
| lie_back_knee_bent | 0.97 | 0.79 | 0.07 (0.07) | 0.15 (0.13) | 0.03 (0.03) | 0.02 (0.01) | 0.00 (0.00) | 0.09 (0.14) | 0.04 (0.08) |
| lie_side | 0.97 | 0.89 | 0.20 (0.08) | 0.36 (0.18) | 0.19 (0.11) | 0.02 (0.01) | 0.00 (0.00) | 0.08 (0.08) | 0.36 (0.04) |
| crunch | 0.98 | 0.44 | 0.00 (0.00) | 0.00 (0.00) | 0.04 (0.07) | 0.01 (0.00) | 0.00 (0.00) | 0.00 (0.00) | 0.00 (0.00) |
| lie_back | 0.97 | 0.86 | 0.24 (0.14) | 0.59 (0.28) | 0.28 (0.18) | 0.05 (0.01) | 0.19 (0.19) | 0.54 (0.23) | 0.43 (0.22) |
| sit_side | 0.98 | 0.93 | 0.03 (0.01) | 0.18 (0.10) | 0.35 (0.17) | 0.00 (0.00) | 0.01 (0.01) | 0.05 (0.10) | 0.28 (0.17) |
| sit_hand_on_legs | 0.98 | 0.97 | 0.29 (0.14) | 0.42 (0.10) | 0.53 (0.06) | 0.00 (0.00) | 0.04 (0.08) | 0.04 (0.03) | 0.59 (0.13) |
| sit_hand_behind | 0.99 | 0.93 | 0.23 (0.16) | 0.66 (0.08) | 0.60 (0.11) | 0.02 (0.02) | 0.03 (0.06) | 0.15 (0.16) | 0.60 (0.11) |
| knees_and_hands | 0.98 | 0.92 | 0.38 (0.15) | 0.71 (0.08) | 0.83 (0.06) | 0.03 (0.01) | 0.18 (0.15) | 0.46 (0.13) | 0.73 (0.11) |
| bridge_front | 0.98 | 0.82 | 0.12 (0.10) | 0.50 (0.41) | 0.74 (0.07) | 0.05 (0.02) | 0.23 (0.11) | 0.44 (0.02) | 0.67 (0.19) |
| push_up | 0.97 | 0.89 | 0.04 (0.05) | 0.35 (0.24) | 0.46 (0.11) | 0.01 (0.01) | 0.01 (0.01) | 0.02 (0.02) | 0.11 (0.05) |
| handstand | 0.84 | 0.00 | 0.00 (0.00) | 0.01 (0.01) | 0.00 (0.00) | 0.02 (0.01) | 0.00 (0.00) | 0.00 (0.00) | 0.05 (0.04) |
| handstand_right_leg_bent | 0.96 | 0.05 | 0.00 (0.00) | 0.00 (0.00) | 0.00 (0.00) | 0.01 (0.01) | 0.00 (0.00) | 0.00 (0.00) | 0.02 (0.02) |
| Average | 0.96 | 0.47 | 0.20 | 0.61 | 0.67 | 0.07 | 0.18 | 0.46 | 0.68 |
| Median | 0.98 | 0.31 | 0.17 | 0.70 | 0.77 | 0.04 | 0.11 | 0.46 | 0.74 |

Table 20: Humanoid Environment. Proximity over goal poses for goal-reaching evaluation.

We further mention results for two baselines that performed poorly in our tests. First, similarly to DIFFUSER, we tested H-GAP (Jiang et al., 2024) trained on the union of the AMASS-ACT dataset and FB-CPR replay buffer. Despite conducting extensive hyper-parameter search based on the default settings reported in Jiang et al. (2024) and scaling the model size, we encountered challenges in training an accurate Prior Transformer and we were unable to achieve satisfactory performance on the downstream tasks. We obtained an average normalized performance of 0.05 in reward optimization on a subset of stand and locomotion tasks. We did not test the other modalities. Second, we also tested planning with a learned model. Specifically, we trained an MLP network on the same offline dataset to predict the next state given a state-action pair. We then used this learned model in MPPI and evaluated its performance on the same subset of tasks as H-GAP. The results showed that MPPI with the learned model achieved a low normalized return of 0.03. We believe that this is due to MPPI's action sampling leading to out-of-distribution action plans, which can cause the model to struggle with distribution shift and compounding errors when chaining predictions. Some form of pessimistic planning is necessary when using a learned model to avoid deviating too much from the observed samples. Unlike MPPI, Diffuser achieves this by sampling action plans that are likely under the offline data distribution. For more details on the results of H-GAP and MPPI with the learned model, see Table 23.

| Goal | TD3 | MPPI | Diffuser | Goal-GAIL | Goal-TD3 | PHC | CALM | ASE | FB-CPR |
|---|---|---|---|---|---|---|---|---|---|
| | | | | Success | | | | | |
| t_pose | 1.00 | 0.75 | 0.80 (0.07) | 1.00 (0.00) | 1.00 (0.00) | 0.09 (0.04) | 0.21 (0.40) | 0.98 (0.04) | 1.00 (0.00) |
| t_pose_lower_arms | 1.00 | 0.75 | 0.78 (0.13) | 1.00 (0.00) | 1.00 (0.00) | 0.35 (0.13) | 0.49 (0.43) | 0.90 (0.19) | 1.00 (0.00) |
| t_pose_bow_head | 1.00 | 0.90 | 0.77 (0.15) | 1.00 (0.00) | 1.00 (0.00) | 0.06 (0.06) | 0.29 (0.39) | 0.37 (0.32) | 1.00 (0.00) |
| u_stretch_y_right | 1.00 | 0.65 | 0.01 (0.02) | 0.36 (0.28) | 0.80 (0.27) | 0.01 (0.02) | 0.00 (0.00) | 0.04 (0.05) | 0.53 (0.32) |
| u_stretch_y_left | 1.00 | 0.65 | 0.00 (0.00) | 0.10 (0.17) | 0.16 (0.31) | 0.00 (0.00) | 0.00 (0.00) | 0.00 (0.00) | 0.30 (0.20) |
| u_stretch_z_right | 1.00 | 0.80 | 0.00 (0.00) | 0.23 (0.30) | 0.38 (0.44) | 0.04 (0.01) | 0.00 (0.00) | 0.01 (0.02) | 0.55 (0.24) |
| u_stretch_z_left | 1.00 | 0.70 | 0.02 (0.02) | 0.82 (0.36) | 0.99 (0.01) | 0.02 (0.00) | 0.00 (0.00) | 0.06 (0.09) | 0.96 (0.07) |
| u_stretch_x_back | 1.00 | 0.25 | 0.00 (0.00) | 0.26 (0.36) | 0.40 (0.42) | 0.04 (0.03) | 0.00 (0.00) | 0.39 (0.45) | 0.87 (0.08) |
| u_stretch_x_front_part | 1.00 | 1.00 | 0.59 (0.18) | 0.93 (0.11) | 1.00 (0.00) | 0.05 (0.09) | 0.05 (0.09) | 0.36 (0.24) | 1.00 (0.00) |
| u_stretch_x_front_full | 1.00 | 1.00 | 0.02 (0.02) | 0.34 (0.32) | 0.64 (0.36) | 0.00 (0.00) | 0.00 (0.00) | 0.21 (0.18) | 0.82 (0.30) |
| crossed_arms | 1.00 | 0.60 | 0.04 (0.05) | 0.40 (0.29) | 0.56 (0.32) | 0.01 (0.02) | 0.01 (0.02) | 0.06 (0.07) | 0.63 (0.22) |
| scratching_head | 1.00 | 0.80 | 0.30 (0.25) | 1.00 (0.00) | 0.99 (0.02) | 0.04 (0.02) | 0.01 (0.02) | 0.96 (0.06) | 1.00 (0.00) |
| right_hand_wave | 1.00 | 0.70 | 0.37 (0.16) | 0.99 (0.02) | 1.00 (0.00) | 0.02 (0.02) | 0.06 (0.12) | 0.99 (0.02) | 1.00 (0.00) |
| x_strech | 1.00 | 0.60 | 0.12 (0.09) | 0.54 (0.40) | 0.87 (0.15) | 0.03 (0.03) | 0.00 (0.00) | 0.45 (0.37) | 0.80 (0.23) |
| i_strech | 0.67 | 0.00 | 0.00 (0.00) | 0.00 (0.00) | 0.30 (0.40) | 0.00 (0.00) | 0.00 (0.00) | 0.00 (0.00) | 0.25 (0.38) |
| arms_stretch | 1.00 | 0.60 | 0.04 (0.05) | 0.00 (0.00) | 0.21 (0.25) | 0.04 (0.03) | 0.00 (0.00) | 0.00 (0.00) | 0.00 (0.00) |
| drinking_from_bottle | 1.00 | 0.70 | 0.01 (0.02) | 0.00 (0.00) | 0.40 (0.49) | 0.02 (0.02) | 0.00 (0.00) | 0.00 (0.00) | 0.86 (0.28) |
| arm_on_chest | 1.00 | 0.80 | 0.02 (0.04) | 0.88 (0.16) | 1.00 (0.00) | 0.00 (0.00) | 0.01 (0.01) | 0.81 (0.21) | 0.99 (0.02) |
| pre_throw | 0.00 | 0.00 | 0.00 (0.00) | 0.00 (0.00) | 0.00 (0.00) | 0.06 (0.04) | 0.00 (0.00) | 0.00 (0.00) | 0.00 (0.00) |
| egyptian | 1.00 | 0.65 | 0.03 (0.02) | 0.43 (0.36) | 0.80 (0.30) | 0.02 (0.02) | 0.00 (0.00) | 0.30 (0.35) | 1.00 (0.00) |
| zombie | 1.00 | 0.75 | 0.35 (0.16) | 0.97 (0.06) | 1.00 (0.00) | 0.04 (0.03) | 0.00 (0.00) | 0.74 (0.26) | 1.00 (0.00) |
| stand_martial_arts | 1.00 | 0.90 | 0.41 (0.18) | 1.00 (0.00) | 1.00 (0.00) | 0.04 (0.04) | 0.00 (0.00) | 0.82 (0.17) | 1.00 (0.00) |
| peekaboo | 0.66 | 0.60 | 0.00 (0.00) | 0.76 (0.35) | 0.51 (0.39) | 0.04 (0.05) | 0.00 (0.00) | 0.58 (0.35) | 0.89 (0.22) |
| dance | 1.00 | 0.70 | 0.16 (0.08) | 0.94 (0.12) | 1.00 (0.00) | 0.00 (0.00) | 0.02 (0.03) | 0.67 (0.39) | 1.00 (0.00) |
| kneel_left | 1.00 | 1.00 | 0.10 (0.12) | 0.31 (0.30) | 1.00 (0.00) | 0.00 (0.00) | 0.00 (0.00) | 0.00 (0.00) | 0.90 (0.10) |
| crouch_high | 1.00 | 1.00 | 0.75 (0.10) | 1.00 (0.00) | 1.00 (0.00) | 0.55 (0.11) | 0.37 (0.41) | 0.67 (0.28) | 1.00 (0.00) |
| crouch_medium | 1.00 | 1.00 | 0.97 (0.04) | 1.00 (0.00) | 1.00 (0.00) | 0.42 (0.14) | 0.44 (0.38) | 0.53 (0.33) | 1.00 (0.00) |
| crouch_low | 1.00 | 0.95 | 0.00 (0.00) | 0.57 (0.38) | 0.45 (0.45) | 0.02 (0.01) | 0.00 (0.00) | 0.01 (0.03) | 0.72 (0.27) |
| squat_pre_jump | 1.00 | 1.00 | 0.02 (0.02) | 0.01 (0.02) | 0.02 (0.03) | 0.01 (0.02) | 0.00 (0.00) | 0.09 (0.16) | 0.25 (0.25) |
| squat_hands_on_ground | 1.00 | 0.40 | 0.00 (0.00) | 0.00 (0.00) | 0.64 (0.45) | 0.00 (0.00) | 0.00 (0.00) | 0.00 (0.00) | 0.10 (0.20) |
| side_high_kick | 1.00 | 0.65 | 0.00 (0.00) | 0.00 (0.00) | 0.00 (0.00) | 0.00 (0.00) | 0.00 (0.00) | 0.00 (0.00) | 0.00 (0.00) |
| pre_front_kick | 1.00 | 0.70 | 0.01 (0.02) | 0.23 (0.39) | 0.40 (0.49) | 0.04 (0.03) | 0.00 (0.00) | 0.02 (0.03) | 0.57 (0.36) |
| arabesque_hold_foot | 0.66 | 0.60 | 0.01 (0.02) | 0.00 (0.00) | 0.00 (0.00) | 0.01 (0.01) | 0.00 (0.00) | 0.00 (0.00) | 0.00 (0.00) |
| hold_right_foot | 1.00 | 0.70 | 0.00 (0.00) | 0.00 (0.00) | 0.01 (0.01) | 0.01 (0.01) | 0.00 (0.00) | 0.11 (0.21) | 0.44 (0.42) |
| hold_left_foot | 1.00 | 0.70 | 0.00 (0.00) | 0.20 (0.26) | 0.25 (0.36) | 0.00 (0.00) | 0.00 (0.00) | 0.00 (0.00) | 0.25 (0.38) |
| bend_on_left_leg | 1.00 | 1.00 | 0.00 (0.00) | 0.00 (0.00) | 0.00 (0.00) | 0.05 (0.04) | 0.00 (0.00) | 0.00 (0.00) | 0.00 (0.00) |
| lie_front | 1.00 | 0.90 | 0.10 (0.20) | 0.01 (0.02) | 0.00 (0.00) | 0.00 (0.00) | 0.00 (0.00) | 0.01 (0.02) | 0.00 (0.00) |
| crawl_backward | 1.00 | 0.95 | 0.00 (0.00) | 0.00 (0.00) | 0.00 (0.00) | 0.00 (0.00) | 0.00 (0.00) | 0.00 (0.00) | 0.00 (0.00) |
| lie_back_knee_bent | 1.00 | 0.85 | 0.00 (0.00) | 0.00 (0.00) | 0.00 (0.00) | 0.02 (0.03) | 0.00 (0.00) | 0.00 (0.00) | 0.00 (0.00) |
| lie_side | 1.00 | 0.90 | 0.00 (0.00) | 0.00 (0.00) | 0.00 (0.00) | 0.02 (0.02) | 0.00 (0.00) | 0.00 (0.00) | 0.00 (0.00) |
| crunch | 1.00 | 0.55 | 0.00 (0.00) | 0.00 (0.00) | 0.00 (0.00) | 0.00 (0.00) | 0.00 (0.00) | 0.00 (0.00) | 0.00 (0.00) |
| lie_back | 1.00 | 0.90 | 0.02 (0.04) | 0.31 (0.39) | 0.00 (0.00) | 0.08 (0.03) | 0.00 (0.00) | 0.13 (0.27) | 0.00 (0.00) |
| sit_side | 1.00 | 0.95 | 0.00 (0.00) | 0.00 (0.00) | 0.00 (0.00) | 0.00 (0.00) | 0.00 (0.00) | 0.00 (0.00) | 0.00 (0.00) |
| sit_hand_on_legs | 1.00 | 1.00 | 0.00 (0.00) | 0.00 (0.00) | 0.00 (0.00) | 0.00 (0.00) | 0.00 (0.00) | 0.00 (0.00) | 0.00 (0.00) |
| sit_hand_behind | 1.00 | 0.95 | 0.01 (0.02) | 0.00 (0.00) | 0.00 (0.00) | 0.02 (0.05) | 0.00 (0.00) | 0.00 (0.00) | 0.00 (0.00) |
| knees_and_hands | 1.00 | 0.95 | 0.06 (0.07) | 0.00 (0.00) | 0.18 (0.27) | 0.04 (0.02) | 0.00 (0.00) | 0.00 (0.00) | 0.01 (0.02) |
| bridge_front | 1.00 | 0.85 | 0.00 (0.00) | 0.06 (0.08) | 0.00 (0.00) | 0.08 (0.04) | 0.00 (0.00) | 0.00 (0.00) | 0.17 (0.31) |
| push_up | 1.00 | 0.95 | 0.00 (0.00) | 0.00 (0.00) | 0.00 (0.00) | 0.00 (0.00) | 0.00 (0.00) | 0.00 (0.00) | 0.00 (0.00) |
| handstand | 0.67 | 0.00 | 0.00 (0.00) | 0.00 (0.00) | 0.00 (0.00) | 0.01 (0.02) | 0.00 (0.00) | 0.00 (0.00) | 0.00 (0.00) |
| handstand_right_leg_bent | 1.00 | 0.10 | 0.00 (0.00) | 0.00 (0.00) | 0.00 (0.00) | 0.01 (0.02) | 0.00 (0.00) | 0.00 (0.00) | 0.00 (0.00) |
| Average | 0.95 | 0.73 | 0.14 | 0.35 | 0.44 | 0.05 | 0.04 | 0.22 | 0.48 |
| Median | 1.00 | 0.75 | 0.01 | 0.22 | 0.39 | 0.02 | 0.00 | 0.01 | 0.48 |

Table 21: Humanoid Environment. Success rate over different goal poses in the goal-reaching evaluation.

| Task | H-GAP | | MPPI with learned world model | |
|---|---|---|---|---|
| | Normalized | | Normalized | |
| move-ego-0-0 | 0.123 | 33.78 | 0.069 | 19.05 |
| move-ego-0-2 | 0.036 | 9.16 | 0.040 | 10.24 |
| move-ego-0-4 | 0.028 | 6.82 | 0.038 | 9.21 |
| move-ego--90-2 | 0.041 | 10.56 | 0.032 | 8.26 |
| move-ego--90-4 | 0.032 | 7.97 | 0.026 | 6.41 |
| move-ego-90-2 | 0.049 | 12.46 | 0.036 | 9.19 |
| move-ego-90-4 | 0.039 | 9.54 | 0.024 | 6.00 |
| move-ego-180-2 | 0.053 | 13.68 | 0.024 | 6.26 |
| move-ego-180-4 | 0.042 | 10.41 | 0.019 | 4.76 |
| Average | 0.05 | 12.71 | 0.03 | 8.82 |
| Median | 0.04 | 10.41 | 0.03 | 8.26 |

Table 23: Humanoid Environment. Average Return of H-GAP and MPPI with learned world model on a subset of stand and locomotion tasks.

**EMD**

| Dataset | Goal-GAIL (1motion) train | test | PHC (1motion) train | test | ASE train | test | CALM train | test | Goal-GAIL train | test | Goal-TD3 train | test | PHC train | test | FB-CPR train | test |
|---|---|---|---|---|---|---|---|---|---|---|---|---|---|---|---|---|
| ACCAD | 1.18 (0.37) | 1.22 (0.35) | 1.13 (1.44) | 0.87 (0.27) | 2.34 (0.03) | 2.53 (0.03) | 2.05 (0.07) | 2.25 (0.04) | 2.02 (0.04) | 2.22 (0.03) | 1.65 (0.09) | 1.77 (0.09) | 1.95 (0.06) | 2.08 (0.04) | 1.67 (0.01) | 1.84 (0.03) |
| BMLhandball | 1.55 (0.14) | 1.55 (0.18) | 1.44 (1.83) | 0.96 (0.14) | 2.63 (0.08) | 2.66 (0.07) | 2.16 (0.05) | 2.24 (0.06) | 2.14 (0.03) | 2.19 (0.06) | 1.73 (0.08) | 1.77 (0.13) | 2.06 (0.09) | 2.07 (0.11) | 1.75 (0.03) | 1.76 (0.05) |
| BMLmovi | 1.06 (0.26) | 1.08 (0.29) | 1.13 (1.54) | 1.15 (1.47) | 2.00 (0.05) | 1.96 (0.02) | 1.71 (0.04) | 1.74 (0.04) | 1.67 (0.01) | 1.69 (0.02) | 1.42 (0.08) | 1.44 (0.10) | 1.76 (0.07) | 1.74 (0.09) | 1.37 (0.01) | 1.38 (0.02) |
| BioMotionLab | 1.24 (0.25) | 1.25 (0.36) | 1.23 (1.56) | 1.26 (1.63) | 2.10 (0.02) | 2.06 (0.02) | 1.78 (0.02) | 1.76 (0.02) | 1.86 (0.02) | 1.86 (0.04) | 1.48 (0.07) | 1.47 (0.08) | 1.70 (0.06) | 1.67 (0.06) | 1.48 (0.01) | 1.47 (0.01) |
| CMU | 1.17 (0.35) | 1.18 (0.38) | 1.15 (1.64) | 1.06 (1.27) | 2.23 (0.02) | 2.23 (0.02) | 1.86 (0.04) | 1.90 (0.03) | 1.87 (0.02) | 1.92 (0.02) | 1.51 (0.08) | 1.54 (0.09) | 1.78 (0.07) | 1.79 (0.06) | 1.52 (0.01) | 1.54 (0.01) |
| DFaust | 0.96 (0.26) | 1.15 (0.33) | 1.71 (2.87) | 0.83 (0.26) | 2.05 (0.06) | 2.28 (0.14) | 1.74 (0.05) | 1.86 (0.06) | 1.72 (0.03) | 1.96 (0.03) | 1.41 (0.07) | 1.51 (0.08) | 1.71 (0.06) | 1.74 (0.07) | 1.43 (0.01) | 1.57 (0.02) |
| DanceDB | 1.48 (0.22) | 1.63 (0.07) | 2.11 (2.35) | 1.54 (0.04) | 2.70 (0.04) | 3.05 (0.06) | 2.39 (0.02) | 2.76 (0.09) | 2.38 (0.03) | 2.78 (0.06) | 1.96 (0.11) | 2.16 (0.11) | 2.19 (0.06) | 2.42 (0.08) | 1.94 (0.02) | 2.08 (0.03) |
| EKUT | 0.79 (0.17) | 0.89 (0.22) | 0.95 (1.63) | 1.49 (2.42) | 1.70 (0.03) | 1.79 (0.03) | 1.33 (0.03) | 1.44 (0.02) | 1.35 (0.02) | 1.45 (0.03) | 1.17 (0.07) | 1.21 (0.06) | 1.38 (0.07) | 1.45 (0.05) | 1.10 (0.00) | 1.23 (0.04) |
| Eyes | 1.32 (0.22) | 1.32 (0.23) | 1.35 (3.12) | 1.44 (1.60) | 2.14 (0.03) | 2.15 (0.04) | 1.90 (0.03) | 1.92 (0.01) | 1.83 (0.03) | 1.85 (0.04) | 1.62 (0.10) | 1.63 (0.11) | 1.85 (0.07) | 1.81 (0.07) | 1.57 (0.01) | 1.55 (0.01) |
| HumanEva | 1.02 (0.23) | 1.11 (0.21) | 0.88 (0.37) | 1.06 (0.14) | 2.05 (0.04) | 2.16 (0.12) | 1.74 (0.08) | 1.87 (0.09) | 1.82 (0.02) | 1.86 (0.06) | 1.42 (0.08) | 1.52 (0.13) | 1.64 (0.08) | 1.74 (0.11) | 1.41 (0.03) | 1.59 (0.05) |
| KIT | 0.89 (0.25) | 0.89 (0.23) | 1.00 (1.24) | 0.98 (1.07) | 1.71 (0.03) | 1.68 (0.03) | 1.35 (0.01) | 1.37 (0.05) | 1.36 (0.03) | 1.36 (0.02) | 1.17 (0.08) | 1.17 (0.08) | 1.42 (0.07) | 1.40 (0.07) | 1.12 (0.01) | 1.13 (0.01) |
| MPI | 1.28 (0.28) | 1.26 (0.27) | 1.23 (1.19) | 1.57 (1.90) | 2.42 (0.02) | 2.42 (0.05) | 2.08 (0.02) | 2.14 (0.06) | 2.04 (0.03) | 2.10 (0.04) | 1.68 (0.08) | 1.72 (0.08) | 1.96 (0.06) | 2.00 (0.07) | 1.68 (0.01) | 1.76 (0.01) |
| SFU | 1.20 (0.37) | 1.43 (0.14) | 0.95 (0.39) | 1.29 (0.42) | 2.63 (0.01) | 3.24 (0.08) | 2.25 (0.06) | 2.68 (0.08) | 2.26 (0.06) | 2.69 (0.04) | 1.77 (0.08) | 2.11 (0.08) | 2.04 (0.08) | 2.41 (0.11) | 1.88 (0.01) | 2.27 (0.04) |
| TotalCapture | 1.15 (0.14) | 1.17 (0.16) | 1.23 (1.21) | 1.10 (0.28) | 2.06 (0.06) | 2.16 (0.05) | 1.74 (0.02) | 1.85 (0.02) | 1.76 (0.03) | 1.86 (0.03) | 1.45 (0.09) | 1.51 (0.12) | 1.73 (0.11) | 1.71 (0.10) | 1.44 (0.03) | 1.50 (0.02) |
| Transitions | 1.15 (0.08) | 1.17 (0.07) | 2.12 (2.90) | 2.65 (3.37) | 2.31 (0.05) | 2.40 (0.04) | 1.99 (0.04) | 2.04 (0.06) | 2.01 (0.05) | 2.05 (0.02) | 1.53 (0.08) | 1.59 (0.09) | 1.77 (0.05) | 1.83 (0.05) | 1.54 (0.01) | 1.59 (0.02) |

**SUCCESS**

| Dataset | Goal-GAIL (1motion) train | test | PHC (1motion) train | test | ASE train | test | CALM train | test | Goal-GAIL train | test | Goal-TD3 train | test | PHC train | test | FB-CPR train | test |
|---|---|---|---|---|---|---|---|---|---|---|---|---|---|---|---|---|
| ACCAD | 0.20 (0.40) | 0.24 (0.43) | 0.94 (0.23) | 1.00 (0.00) | 0.31 (0.02) | 0.25 (0.02) | 0.58 (0.05) | 0.46 (0.05) | 0.24 (0.01) | 0.22 (0.04) | 0.80 (0.02) | 0.66 (0.04) | 0.68 (0.03) | 0.56 (0.08) | 0.67 (0.03) | 0.49 (0.03) |
| BMLhandball | 0.00 (0.00) | 0.00 (0.00) | 0.91 (0.28) | 1.00 (0.00) | 0.02 (0.03) | 0.00 (0.00) | 0.10 (0.07) | 0.04 (0.08) | 0.00 (0.00) | 0.00 (0.00) | 0.80 (0.12) | 0.88 (0.16) | 0.50 (0.04) | 0.40 (0.18) | 0.30 (0.13) | 0.24 (0.15) |
| BMLmovi | 0.22 (0.41) | 0.19 (0.39) | 0.96 (0.20) | 0.96 (0.20) | 0.51 (0.01) | 0.57 (0.02) | 0.78 (0.02) | 0.82 (0.03) | 0.28 (0.02) | 0.25 (0.02) | 0.97 (0.00) | 0.96 (0.01) | 0.87 (0.01) | 0.87 (0.03) | 0.88 (0.02) | 0.89 (0.02) |
| BioMotionLab | 0.04 (0.18) | 0.06 (0.23) | 0.91 (0.28) | 0.92 (0.27) | 0.12 (0.02) | 0.14 (0.03) | 0.53 (0.06) | 0.60 (0.04) | 0.04 (0.00) | 0.06 (0.01) | 0.80 (0.03) | 0.83 (0.02) | 0.72 (0.02) | 0.76 (0.01) | 0.75 (0.02) | 0.79 (0.02) |
| CMU | 0.16 (0.37) | 0.18 (0.39) | 0.93 (0.26) | 0.95 (0.23) | 0.27 (0.02) | 0.31 (0.02) | 0.60 (0.02) | 0.63 (0.04) | 0.21 (0.01) | 0.22 (0.02) | 0.86 (0.01) | 0.86 (0.01) | 0.77 (0.01) | 0.78 (0.03) | 0.86 (0.01) | 0.74 (0.02) |
| DFaust | 0.47 (0.50) | 0.33 (0.47) | 0.89 (0.32) | 1.00 (0.00) | 0.48 (0.03) | 0.47 (0.19) | 0.74 (0.02) | 0.71 (0.05) | 0.48 (0.03) | 0.53 (0.04) | 0.95 (0.01) | 1.00 (0.00) | 0.86 (0.03) | 0.96 (0.05) | 0.27 (0.06) | 0.84 (0.05) |
| DanceDB | 0.04 (0.20) | 0.00 (0.00) | 0.61 (0.49) | 1.00 (0.00) | 0.04 (0.00) | 0.00 (0.00) | 0.10 (0.02) | 0.00 (0.00) | 0.05 (0.02) | 0.00 (0.00) | 0.62 (0.08) | 0.70 (0.24) | 0.30 (0.08) | 0.40 (0.20) | 0.94 (0.04) | 0.50 (0.00) |
| EKUT | 0.30 (0.46) | 0.36 (0.48) | 0.96 (0.20) | 0.86 (0.35) | 0.49 (0.05) | 0.51 (0.11) | 0.90 (0.02) | 0.84 (0.03) | 0.32 (0.02) | 0.34 (0.08) | 0.99 (0.01) | 1.00 (0.00) | 0.94 (0.02) | 0.84 (0.05) | 0.79 (0.02) | 0.81 (0.07) |
| Eyes | 0.00 (0.04) | 0.00 (0.00) | 0.91 (0.29) | 0.85 (0.35) | 0.24 (0.05) | 0.29 (0.10) | 0.65 (0.02) | 0.66 (0.02) | 0.11 (0.02) | 0.18 (0.08) | 0.92 (0.01) | 0.91 (0.02) | 0.76 (0.01) | 0.83 (0.03) | 0.92 (0.04) | 0.79 (0.03) |
| HumanEva | 0.20 (0.40) | 0.00 (0.00) | 0.96 (0.20) | 1.00 (0.00) | 0.43 (0.08) | 0.27 (0.39) | 0.83 (0.08) | 0.87 (0.16) | 0.17 (0.02) | 0.00 (0.00) | 0.99 (0.02) | 1.00 (0.00) | 0.94 (0.03) | 0.93 (0.13) | 0.95 (0.01) | 0.93 (0.13) |
| KIT | 0.41 (0.49) | 0.44 (0.50) | 0.97 (0.17) | 0.97 (0.18) | 0.56 (0.04) | 0.59 (0.05) | 0.91 (0.01) | 0.92 (0.01) | 0.40 (0.02) | 0.40 (0.04) | 0.98 (0.00) | 0.98 (0.00) | 0.95 (0.00) | 0.94 (0.01) | 0.51 (0.02) | 0.96 (0.01) |
| MPI | 0.07 (0.25) | 0.07 (0.25) | 0.86 (0.35) | 0.83 (0.38) | 0.12 (0.01) | 0.14 (0.04) | 0.35 (0.02) | 0.39 (0.04) | 0.09 (0.01) | 0.13 (0.03) | 0.71 (0.02) | 0.74 (0.03) | 0.53 (0.02) | 0.50 (0.08) | 0.50 (0.06) | 0.56 (0.05) |
| SFU | 0.00 (0.00) | 0.00 (0.00) | 0.97 (0.18) | 0.67 (0.47) | 0.05 (0.03) | 0.00 (0.00) | 0.38 (0.05) | 0.07 (0.13) | 0.00 (0.00) | 0.00 (0.00) | 0.73 (0.03) | 0.60 (0.13) | 0.55 (0.03) | 0.47 (0.27) | 0.55 (0.07) | 0.13 (0.16) |
| TotalCapture | 0.00 (0.00) | 0.00 (0.00) | 0.73 (0.45) | 0.75 (0.43) | 0.00 (0.00) | 0.00 (0.00) | 0.16 (0.04) | 0.20 (0.19) | 0.00 (0.00) | 0.00 (0.00) | 0.79 (0.03) | 0.70 (0.10) | 0.46 (0.04) | 0.40 (0.12) | 0.62 (0.04) | 0.35 (0.12) |
| Transitions | 0.00 (0.00) | 0.00 (0.00) | 0.84 (0.36) | 0.82 (0.39) | 0.04 (0.02) | 0.04 (0.04) | 0.33 (0.03) | 0.36 (0.16) | 0.00 (0.00) | 0.00 (0.00) | 0.81 (0.03) | 0.78 (0.09) | 0.58 (0.04) | 0.40 (0.04) | 0.62 (0.04) | 0.65 (0.11) |

Table 22: Humanoid Environment. Average performance over each sub-set of the AMASS dataset used in the tracking evaluation.

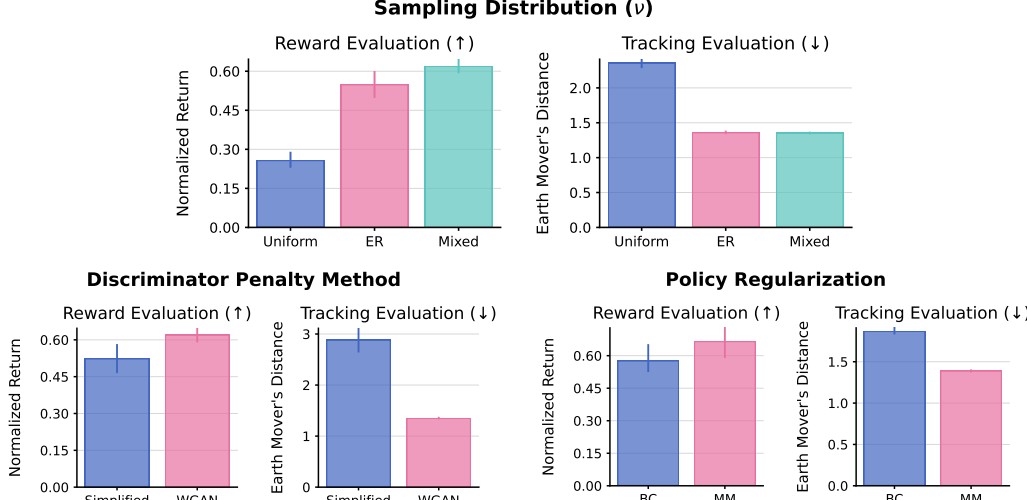

Figure 6: **Additional FB-CPR Ablations**. (TOP) Ablating the sampling distribution $\nu$. (BOTTOM LEFT) Ablating the discriminator gradient penalty method. (BOTTOM RIGHT) Ablating the policy regularization method between behavior cloning and moment matching when given action labels. All ablations are averaged over 5 seeds with ranges denoting bootstrapped 95% confidence intervals.

## D.2 ABLATIONS

In this section we detail additional ablations into the components of FB-CPR.

**Which gradient penalty better stabilizes the discriminator in FB-CPR?** Algorithms requiring bi-level optimization through a min-max game are known to be unstable and typically require strong forms of regularization (e.g., Gulrajani et al., 2017; Miyato et al., 2018). Prior works like CALM (Tessler et al., 2023), ASE (Peng et al., 2022), and AMP (Peng et al., 2021) employ what we will refer to as the simplified gradient penalty on the discriminator to stabilize training:

$$\lambda_{\mathrm{GP}}\,\mathbb{E}_{\tau\sim\mathcal{M},s\sim\tau}\left[\left\|\left.\nabla_{x,z}D(x,z)\right|_{(x,z)=(s,\mathrm{ER}_{\mathrm{FB}}(\tau))}\right\|_2^2\right]\,.$$

Alternatively, other works in Inverse Reinforcement Learning (e.g., Swamy et al., 2021; 2022; Ren et al., 2024) have had success employing the Wasserstein gradient penalty of Gulrajani et al. (2017):

$$\lambda_{\mathrm{GP}}\,\mathbb{E}_{\substack{z\sim\nu,s\sim\rho^{\pi_z},\tau\sim\mathcal{M},s'\sim\tau \\ t\sim\mathrm{Unif}(0,1)}}\left[\left(\left\|\left.\nabla_{x,z'}D(x,z')\right|_{x=ts+(1-t)s',z'=tz+(1-t)\mathrm{ER}_{\mathrm{FB}}(\tau)}\right\|_2^2 - 1\right)^2\right]\,.$$

We want to verify which of these two methods better stabilizes training of the discriminator in FB-CPR. To this end, we perform a sweep over $\lambda_{\mathrm{GP}} \in \{0, 1, 5, 10, 15\}$ for both the aforementioned gradient penalties and further averaged over 5 independent seeds. We found that without a gradient penalty, i.e., $\lambda_{\mathrm{GP}} = 0$ training was unstable and lead to subpar performance. For both gradient penalty methods we found that $\lambda_{\mathrm{GP}} = 10$ performed best and as seen in Figure 6 (Left) the Wasserstein gradient penalty ultimately performed best.

**What is gained or lost when ablating the mixture components of $\nu$?** By modelling $\nu$ as a mixture distribution we hypothesize that a tradeoff is introduced depending on the proportion of each component. One of the most natural questions to ask is whether there is anything to be gained by only sampling $\tau \sim \mathcal{M}$ and encoding with $z = \mathrm{ER}_{\mathrm{FB}}(\tau)$. If indeed this component is enabling FB-CPR to accurately reproduce trajectories in $\mathcal{M}$ we may see an improvement in tracking performance perhaps at the cost of diversity impacting reward-optimization performance. On the other hand, increased diversity by only sampling uniformly from the hypersphere may improve reward evaluation performance for reward functions that are not well aligned with any motion in $\mathcal{M}$. We test these hypotheses by training FB-CPR on 1) only $\mathrm{ER}_{\mathrm{FB}}$ encoded subtrajectories from $\mathcal{M}$, 2) only uniformly sampled embeddings from the hypersphere, and 3) the default mixture weights reported in Table 9.

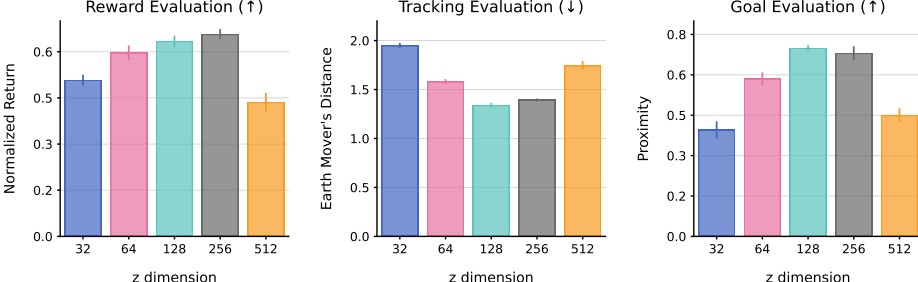

Figure 7: Performance of FB-CPR in the same setting as Table 1 but with different dimensions of the latent space. Results are averaged over 5 seeds with ranges denoting bootstrapped 95% confidence intervals.

Figure 6 confirms that mixed sampling strikes a nice balance between these trade-offs. Indeed, only using $ER_{FB}$ encoded subtrajectories from $\mathcal{M}$ harms reward evaluation performance but surprisingly does not improve on tracking performance. Perhaps unsurprisingly sampling only uniformly from the hypersphere is a weak prior and does not fully leverage the motion dataset resulting in substantially degraded performance across the board.

**Is CPR regularization better than BC if given action labels?** In our work we adopt the moment matching framework to perform policy regularization (Swamy et al., 2021). This framework can be naturally extended to the action-free setting whereas most imitation learning methods require action labels. If we are provided a dataset with action-labels should we continue to adopt the moment matching framework with the conditional discriminator presented herein? To answer this question we curate our own action labelled dataset by relabelling the AMASS dataset with a pre-trained FB-CPR policy. Given this dataset we directly compare the conditional discriminator (Eq. 11) with a modified form of the FB-CPR actor loss that instead performs regularization via behavior cloning,

$$\mathscr{L}_{\text{FB-CPR-BC}}(\pi) = -\mathbb{E}_{z\sim\nu, s\sim\mathcal{D}_{\text{online}}, a\sim\pi_z(\cdot|s)} \left[ F(s,a,z)^\top z \right] - \alpha_{\text{BC}}\, \mathbb{E}_{z\sim\nu, (s,a)\sim\mathcal{M}} \left[ \log \pi_z(a\,|\,s) \right] . \tag{14}$$

We perform a sweep over the strength of the behavior cloning regularization term $\alpha_{\text{BC}} \in \{0.1, 0.2, 0.4, 0.5\}$ and further average these results over 5 seeds. Furthermore, we re-train FB-CPR on the relabelled dataset and also perform a sweep over the CPR regularization coefficient $\alpha_{\text{CPR}} \in \{0.01, 0.03, 0.05\}$. Ultimately, $\alpha_{\text{BC}} = 0.2$ and $\alpha_{\text{CPR}} = 0.01$ performed best with results on reward and tracking evaluation presented in the bottom right panel of Figure 6. We can see that even when given action-labels our action-free discriminator outperforms the BC regularization in both reward and tracking evaluation. This highlights the positive interaction of the conditional discriminator with FB to provide a robust method capable of leveraging action-free demonstrations and notably outperforming a strong action-dependent baseline.

**How does the latent space dimension affect the performance of FB-CPR?** Choosing the dimension $d$ of the latent space built by FB-CPR involves an important trade-off: on the one hand, we would like $d$ to be large so as to have an accurate estimation of the successor measure of the learned policies, which in turns would yield accurate evaluation of the Q function for many rewards and accurate trajectory encoding through $ER_{FB}$ (cf. Section 2). Moreover, as we recall that task inference involves mapping functions of the state space to latent vectors (e.g., by $z = \mathbb{E}_\rho[B(s)R(s)]$ for a reward function $R$ and $z = B(g)$ for a goal $g$), a large dimension $d$ is desirable to make sure as many tasks/behaviors as possible are learned reliably. On the other hand, it is desirable to use a small $d$ to learn a set of behaviors which is as succinct as possible, which would be more efficient to train and to query at inference time, as argued in several works on unsupervised skill discovery (e.g., Eysenbach et al., 2019; Peng et al., 2022; Tessler et al., 2023; Park et al., 2024c).

We demonstrate this trade-off empirically in Figure 7, where we repeat the same experiment as in Table 1 for different values of $d$. We observe a nearly monotonic performance improvement up to dimensions 128 and 256, were performance saturate (with the latter being slightly better on reward tasks and the former being slightly better on tracking and goal reaching). As expected, we qualitatively observe that $d = 32$ and $d = 64$ limit too much the capacity of the latent space, as

| Algorithm | Reward (↑) | Goal | | Tracking - EMD (↓) | | Tracking - Success (↑) | |
|---|---|---|---|---|---|---|---|
| | | Proximity (↑) | Success (↑) | Train | Test | Train | Test |
| FB | 24.47 (1.88) | 0 (0) | 0 (0) | 8.09 (0.21) | 8.19 (0.14) | 0 (0) | 0 (0) |
| SCORE$_\text{norm}$ | 0.10 | 0 | 0 | 0.13 | 0.13 | 0 | 0 |

Table 24: Performance of the FB algorithm (Touati & Ollivier, 2021) in the same setting as Table 1, where SCORE$_\text{norm}$ are normalized w.r.t. the performance of the best baseline in such table.

several of the hardest tasks (e.g., cartwheels or backflips) or the hardest goals (e.g., yoga poses) are not learned at all. On the other hand, we observe a collapse in the learned representation B when moving to very large $d$, which results in the performance drop at $d = 512$. This is mostly due to the fact that several parameters used for the "default" configuration reported in Table 1, and kept constant for all runs in this ablation, are not suitable for training with such large $d$. For instance, the network architecture of F is too small to predict successor features over 512 dimensions, and should be scaled proportionally to d. Similarly, a batch size of 1024 is likely not sufficient to accurately estimate the covariance matrix of B, which is required by the orthonormality and temporal difference losses (cf. Appendix B). Overall we found $d = 256$ to be a good trade-off between capacity, succinctness, and training stability, as FB+CPR with such dimension does not suffer the collapsing issue of $d = 512$ and learns more difficult behaviors than $d = 128$.

**What is the importance of regularizing with unlabeled data?** One may wonder whether regularizing the learned policies towards behaviors in the unlabeled dataset is really needed, or whether the plain FB algorithm of Touati & Ollivier (2021) (i.e., without the CPR part) trained online can already learn useful behaviors and solve many tasks. We report the results of such algorithm, trained with the same parameters used for FB-CPR, in Table 24. The algorithm achieves near-zero performance in all tasks, with only a small improvement over a randomly-initialized untrained policy in reward-based problems and tracking. Such small improvements is due to the fact that the algorithm learned how to roughly stand up, although without being able to maintain a standing position. The main reason behind this failure is that the FB algorithm has no explicit component to encourage discovery of diverse behaviors, except for the purely myopic exploration of TD3 (i.e., perturbing each action component with random noise) which obviously would fail in problems with large state and action spaces. On the other hand, the regularization in FB-CPR overcomes this problem by directing the agent towards learning behaviors in the unlabeled dataset.

### D.3 DIVERSITY, DATASET COVERAGE AND TRANSITIONS

In this section we intend to further investigate the behaviors learned by FB-CPR beyond its performance in solving downstream tasks.

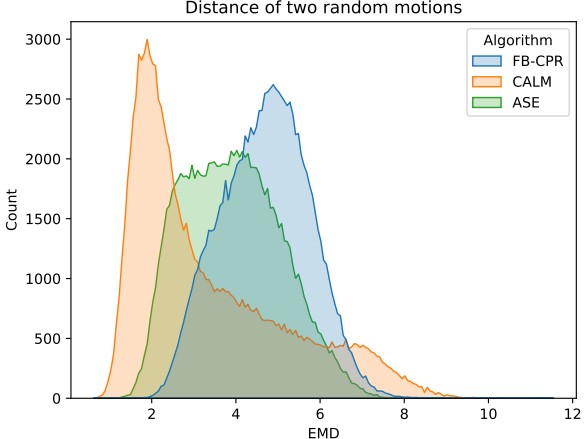

Figure 8: Distribution of EMD distance between trajectories generated by two randomly sampled policies $\pi_z$ and $\pi_{z'}$.

| Algorithm | Diversity |
|-----------|-----------|
| FB-CPR | 4.70 (0.66) |
| CALM | 3.36 (1.15) |
| ASE | 3.91 (0.73) |

Figure 9: Average diversity.

**How diverse are the behaviors learned by FB-CPR?** We want to evaluate the diversity of behaviors encoded in $(\pi_z)$. Given two randomly drawn $z$ and $z'$, we run the two associated policies from the same initial state and we compute the EMD distance between the two resulting trajectories. We repeat this procedure for $n = 100,000$ times and compute

$$\text{Diversity} = \frac{1}{n} \sum_{i=1}^{n} \text{EMD}(\tau_i, \tau_i'). \qquad (15)$$

The values of diversity are presented in Table 9. FB-CPR has the highest diversity. This result is confirmed by looking at the distribution of EMD values between $\tau_i$ and $\tau_i'$ in Fig. 8. FB-CPR has consistently the most diverse results. ASE distribution is shifted toward lower EMD values, which means that its behaviors are less diverse. CALM has mode around 2, which means that its representation has clusters of similar motions, but it is also the algorithm with the wider distribution with EMD distance above 7.0.

**Are FB-CPR behaviors grounded in the behavior dataset $\mathcal{M}$?** While this question is partially answered in the tracking evaluation, we would like to evaluate how much of the motion dataset is actually covered. In fact, a common failure mode of imitation regularization algorithms is the collapse of the learned policies towards accurately matching only a small portion of the demonstrated behaviors. In order to evaluate the level of coverage of the training motion dataset[13], we use a similar metric to the one proposed in (Peng et al., 2022), while accounting for the differences in the dataset: we have a much larger (8902 vs 187 motions) and less curated dataset, where the length of the motions has much larger variance.

We first sample a random $z$ and generate a trajectory $\tau_z$ by executing the corresponding policy $\pi_z$ for 200 steps starting from a T-pose configuration. Then, we calculate the EMD between $\tau_z$ and each motion in $\mathcal{M}$ and we select the motion $m_z^*$ with the lowest EMD as the one best matching $\tau$:

$$m_z^\star = \underset{m^i \in \mathcal{M}}{\arg\min} \, \text{EMD}(\tau_z, m^i). \qquad (16)$$

We use EMD instead of time-aligned distance metrics to account for the fact that $\tau_z$ is executed from an initial state that could be fairly far from a motion in $\mathcal{M}$. We repeat this procedure 10,000

---

[13]Notice that here we are not trying to evaluate the generalization capabilities of the model, which is the focus of Sect. 4.

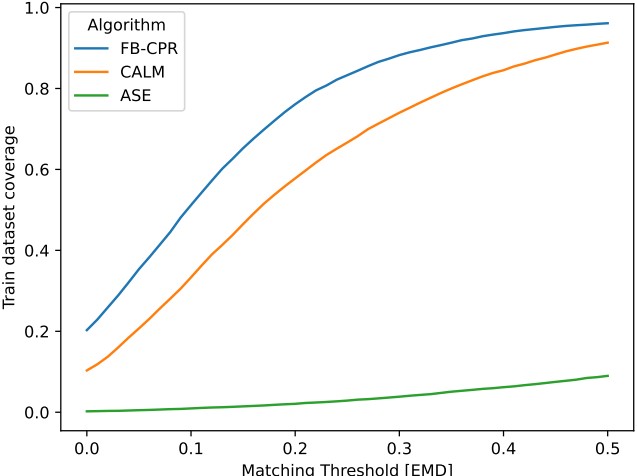

Figure 10: Relation between the threshold used to determine motion matching and the coverage of the train dataset by the randomly sampled policies.

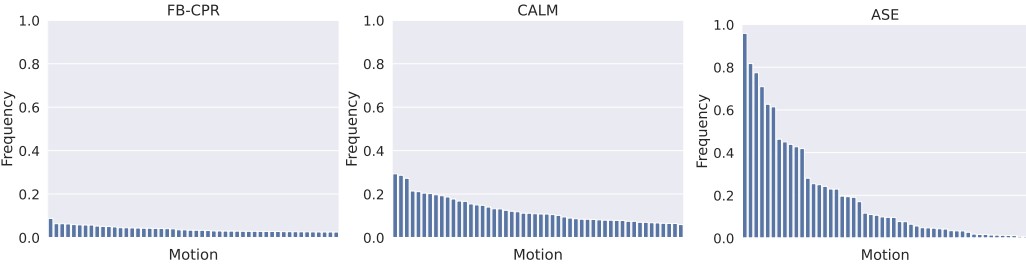

Figure 11: The frequency of the 50 most matched motions with multi-matching and $\text{MATCH}_{\text{THRESHOLD}} = 0.1$. Note that each algorithm matches to a different set of most frequent motions.

times and calculate the frequency of selecting each motion from the dataset. The dataset coverage is defined as the ratio of the number of the motions selected at least once to the number of motions in the training dataset.

As the train motion dataset is two orders of magnitude larger than the one used in (Peng et al., 2022), it is naturally harder to cover $\mathcal{M}$. To mitigate this issue, we propose a multiple-matching approach: a motion $m$ is considered as matching, if its EMD to the closest motion from $\mathcal{M}$ is no larger than

$$\text{EMD}(\tau_z, m_z^\star) + \text{MATCH}_{\text{THRESHOLD}}. \tag{17}$$

By definition, greater values of the $\text{MATCH}_{\text{THRESHOLD}}$ results in greater coverage, as further motions are matched. Additionally, we observed by qualitative assessment, that when EMD is larger than $4.5$, then the two trajectories are distinct enough to be considered as different behaviors. We therefore discard a matching if the EMD distance of $m^*$ is above $4.5$. The relation between $\text{MATCH}_{\text{THRESHOLD}}$ and the coverage is presented on Fig. 10. It can be observed that FB-CPR has consistently the highest coverage and it smoothly increases with the EMD threshold. CALM has lower coverage, but presents similar coverage pattern. In comparison, the coverage of ASE remains consistently low.

In order to calculate the matching of the top 50 most matched motions used in the further comparison, we used this multi-matching variant with $\text{MATCH}_{\text{THRESHOLD}} = 0.1$. In Fig. 11 we report the frequency of the top 50 most matched motions through this procedure for FB-CPR, CALM, and ASE. ASE has a very skewed distribution, meaning that many policies $\pi_z$ tend to produce trajecto-

ries similar to a very small subset of motions, which suggests some form of coverage collapse. On the other extreme, FB-CPR has a very flat distribution, suggesting that it has a more even coverage of the motions dataset.

**Is FB-CPR capable of motion stitching?** Another possible failure mode is to learn policies that are accurately tracking individual motions but are unable to *stitch* together different motions, i.e., to smoothly transition from one behavior to another. In this case, we sample two embeddings $z_S$ and $z_D$ (respectively source and destination) and we use them to generate a trajectory $\tau$ which is composed of two disjoint sub-trajectories: the first 200 steps are generated with $\pi_{z_S}$ and form sub-trajectory $\tau_S$; after that, the second sub-trajectory $\tau_D$ is generated as the continuation of $\tau_S$, while running policy $\pi_{z_D}$. After their generation, $\tau_S$ and $\tau_D$ are separately matched to the motions using Eq. 16, and a pair of source and destination motion is recorded. To make the process computationally feasible, we restrict our attention to the 50 most frequently matched motions selected in the previous evaluation with Eq. 16, and presented in Fig. 11. The procedure of generating transitioning trajectory is repeated 10,000 times. The *pairwise transition probability* is defined as the probability of matching a destination motion, conditioned on the source motion.

We also define pairwise transition coverage on a dataset as the ratio of the number of pairwise transitions with frequency larger than 0, to the number of all possible pairwise transitions. The pairwise transition probability and respective coverage is reported in Fig. 12. All algorithms have similar overall coverage.

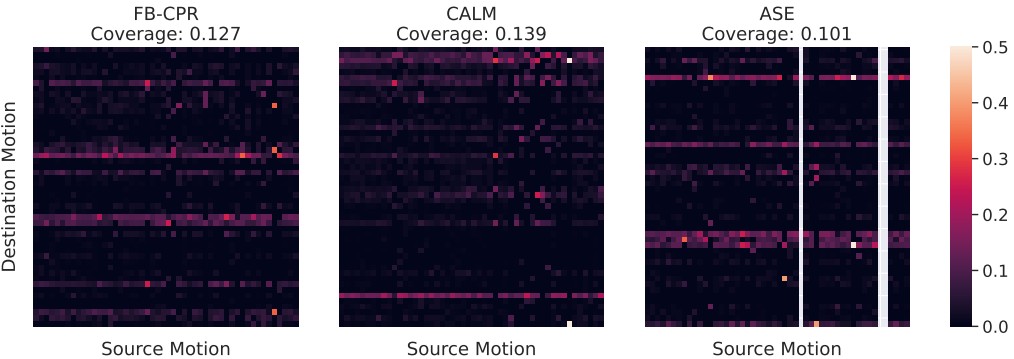

Figure 12: The probability of transitioning to destination motion conditioned on the source motion. For ASE, there was no random trajectory matched to source motion in three cases, and the corresponding columns of the heatmap are left empty.

**Is FB-CPR learning more than imitating the motions in $\mathcal{M}$?** While the good coverage highlighted above and the good tracking performance shown in Sect. 4 illustrate that FB-CPR successfully ground its behaviors on the training motions, a remaining question is whether it has learned *more* than what is strictly in $\mathcal{M}$. In order to investigate this aspect we analyze the distribution of the closest EMD distance $EMD(\tau_z, m_z^\star)$ w.r.t. random policies $\pi_z$. Fig. 13 highlights the most of the behaviors in $(\pi_z)$ do not necessarily have a very tight connection with motions in the dataset. This is contrast with CALM and ASE, which have much smaller EMD distances, thus showing that they tend to use a larger part of the policy capacity to accurately reproduce motions rather than learning other behaviors.

## D.4 Qualitative Evaluation

### D.4.1 Human Evaluation

In most of reward-based tasks, the reward function is under-specified and different policies may achieve good performance while having different levels of *human-likeness*. In the worst case, the agent can learn to *hack* the reward function and maximize performance while performing very unnatural behaviors. On the other hand, in some cases, more human-like policies may not be "optimal". Similarly, in goal-based tasks, different policies may achieve similar success rate and proximity, while expressing very different behaviors.

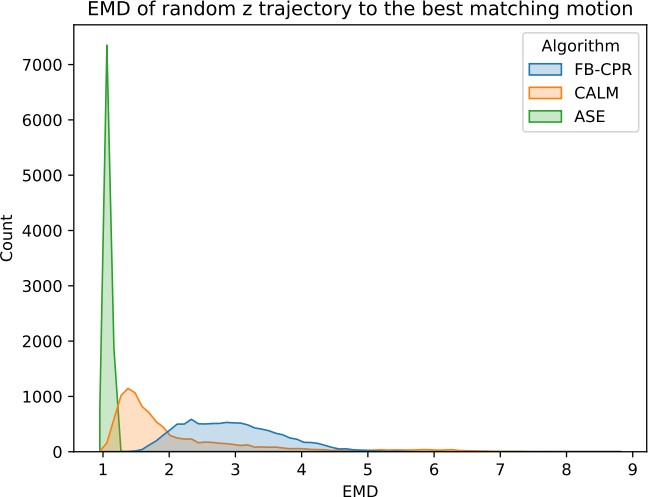

Figure 13: Histogram of the values of distance of trajectories generated from random $z$ to the best matching motion from the training dataset.

In this section, we complement the quantitative analysis in Sect. 4 with a qualitative evaluation assessing whether FB-CPR is able to express more "human-like" behaviors, similar to what is done in (Hansen et al., 2024a). For this purpose, we enroll human raters to compare TD3 and FB-CPR policies over 45 reward and 50 goal tasks. Similar to the protocol in Sect. 4, for each single reward or goal task, we train three single-task TD3 agents with different random seeds. We then compare the performance of the TD3 agent with the best metric against the zero-shot policy of FB-CPR.

We generate videos of the two agents for each task. Each pair of matching videos is presented to 50 human raters, who fill the forms presented on Fig. 14. The position of the videos is randomized and the type of the agent on a video is not disclosed to the raters.

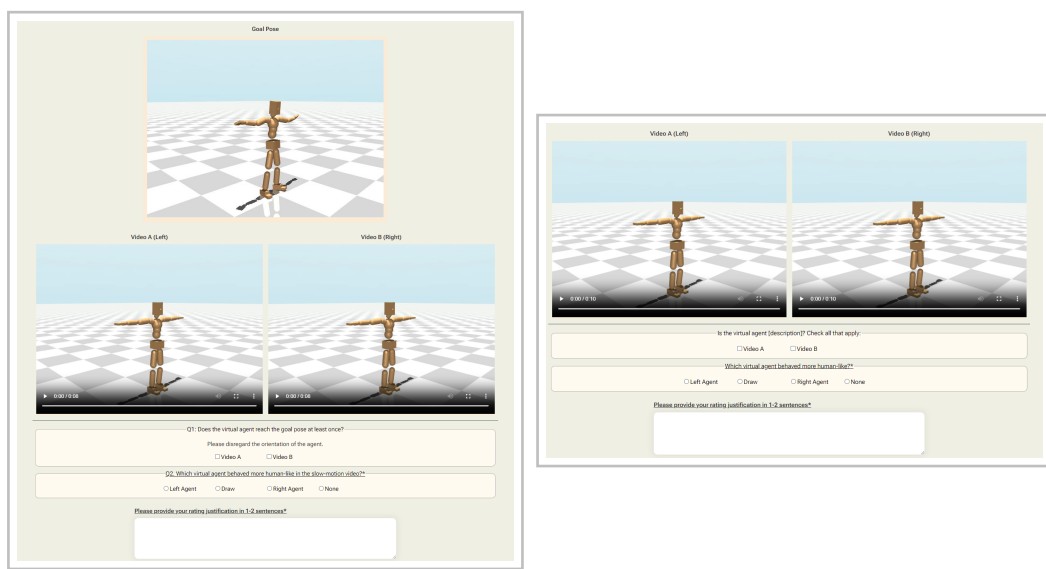

Figure 14: The online forms presented to the human raters to evaluate human-likeness for goal and reward tasks.

| Task | TD3 | ORACLE MPPI | Normalized | DIFFUSER | Normalized | ASE | Normalized | FB-CPR | Normalized |
|---|---|---|---|---|---|---|---|---|---|
| move-ego-0-2-raisearms-l-l | 191.13 | 168.22 | 0.88 | 148.10 (0.47) | 0.77 (0.00) | 145.78 (7.59) | 0.76 (0.04) | 145.59 (4.38) | 0.76 (0.02) |
| move-ego-0-2-raisearms-l-m | 174.97 | 194.84 | 1.11 | 125.14 (2.16) | 0.72 (0.01) | 109.36 (30.34) | 0.63 (0.17) | 143.90 (7.09) | 0.82 (0.04) |
| move-ego-0-2-raisearms-l-h | 194.72 | 114.30 | 0.59 | 103.11 (1.22) | 0.53 (0.01) | 129.21 (31.41) | 0.66 (0.16) | 123.14 (15.90) | 0.63 (0.08) |
| move-ego-0-2-raisearms-m-l | 179.42 | 199.26 | 1.11 | 124.31 (4.28) | 0.69 (0.02) | 125.39 (5.79) | 0.70 (0.03) | 136.74 (2.40) | 0.76 (0.01) |
| move-ego-0-2-raisearms-m-m | 178.42 | 155.28 | 0.87 | 121.55 (3.97) | 0.68 (0.02) | 60.19 (24.89) | 0.34 (0.14) | 139.19 (18.63) | 0.78 (0.10) |
| move-ego-0-2-raisearms-m-h | 179.02 | 129.99 | 0.73 | 116.50 (3.88) | 0.65 (0.02) | 123.84 (6.10) | 0.69 (0.03) | 128.15 (0.86) | 0.72 (0.00) |
| move-ego-0-2-raisearms-h-l | 191.00 | 115.25 | 0.60 | 101.58 (2.72) | 0.53 (0.01) | 85.89 (7.09) | 0.45 (0.04) | 111.92 (1.20) | 0.59 (0.01) |
| move-ego-0-2-raisearms-h-m | 175.72 | 130.86 | 0.74 | 113.81 (3.34) | 0.65 (0.02) | 121.19 (4.20) | 0.69 (0.02) | 128.10 (0.78) | 0.73 (0.00) |
| move-ego-0-2-raisearms-h-h | 165.19 | 112.35 | 0.68 | 102.09 (3.56) | 0.62 (0.02) | 133.96 (14.35) | 0.81 (0.09) | 143.83 (14.21) | 0.87 (0.09) |
| Average | 181.06 | 146.70 | 0.81 | 117.36 | 0.65 | 114.98 | 0.64 | 133.40 | 0.74 |
| Median | 179.02 | 130.86 | 0.74 | 116.50 | 0.65 | 123.84 | 0.69 | 136.74 | 0.76 |

Table 25: Average return for each task in the composite reward evaluation. These tasks combine between locomotion and arm-raising behaviors

We gather two subjective metrics: *success*, and *human-likeness*. For success, we ask the rater to evaluate whether the presented behavior is actually achieving the desired objective. For goal-based task, the objective is directly illustrated as the target pose, while for reward functions it is a text formulated in natural language which replaces the [description] placeholder in the template shown in Fig. 14 (e.g., for the task "raisearms-l-h" we generate text "standing with left hand low (at hip height) and right hand high (above head)"). For human-likeness, the rater has to choose among four options where they can express preference for either of the two behaviors, or both (a draw), or none of them. We then compute success rate and average human-likeness by taking the ratio between the positive answer and the total number of replies. The FB-CPR is considered more human like than TD3 in the large majority of cases. FB-CPR is sometimes assessed as human-like by raters, even in tasks when they consider it failed completing the task. Interestingly, while the human-likeness of FB-CPR may come at the cost of lower reward scores, it does not affect the perceived success in accomplishing the assigned goal tasks and FB-CPR has better success rate than TD3 for those tasks.

More in detail, per-task success rate scores are presented in Fig. 15 and Fig. 16.

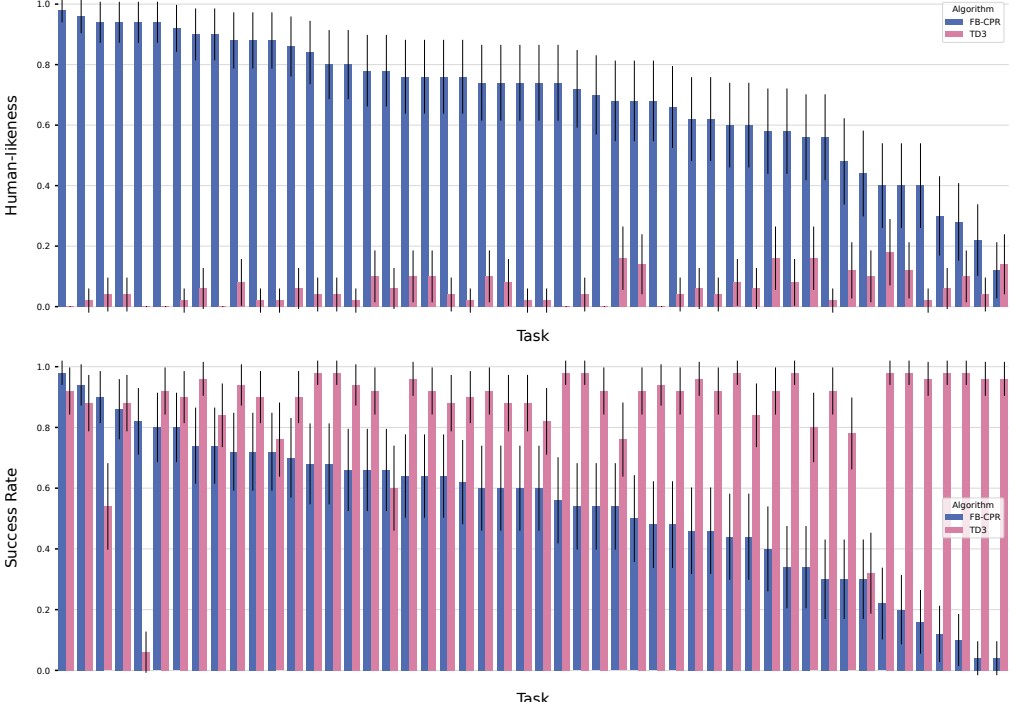

Figure 15: Human-likeness and success rate scores of algorithms per goal task sorted by FB-CPR performance.

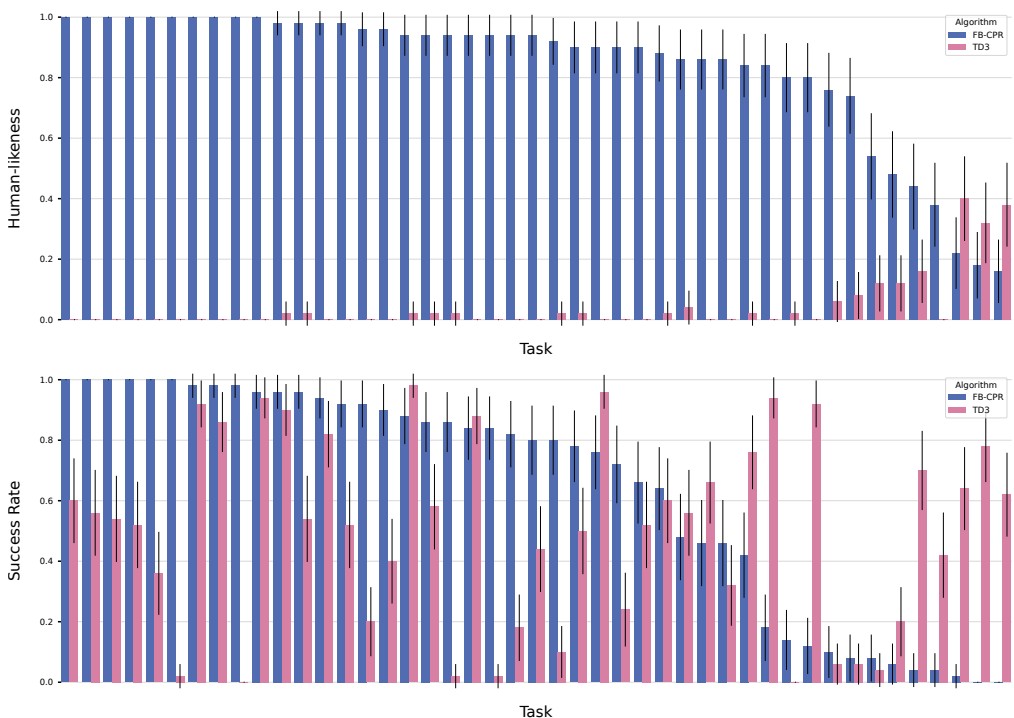

Figure 16: Human-likeness and success rate scores of algorithms per reward task sorted by FB-CPR performance.

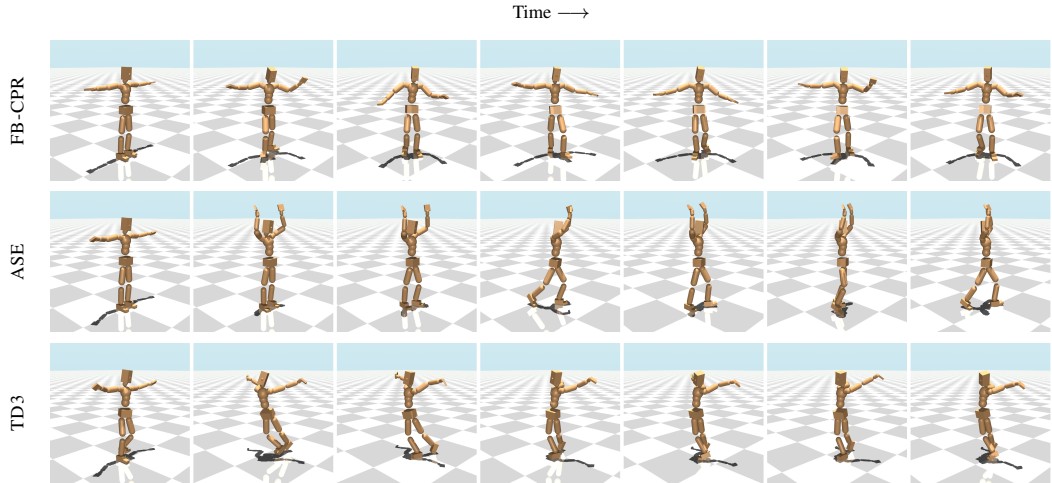

Figure 17: Example of combination of locomotion and arm raising tasks (`move-ego-0-2-raisearms-m-m`). Our FB-CPR (top) agent produces natural human motions while TD3 (bottom) learns high-performing but unnatural behaviors. ASE (middle) has a natural behavior but it is not correctly aligned with the tasks (arms are in the high position not medium).

### D.4.2 REWARD-BASED TASKS

We provide a further investigation of the performance of our FB-CPR agent on tasks that are i) a combination of tasks used for the main evaluation; and ii) highly under-specified.

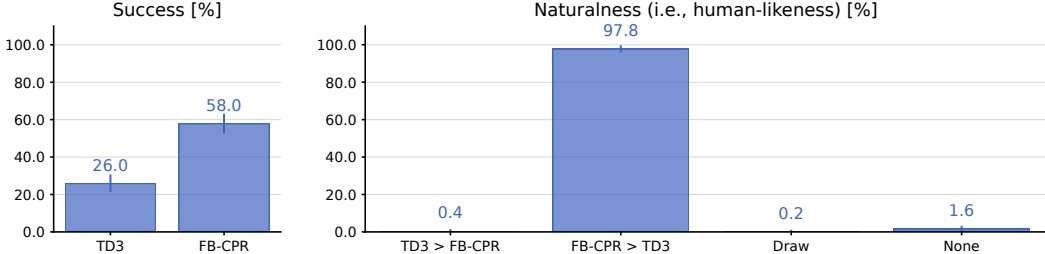

Figure 18: Human-evaluation on locomotion combined with arm raising. Left figure reports the percentage of times a behavior solved a reward-based task (tasks are independently evaluated). Right figure reports the score for human-likeness by direct comparison of the two algorithms.

The objective *i)* is to evaluate the ability of FB-CPR of composing behaviors. We thus created a new category of reward-based tasks by combining locomotion and arm-raising tasks. Specifically, we pair the medium-speed forward locomotion task (with an angle of zero and speed of 2) with all possible arm-raising tasks. Since these two types of tasks have conflicting objectives – locomotion requires movement, while arm-raising rewards stillness – we define a composite reward function that balances the two. This is achieved by taking a weighted average of the individual task rewards, where the weighting varies depending on the specific task combination. Tab. 25 reports the performance of the algorithms on these "combined" tasks. We can see that FB-CPR is able to achieve 74% of the performance of TD3 trained on each individual task. Despite the higher performance, even in this case, TD3 generates unnatural behaviors. The higher quality of FB-CPR is evident in Fig. 17 where we report a few frames of an episode for the task `move-ego-0-2-raisearms-m-m`. Similarly, almost the totality (about 98%) of human evaluators rated FB-CPR as more natural than TD3 on these tasks.

The objective of *ii)* is to evaluate the ability of our model to solve task with a human-like bias. To show this, we designed a few reward functions inspired by the way human person would describe a task.

**Run.** The simplest way to describe running is "move with high speed". Let $v_x$ and $v_y$ the horizontal velocities of the center of mass at the pelvis joint. Then, we define the reward for the task $\text{RUN}_{\text{eq}}$ as

$$r(s') = \mathbb{I}(v_x^2 + v_y^2 > 2)$$

**Walking with left hand up.** This task has two component: walking requires moving with low speed; raising the hand means having the hand z-coordinate above a certain threshold. Then, we define the reward for the task $\text{WALK-LAM}_{\text{eq}}$ as

$$r(s') = \mathbb{I}\left[1 < (v_x^2 + v_y^2) < 1.5\right] \cdot \mathbb{I}\left[z_{\text{left wrist}} > 1.2\right]$$

**Standing with right foot up.** This is the most complex task. We define standing at being in upright position with the head z-coordinate above a certain threshold and zero velocity. Similar to before, we ask the right ankle to be above a certain threshold. Then, we define the reward for the tasks $\text{STAND-RTM}_{\text{eq}}$ ($\beta = 0.5$) and $\text{STAND-RTH}_{\text{eq}}$ ($\beta = 1.2$) as

$$r(s') = \mathbb{I}\left[\text{up} > 0.9\right] \cdot \mathbb{I}\left[z_{\text{head}} > 1.4\right] \cdot \exp\left(-\sqrt{v_x^2 + v_y^2}\right) \cdot \mathbb{I}\left[z_{\text{right ankle}} > \beta\right]$$

It is evident to any expert in Reinforcement Learning (RL) that the reward functions in question are not optimal for learning from scratch. These reward functions are too vague, and a traditional RL algorithm would likely derive a high-performing policy that deviates significantly from the natural "behavioral" biases. For instance, with TD3, we observe completely unnatural behaviors. In stark contrast, FB-CPR manages to address the tasks in a manner that closely resembles human behavior (refer to Fig. 19). Intriguingly, FB-CPR appears to identify the "simplest" policy necessary to solve a task. It effectively distinguishes between two different policies, $\text{STAND-RTM}_{\text{eq}}$ and $\text{STAND-RTH}_{\text{eq}}$, even though the policy designed for the higher task would suffice for the medium

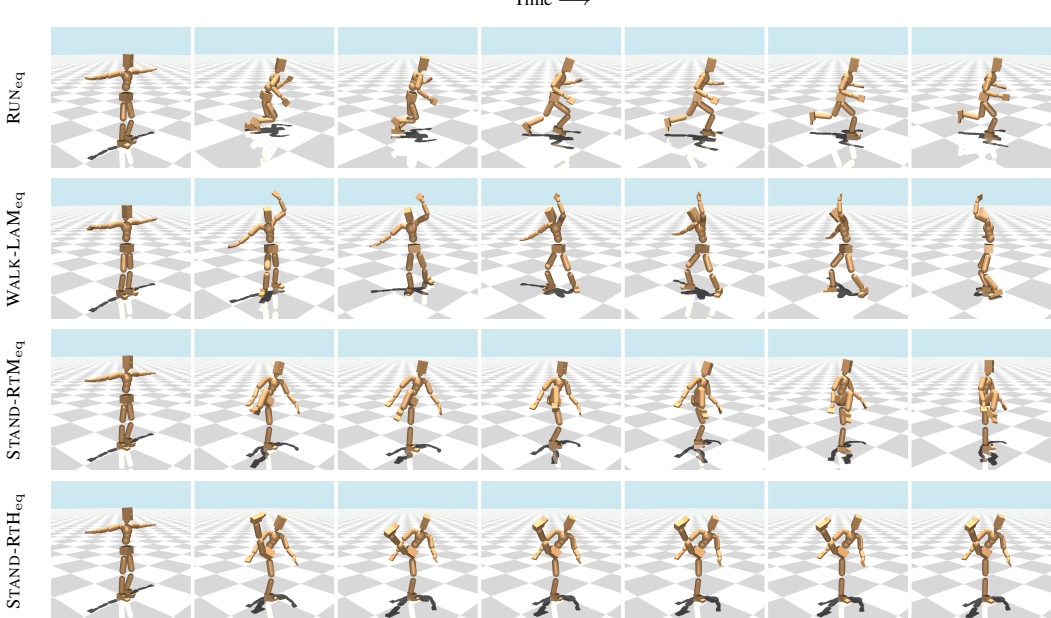

Figure 19: Example of behaviors inferred by FB-CPR from under-specified reward equations.

task, provided that the foot remains above a certain threshold. It is also evident the data bias. For example, we do not specify the direction of movement in run, just the high speed. FB-CPR recovers a perfect forward movement probably because the majority of run motions in $\mathcal{M}$ show this behavior. ASE is not able to solve all the tasks.

# E  ABLATIONS ON BIPEDAL WALKER

| Method | Data | Reward *Return* | Demonstration *Return* | Goal *Proximity* |
|---|---|---|---|---|
| FB | RND | $0.52 \pm 0.02$ | $0.43 \pm 0.02$ | $127.38 \pm 20.51$ |
| FB | RND+$\mathcal{M}_{\text{TRAIN}}$ | $0.60 \pm 0.03$ | $0.56 \pm 0.03$ | $211.46 \pm 17.78$ |
| FB+AWAC | $\mathcal{M}_{\text{TRAIN}}$ | $0.51 \pm 0.02$ | $0.54 \pm 0.02$ | $\mathit{279.90 \pm 44.07}$ |
| FB+AWAC | RND+$\mathcal{M}_{\text{TRAIN}}$ | $0.42 \pm 0.03$ | $0.43 \pm 0.05$ | $249.72 \pm 23.92$ |
| FB Online | None | $0.19 \pm 0.03$ | $0.19 \pm 0.02$ | $120.51 \pm 10.83$ |
| FB-CPR | $\mathcal{M}_{\text{TRAIN}}$ | $\mathit{0.71 \pm 0.02}$ | $\mathit{0.75 \pm 0.01}$ | $\mathbf{297.17 \pm 52.14}$ |
| FB-MPR | $\mathcal{M}_{\text{TRAIN}}$ | $\mathbf{0.77 \pm 0.02}$ | $\mathbf{0.78 \pm 0.01}$ | $258.66 \pm 43.89$ |

Table 26: Mean and standard deviation of performance with different prompts. Averaged over 10 random seeds. Higher is better. Normalized returns are normalized w.r.t expert TD3 policy in the same, rewarded task. RND data is generated by RND policy (Burda et al., 2019), while $\mathcal{M}_{\text{TRAIN}}$ data was generated by rolling out TD3 policies trained for each task separately.

We conduct an ablation study in the Walker domain of dm_control (Tunyasuvunakool et al., 2020) to better understand the value of combining FB with a conditional policy regularization and online training.

**Tasks.** For this environment only a handful of tasks have been considered in the literature (Laskin et al., 2021). In order to have a more significant analysis, we have developed a broader set of tasks. We consider three categories of tasks: **run**, **spin**, **crawl**. In each category, we parameterize *speed* (or angular momentum for spin) and *direction*. For instance, walker_crawl-{bw}-{1.5} refers to a task where the agent receives positive reward by remaining below a certain height while moving backward at speed 1.5. By combining category, speed, and direction, we define 90 tasks. We also create a set of 147 poses by performing a grid sweep over different joint positions and by training TD3 on each pose to prune unstable poses where TD3 does not reach a satisfactory performance.

**Data.** We select a subset of 48 reward-based tasks and for each of them we a TD3 policy to obtain 50 *expert* trajectories that we add to dataset $\mathcal{M}_{\text{TRAIN}}^{\text{demo}}$. We also run TD3 policies for a subset of 122 goals, while using the same 122 states as initial states, thus leading to a total of 14884 goal-based trajectories that are added to $\mathcal{M}_{\text{TRAIN}}^{\text{goal}}$. We then build $\mathcal{M}_{\text{TRAIN}} = \mathcal{M}_{\text{TRAIN}}^{\text{demo}} \cup \mathcal{M}_{\text{TRAIN}}^{\text{goal}}$, which contains demonstrations for a mix of reward-based and goal-reaching policies. For algorithms trained offline, we use either data generated by random network distillation (RND) (Burda et al., 2019)[14] or combining RND with $\mathcal{M}_{\text{TRAIN}}$. The $\mathcal{M}_{\text{TRAIN}}$ dataset contains 17,284 rollouts and 1,333,717 transitions[15], while the "RND" dataset contains 5000 episodes of 100 transitions for a total of 5,000,000 transitions.

**Evaluation.** For reward-based evaluation, we use the 42 tasks that were not used to build the demonstration dataset. For imitation learning, we consider the same 42 tasks and only 1 demonstration is provided. For goal-based evaluation, we use the 25 goals not considered for data generation.

**Baselines.** For ablation, we compare FB-CPR to the original FB algorithm (Touati et al., 2023) trained offline, offline FB with advantage-weighted actor critic (AWAC) (Cetin et al., 2024b), FB trained online, and FB-CPR with an unconditional discriminator (*i.e* discriminator depends solely on the state), that we refer to as FB-MPR (FB with marginal policy regularization).

**Results.** Table 26 shows the results for each evaluation category averaged over 10 seeds. For reward-based and imitation learning evaluation, we compute the ratio between each algorithm and the TD3/expert's performance for each task and then average it. For goal-reaching evaluation, we report the average proximity. We first notice that training FB online without access to any demonstration or unsupervised dataset leads to the worst performance among all algorithms. This suggests that FB representations collapse due to the lack of useful samples and, in turn, the lack of a good representation prevents the algorithm from performing a good exploration. Offline FB with only RND data achieves a good performance coherently with previous results reported in the literature.

---

[14]For walker, RND is successful in generating a dataset with good coverage given the low dimensionality of the state-action space. In humanoid, this would not be possible.

[15]Notice that goal-based trajectories have different lengths as episodes are truncated upon reaching the goal.

This confirms that once provided with a dataset with good coverage, the unsupervised RL training of FB is capable of learning a wide range of policies, including some with good performance on downstream tasks. Adding demonstration samples to RND further improves the performance of FB by 15% for reward-based tasks, 30% for imitation learning, and by 60% for goal-reaching. This shows that a carefully curated mix of covering samples and demonstrations can bias FB offline training towards learning behaviors that are closer to the data and improve the downstream performance. Nonetheless, the gap to FB-CPR remains significant, suggesting that regularizing the policy learning more explicitly is beneficial. Interestingly, behavior cloning regularization used in FB-AWAC does not significantly improve the performance of FB. When trained on $\mathcal{M}_{\text{TRAIN}}$, FB-AWAC significantly improves in goal-based problems, but in reward and imitation learning it is only able to match the performance of FB with RND. Mixing the two datasets only marginally improves the goal-based performance, while degrading other metrics. Overall FB with online training with a policy regularization emerges as the best strategy across all tasks. Interestingly, the version with unconditional discriminator achieves better performance for reward and demonstration tasks, while it is significantly worse for goal reaching problems, where FB-CPR is best. We conjecture that this result is due to the fact that the dataset $\mathcal{M}$ is well curated, since trajectories are generated by optimal policies and they cover close regions of the state space, whereas in the humanoid case, $\mathcal{M}$ is made of real data where different motions can be very distinct from each other and are very heterogeneous in nature and length. While in the former case just reaching similar states as in $\mathcal{M}$ is sufficient to have a good regularization, in the latter a stronger adherence to the motions is needed.

## F    ABLATIONS ON ANTMAZE

We conduct an ablation study in the antmaze domains from the recently introduced goal-conditioned RL benchmark (Park et al., 2024a) to better understand the value of combining FB with a conditional policy regularization and online training. Antmaze domains involve controlling a quadrupedal Ant agent with 8 degrees of freedom.

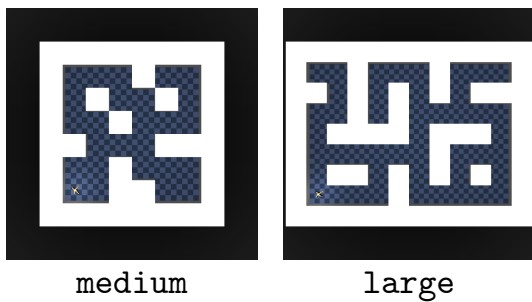

medium             large

Figure 20: Layout of antmaze-medium and antmaze-large domains from (Park et al., 2024a)

**Data.**    We use *stitch datasets* for antmaze domains provided in Park et al. (2024a), which consist of short goal-reaching demonstrations trajectories. These datasets are designed to challenge agent's stitching ability over subgoals to complete the downstream tasks.

**Evaluation.**    We use the same evaluation protocol employed in Park et al. (2024a). Each domain has 5 downstream tasks. The aim of these tasks is to control the agent to reach a target $(x, y)$ location in the given maze. The task is specified by the full state, but only the $(x, y)$ coordinates are set to the target goal, while the remaining state components are randomly generated. For each goal, we evaluate the agent using 50 episodes.

**Results.**    We present a comparison of three methods in Table 27: online FB trained solely on environment interactions, offline FB with advantage weighting (AWAC) using the offline stitch datasets, and online FB-CPR that utilizes stitch datasets for policy regularization. We report both success rate and proximity (averaged distance to the goal) averaged across 3 models trained with different random seeds. Online FB fails to reach any test goals, achieving zero success rate due to the lack of exploration. In contrast, FB-AWAC achieves decent performance, which is indeed competitive

| Algorithm | Antmaze-medium | | Antmaze-large | |
|---|---|---|---|---|
| | Proximity ($\downarrow$) | Success ($\uparrow$) | Proximity ($\downarrow$) | Success ($\uparrow$) |
| (online) FB | 19.71 (0.11) | 0 (0) | 25.74 (0.05) | 0 (0) |
| (offline) FB-AWAC | 6.70 (0.4) | 0.67 (0.08) | 18.00 (1.54) | 0.28 (0.05) |
| (online) FB-CPR | 3.19 (0.13) | 0.90 (0.1) | 7.97 (0.39) | 0.53 (0.08) |

Table 27: Performance of different algorithms in Antmaze domains (medium and large mazes). We report mean and standard deviation of the performance over three random seeds.

with the non-hierarchical offline goal-conditioned RL algorithms reported in Park et al. (2024a). Finally, FB-CPR achieves the strongest performance and it outperforms the other FB-variants by a significant margin, both in success rate and proximity.

## G  UNDERSTANDING THE BEHAVIORAL LATENT SPACE

In this section, we summarize results from a qualitative investigation aimed at better understanding the structure of the latent space learned by FB-CPR. We recall that the latent space $Z$ works at the same time as a state embedding through $B(s)$, a trajectory embedding through $\text{ER}_{\text{FB}}$, and a policy embedding through $\pi_z$.

### G.1  DIMENSIONALITY REDUCTION OF THE BEHAVIORAL LATENT SPACE

We investigate the structure of the latent space learned through FB-CPR by performing dimensionality reduction via UMAP (McInnes et al., 2018) on the embeddings $z$ coming from two sources: 1) motion embeddings using $\text{ER}_{\text{FB}}$ and 2) reward embeddings computed via weighted regression. In order to see meaningful structure in the latent space we decide to classify various motions into five categories: jumping, running, walking, crawling, and motions containing headstands or cartwheels.

Given these categories we construct a dataset of motions by first choosing a single representative motion for each category and subsequently searching for other motions that are sufficiently close to the reference motion as measured be the Earth Mover's Distance (EMD). We chose all motions where the EMD fell below some threshold that was chosen by visual inspection. With this dataset of motions $\tau_i = \{x_1, \ldots, x_n\}$ of length $n$ we embed the center most subsequence, i.e., $\tau_i^\perp = \{x_i : i \in [\lfloor n/2 \rfloor - 4, \lfloor n/2 \rfloor + 4]\}$ using $\text{ER}_{\text{FB}}$. The center subsequence was chosen as it was most representative of the category whereas other locations usually had more "set up" in preparation for the motion, e.g., walking before performing a headstand.

Reward embeddings were chosen from Appendix C.3.1 to be representative of the motion category. Specifically, we use the following reward functions for each class:

1. Jumping: `smpl_jump-2`
2. Running: `smpl_move-ego-90-4`
3. Walking: `smpl_move-ego-90-2`
4. Crawling: `smpl_crawl-0.5-2-d`
5. Headstand: `smpl_headstand`

Figure 21 depicts both motion and reward embeddings along with illustrative visualizations for each class of behaviors. Interestingly, the motions involving similar activities are accurately clustered in similar regions through the embedding process. Furthermore, even the reward tasks are embedded within the clusters of motions they are closely connected to. This reveals that the training of FB-CPR leads to learning representations that effectively align motions and rewards in the same latent space.

### G.2  BEHAVIOR INTERPOLATION

While the analysis in App. G.1 shows that the latent space effectively clusters behaviors that are semantically similar, we would like to further understand whether it also supports meaningful interpo-

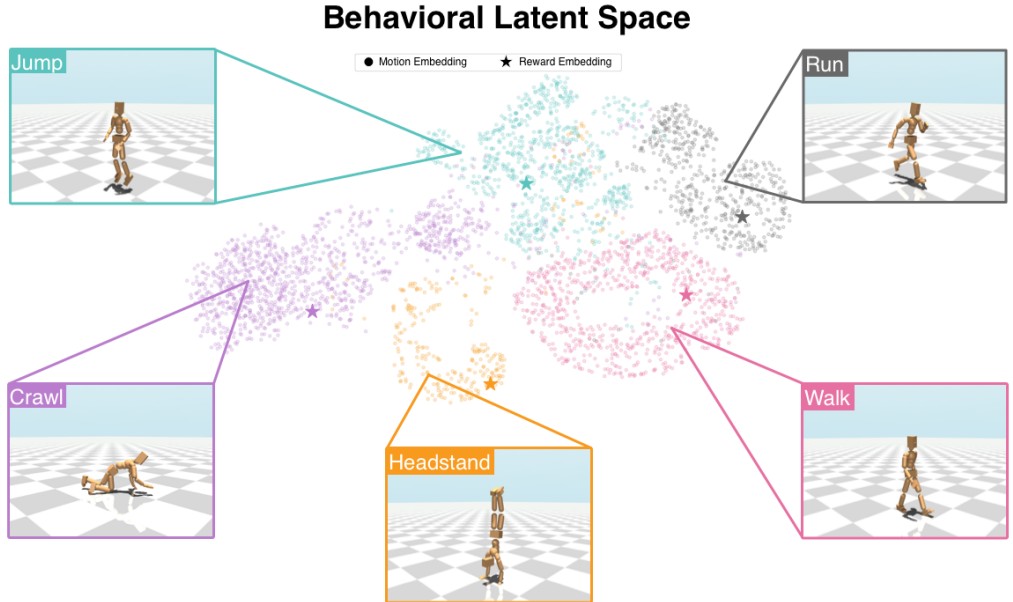

Figure 21: UMAP (McInnes et al., 2018) plot of the latent space of FB-CPR with both motion embeddings (circle) and reward embeddings (star). We can see that reward functions are projected to clusters that correspond with motions of the same class of behaviors.

lation between any two points. We have first selected a few reward functions that are underspecified enough that can be combined together (e.g., "run" and "raise left hand" tasks could be composed into "run with left hand up"). We make this choice to investigate whether interpolating between the behaviors associated to each reward function would produce a resulting behavior that is the result of the composition of the two original behaviors. More precisely, given the reward functions $r_1$ and $r_2$, we first perform inference to compute $z_1$ and $z_2$ and we then define $z_\alpha = \alpha z_1 + (1 - \alpha)z_2$ and we let vary $\alpha$ in $[0, 1]$. Refer to the supplmentary material for videos illustrating the behaviors that we obtained through this protocol for a few pairs of reward functions. In general, not only we observed a smooth variation of the behavior as $\alpha$ changes, but the interpolated policies often combine the two original tasks, obtaining more complex behaviors such as running with left hand up or moving and spinning at the same time.

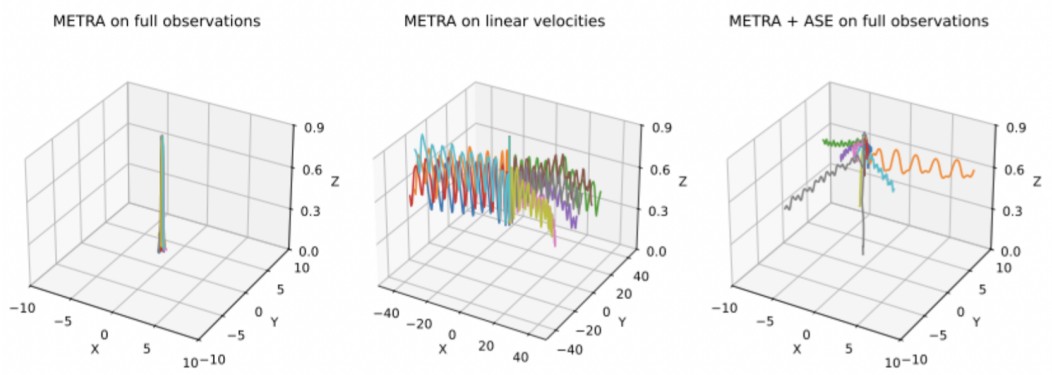

Figure 22: Rollouts of policies learned by different variants of METRA on Humanoid. Each line corresponds to a trajectory in $(x, y, z)$ space generated by a policy $\pi_z$ with $z$ uniformly sampled from the unit sphere. *(left)* The original METRA algorithm trained from scratch (no unlabeled data) with representation $\phi$ taking as input the full observation vector. *(middle)* The original METRA algorithm trained from scratch (no unlabeled data) with representation $\phi$ taking as input only the linear velocities of the robot's pelvis along the x,y,z axes. *(right)* The ASE algorithm trained within the same setting as in Table 1 but with METRA replacing DIAYN as the skill discovery component.

| Algorithm | Reward (↑) | Goal | | Tracking - EMD (↓) | | Tracking - Success (↑) | |
| --- | --- | --- | --- | --- | --- | --- | --- |
| | | Proximity (↑) | Success (↑) | Train | Test | Train | Test |
| METRA | 6.37 (1.04) | 0 (0) | 0 (0) | 9.92 (0.13) | 9.95 (0.18) | 0 (0) | 0 (0) |
| METRA-ASE | 37.98 (6.61) | 0.30 (0.01) | 0.24 (0.05) | 2.11 (0.07) | 2.12 (0.05) | 0.54 (0.04) | 0.56 (0.06) |
| DIAYN-ASE | 105.73 (3.82) | 0.46 (0.37) | 0.22 (0.37) | 2.00 (0.02) | 1.99 (0.02) | 0.37 (0.02) | 0.40 (0.03) |

Table 28: Performance of METRA (Park et al., 2024c) and ASE (Peng et al., 2022) with METRA replacing DIAYN as the skill discovery component in the same setting as Table 1. We also include the original ASE algorithm from such table (called DIAYN-ASE) to ease comparison.

## H   COMPARISON TO UNSUPERVISED SKILL DISCOVERY METHODS

In FB-CPR, we leverage unlabeled datasets to scale unsupervised RL to high-dimensional problems like Humanoid control. The main conjecture is that unlabeled datasets provide a good inductive bias towards the manifold of behaviors of interest (e.g., those that are human-like), and that this bias is crucial to avoid the "curse of dimensionality" suffered when learning over the (probably intractable) space of all expressible behaviors. On the other hand, there is a vast literature on Unsupervised Skill Discovery (USD) which focuses on learning over such full space of behaviors while providing inductive biases through notions of, e.g., curiosity (e.g., Pathak et al., 2017; Rajeswar et al., 2023), coverage (e.g., Burda et al., 2019; Liu & Abbeel, 2021), or diversity (e.g., Gregor et al., 2016; Eysenbach et al., 2019; Sharma et al., 2020; Park et al., 2022; 2024c).

In this section, we compare to METRA (Park et al., 2024c), the current state-of-the-art USD method, and show that it fails on our high-dimensional Humanoid control problem unless given extra inductive biases through unlabeled data or by restricting the set of variables on which to focus the discovery of new behaviors. Given that METRA remains, to our knowledge, the only USD method to discover useful behaviors in high-dimensional problems like humanoid and quadruped control, we conjecture that this "negative" result also applies to all existing USD methods.

**Implementation and parameters**. We implemented METRA following the original code of Park et al. (2024c), with the only difference that we replaced SAC with TD3 as RL optimizer since we used the latter for all algorithms considered in this paper. We also follow Park et al. (2024c) to tune the hyperparameters related to the representation learning component, while for TD3 we use the same parameters and network architectures we found to work well across all baselines tested in this paper. We found the dimension $d$ of the latent space to be the most important parameter, and we found $d = 16$ to work best after searching over 2,4,8,16,32,64,128,256. All parameters are summarized in the following table.

Table 29: Hyperparameters used for METRA pretraining.

| Hyperparameter | Value |
|---|---|
| General training parameters | See Tab. 3 |
| General prioritization parameters | See Tab. 4 |
| $z$ update frequency during rollouts | once every 150 steps |
| $z$ dimension $d$ | 16 |
| actor network | third column of Tab. 6, output dim = action dim |
| critic networks | second column of Tab. 6, output dim 1 |
| $\phi$ encoder network | fourth column of Tab. 5, output dim 16, 2 hidden layers |
| Learning rate for actor | $10^{-4}$ |
| Learning rate for critic | $10^{-4}$ |
| Learning rate for $\phi$ | $10^{-6}$ |
| Constraint slack $\epsilon$ | $10^{-3}$ |
| Initial Lagrange multiplier $\lambda$ | 30 |
| $z$ distribution $\nu$ | uniform on unit sphere |
| Probability of relabeling zs | 0.8 |
| Polyak coefficient for target network update | 0.005 |
| Actor penalty coefficient | 0.5 |
| Critic penalty coefficient | 0.5 |

**Inference methods**. For goal-based inference, we follow the zero-shot scheme proposed by Park et al. (2024c): when given a goal state $g$ to reach from state $s$, we set $z = (\phi(g) - \phi(s))/\|\phi(g) - \phi(s)\|_2$. Similarly, for tracking we set $z_t = (\phi(g_{t+1}) - \phi(s_t))/\|\phi(g_{t+1}) - \phi(s_t)\|_2$ at each step $t$ of the episode, where $g_{t+1}$ is the next state in the trajectory to be tracked, while $s_t$ is current agent state. Finally, for reward inference, given a dataset of transitions $(s, s', r)$ sampled from the train buffer and labeled with the corresponding reward $r$, we infer $z$ through linear regression on top of features $\phi(s') - \phi(s)$. This is motivated by the fact that METRA's actor is pretrained to maximize a self-supervised reward function given by $r(s, s', z) := (\phi(s') - \phi(s))^T z$. Notice, however, that we do not expect this to work well since such a reward, up to discounting, yields a telescopic sum which eventually makes the agent care only about the reward received at the end of an episode instead of the cumulative sum. Thus we report its performance for completeness.

**Results**. We test METRA in the same setting as Table 1. The results are reported in the first row of Table 28, where we find that METRA achieves near zero performance in all tasks. After a deeper investigation, we found that in all runs, and with all hyperparameters we tested, the agent simply learned to fall on the floor and remain still in different positions, as shown in Figure 22 (left). Interestingly, this happens despite all the objectives, and in particular the "diversity loss" for representation learning, are well optimized during pre-training. This is due to the fact that, from the agent perspective, lying still on the floor in different positions can be regarded as displaying diverse behaviors, and no extra inductive bias would push the agent to learn more complicated skills (e.g., locomotion ones). On the other hand, we believe that METRA manages to learn few of such skills in the Humanoid experiments of Park et al. (2024c) given that it is pretrained on pixel-based observations (instead of proprioception) with a color map on the ground and very small dimension of the latent space ($d = 2$). This may provide an implicit inductive bias towards locomotion behaviors that make the robot move around the x,y coordinates, which are likely to be the observation variables that can be maximally spread out by the agent's controls. On the other hand, we do not have any such bias in our setup, where each joint has roughly the same "controllability" and the agent thus learns the simplest way to display diverse behaviors.

To verify this last conjecture, we retrained METRA with the same parameters except that we make the representation $\phi$ only a function of the linear velocities of the robot's pelvis along the three x,y,z directions. Intuitively, this should provide an inductive bias that makes the agent focus on controlling those variables alone, thus learning locomotion behaviors to move around the x,y,z space. This is confirmed in Figure 22 (middle), where we see that the learned skills do not collapse anymore but rather move around different directions of the space.

**METRA with ASE regularization**. Finally, we tried to combine METRA with the same policy regularization on top of unlabeled data as used by ASE. As we recall that ASE (Peng et al., 2022) combines a USD algorithm (DIAYN) with an unconditional policy regularization term, we simply replace DIAYN with METRA and keep all other components the same. The results are shown in Table 28, where we see that the ASE regularization improves the performance of METRA significantly on goal reaching and tracking. Moreover, METRA-ASE achieves competitive performance w.r.t. the original DIAYN-based ASE, improving its success rate in those tasks. Both DIAYN-ASE

and METRA-ASE perform, however, significantly worse than FB-CPR. Finally, we note from Figure 22 (right) that METRA-ASE learns to navigate along different directions, though less far than plain METRA trained only on the pelvis' velocities. This is likely due to the regularization w.r.t. unlabeled data, which makes the agent focus on human-like behaviors, thus avoiding over-actuated movements that would be otherwise learned when naively trying to maximize controls of a subset of the observation variables.

