# OpenReview forum: "Zero-Shot Whole-Body Humanoid Control via Behavioral Foundation Models"
_ICLR.cc/2025/Conference — ICLR 2025 Poster_

### Official Review · Reviewer_hx2C · 2024-10-20

**Soundness:** 3
**Presentation:** 2
**Contribution:** 3
**Rating:** 6
**Confidence:** 3

**Summary:**

This paper proposes an algorithm that pre-trains behavioral foundation models (BFMs) using unlabeled motion capture data for zero-shot generalization to humanoid control tasks. It combines the Forward-Backward (FB) representation with Conditional Policy Regularization (CPR) to solve tasks like motion tracking, goal-reaching, and reward optimization. FB-CPR outperforms existing unsupervised RL algorithms and model-based methods, achieving competitive results compared to task-specific models while also producing more human-like behavior.

**Post-Rebuttal Review**

Thanks for the detailed response. I have carefully read your rebuttal and it resolved most of my concerns. I will raise my score accordingly.

**Strengths:**

The authors address the challenging problem of generalizing to unseen tasks in humanoid whole-body control without the need for task-specific training. The paper provides a lot of details in the appendix, a well-designed and thorough experimental section. The experiments cover a wide range of scenarios, demonstrating the method's effectiveness across various tasks.

**Weaknesses:**

The problem definition is not very clear, making it difficult to understand at the beginning. The theoretical explanations about FB representation might be a little complicated for readers lacking corresponding background. Some explicit examples or pictures may help.

**Questions:**

* It is unclear how the advantages and differences of this method compare to previous approaches, such as AMP[1] and ASE[2], as well as to recent work (like Omnigrasp[3], MaskedMimic[4]), when applied to the demonstration data settings discussed in this paper.
* I want to know if it is possible to deploy the method on real robots, just like RobotMDM[5] (I feel that this method is difficult to have this scalability).
* While the paper focuses on motion capture datasets, how would FB-CPR perform when trained on more diverse or noisy datasets, such as uncurated video footage?


[1]Peng, Xue Bin, et al. "Amp: Adversarial motion priors for stylized physics-based character control." ACM Transactions on Graphics (ToG) 40.4 (2021): 1-20.

[2]Peng, Xue Bin, et al. "Ase: Large-scale reusable adversarial skill embeddings for physically simulated characters." ACM Transactions On Graphics (TOG) 41.4 (2022): 1-17.

[3]Luo, Zhengyi, et al. "Grasping diverse objects with simulated humanoids." arXiv preprint arXiv:2407.11385 (2024).

[4]Tessler, Chen, et al. "MaskedMimic: Unified Physics-Based Character Control Through Masked Motion Inpainting." arXiv preprint arXiv:2409.14393 (2024).

[5]SERIFI, AGON, and MARKUS GROSS. "Robot Motion Diffusion Model: Motion Generation for Robotic Characters." (2024).

---

> ### Author Response · Authors · 2024-11-25
> **Rebuttal**
>
> Thanks for your review and the additional references! Please find below our answers.
>
> > The problem definition is not very clear, making it difficult to understand at the beginning.
>
> We consider a setting where a model can be pre-trained from a dataset of unlabeled observation-only trajectories and online interaction with the environment. At test time, the resulting model should be able to solve different types of tasks, including trajectories tracking, goal reaching, and reward optimization. We will further clarify the setting informally in the introduction and formally in section 2.
>
> > The theoretical explanations about FB representation might be a little complicated ... some explicit examples or pictures may help
>
> Due to space constraints, in the main paper we have focused on the core aspects of the FB model and its losses to allow the reader to understand the key differences we introduced in the FB-CPR model. We will expand on more algorithmic aspects and theoretical properties in the supplementary material to make our contribution more self-contained including illustrations of the inference process.
>
> > It is unclear how the advantages and differences of this method compare to previous approaches, such as AMP[1] and ASE[2], as well as to recent work (like Omnigrasp[3], MaskedMimic[4]).
>
> AMP and ASE share a similar intuition that policy learning should be regularized by some additional imitation learning objectives. While AMP is designed for single-task problems, ASE aims at pre-training a behavioral foundation model. The main differences between ASE and FB-CPR are: 1) ASE employs an unconditional discriminator which encourages policies to generically cover the same states as in the behavior dataset. This does not guarantee that the resulting model can actually reproduce any of the motions in the dataset. On the other hand, the conditional-policy discriminator of FB-CPR encourages learning policies that can reproduce fragments of the trajectories shown in the dataset, which makes FB-CPR better at tracking. 2) ASE relies on a DIYAN objective to enforce diversity across policies, whereas FB-CPR leverages an FB loss component which favors policies that are optimal for some reward function. This leads FB-CPR to express policies that are better at reward optimization than ASE. Please refer to Table 1 for an extensive comparison. Omnigrasp is a version of PHC specifically designed for object manipulation and it is focused on tracking tasks. We report a comparison of FB-CPR with PHC in Table 1. Finally, MaskedMimic, which was released only a week before the submission, relies on a complex pipeline where first an imitation policy is learned (using a carefully crafted reward) and then it is distilled in a masked version of the policy to support different downstream use cases. Unlike FB-CPR the resulting model does not support reward inference and it rather relies on hand-defined finite-state automata (``goal-engineering’’) to solve more complex downstream tasks.
>
> > I want to know if it is possible to deploy the method on real robots, just like RobotMDM[5] (I feel that this method is difficult to have this scalability).
>
> FB-CPR does assume access to the environment through direct online interaction. While this is not desirable in real robotic applications, we could follow the standard sim2real protocol by first training FB-CPR in simulation and then deploy it and fine-tune it on an actual robot (the same approach is used in RobotMDM). The other assumption we have is that behavior data are expressed in the same embodiment as the agent we are training. In a robotic application, this could be obtained by data collection from the robot itself and/or from retargeting e.g., motion capture datasets. Regarding scaling, we do not anticipate any specific challenge since FB-CPR is already trained on a humanoid with 23 joints, whereas in RobotMDM a bipedal robot with 20 degrees of freedom is considered.
>
> > While the paper focuses on motion capture datasets, how would FB-CPR perform when trained on more diverse or noisy datasets, such as uncurated video footage?
>
> In the short time of the rebuttal, unfortunately we could not run additional experiments from video datasets. Nonetheless, we would like to point out that the AMASS dataset already contains motions with noise, recording and reconstruction artifacts, and in general they may not be realizable in the physics-based simulation we consider (i.e., there may not be any sequence of actions able to reproduce the same transitions). This is already a significant departure from the protocols often used in RL literature, where demonstrations datasets are generated by rolling out policies in the actual environment without any additional noise, thus avoiding any non-realizability issue.

---

> > ### Author Response · Authors · 2024-12-01
> >
> > Dear reviewer, we hope our rebuttal helped in resolving your concerns and we are wondering whether there is any additional point you would like us to clarify. Thanks!

---

### Official Review · Reviewer_Gh4R · 2024-10-27

**Soundness:** 3
**Presentation:** 4
**Contribution:** 3
**Rating:** 8
**Confidence:** 3

**Summary:**

The authors propose to augment the forward-backwards unsupervised RL framework with a policy regularization term that encourages covering the entire set of behaviors present in the training dataset. They do so by augmenting the FB loss with a discriminator that determines whether a state came from the dataset or from the policy. The authors provide extensive experiments on humanoid control problems showing that FB-CPR approaches the performance of policies trained on individual goals and outperforms other multi-task baselines.

**Strengths:**

- The authors present their method clearly and motivate the problem setting well.
- The proposed method is a simple addition to the standard FB framework.
- The proposed implementation allows steering the model to learn useful behaviors by modifying the latent distribution of skills used during training.

**Weaknesses:**

- The majority of the evaluations are conducted only on the humanoid environment. Although the number of experiments done in this environment is diverse, higher diversity in environments would make the paper even stronger.
- Although more “human like” behavior (more like the training dataset) might be desirable in a humanoid environment, could the regularization negatively affect performance where the majority of the data is very suboptimal?
- It feels like something like METRA [1], which is explicitly trained to span a diverse set of behaviors, is also a relevant baseline. It tackles a similar problem that the proposed regularization intends to solve: spanning a diverse set of useful behaviors.

**Questions:**

- On the humanoid experiments, why aren’t there more direct comparisons to FB without the proposed regularization? There are a few ablations in the appendix with direct comparison to FB, but this also feels relevant to the main experiments.
- How does the proposed regularization affect the nature of the embedding space? Could a nicely regularized latent space enable something like interpolations between skills etc.?

References:
[1] Seohong Park, Oleh Rybkin, and Sergey Levine. METRA: scalable unsupervised RL with metric-aware abstraction. In ICLR. OpenReview.net, 2024b.

---

> ### Author Response · Authors · 2024-11-25
> **Rebuttal**
>
> Thanks for your review! We addressed your questions in the general answer and in the following replies.
>
> > The majority of the evaluations are conducted only on the humanoid environment.
>
> We have now included experiments in the AntMaze domain from the recent OGBench benchmark. Please refer to the general answer and the revised paper for more details.
>
> > Although more “human like” behavior (more like the training dataset) might be desirable in a humanoid environment, could the regularization negatively affect performance where the majority of the data is very suboptimal?
>
> In general, the “behavior dataset” is intended to restrict the scope of unsupervised RL and focus it on tasks/policies that are somehow related to the data. While FB-CPR is indeed trying to learn policies that can reproduce segments of the demonstrations, the FB part of the loss is also pushing towards learning optimal policies for rewards in the span of the representation B. In the humanoid case, most motions are not generated by a stationary Markov policy optimizing a reward function. As such, most of them are heavily suboptimal for the reward functions that we consider in the reward-based evaluation. Nonetheless, FB-CPR manages to learn policies achieving satisfactory performance in most of the tasks, while retaining the human-like nature of the demonstrations.
>
> Furthermore, in the new AntMaze domain and in the original ablations performed in the bipedal walker domain (In App. E) we do not target any “qualitative” bias and the behavior datasets were only used to ground unsupervised RL. While the demonstrated behaviors may be optimal for some specific tasks, they are suboptimal for the reward functions used at test time. Also in this case, the regularization is effective in skewing the learning towards policies that are “similar” to the demonstrations, while achieving good performance in the reward-based tasks.
>
> > It feels like something like METRA [1], which is explicitly trained to span a diverse set of behaviors, is also a relevant baseline.
>
> Thanks for the suggestion! We have implemented METRA and some related variants. Please refer to the general rebuttal and App. H in the revised paper for further details.
>
> > On the humanoid experiments, why aren’t there more direct comparisons to FB without the proposed regularization?
>
> Please refer to the general answer and the additional experiments in App. D.2 in the revised paper. Unfortunately basic FB trained online achieves very poor performance as it does not collect useful samples and in turns it does not learn any effective behavior.
>
> > How does the proposed regularization affect the nature of the embedding space? Could a nicely regularized latent space enable something like interpolations between skills etc.?
>
> Thanks for the question and the suggestion! We have spent some time digging more into the structure of the embedding space and we have now convincing illustrations on how the latent space effectively clusters motions together while aligning them with reward embeddings. Furthermore, preliminary tests on task interpolations shows that the policy embedding varies the behavior smoothly and it is capable of composing tasks with different objectives (e.g., "spin" and "move" produces "spinning and moving", while "move" and "raise arm" produces "move while raising arm"). Please refer to revised paper and supplementary material, and the general answer for more details.

---

> > ### Author Response · Authors · 2024-12-01
> >
> > Dear reviewer, we hope our rebuttal helped in resolving your concerns and we are wondering whether there is any additional point you would like us to clarify. Thanks!

---

> > > ### Comment · Reviewer_Gh4R · 2024-12-03
> > >
> > > Thank you for the detailed response from the authors. The additional evaluations and results address my points and make the paper stronger (the interpolation videos are very interesting as well!). I raise my score to accept.

---

### Official Review · Reviewer_whMZ · 2024-10-31

**Soundness:** 3
**Presentation:** 3
**Contribution:** 2
**Rating:** 6
**Confidence:** 5

**Summary:**

This paper proposes FB-CPR, a variant of FB representations that regularize behaviors to offline trajectory data. Their main idea is to define a regularization reward with a GAN-style discriminator, and add this as a bonus to the original FB reward. Importantly, the authors condition this discriminator on an inferred $z$, making the regularization more targeted. They apply FB-CRP to Humanoid control, showing that it achieves better performance and naturality than other baselines.

**Strengths:**

* The paper is of high quality and well-written.
* To my knowledge, this is one of the few works that scale unsupervised RL up to Humanoid control.
* The proposed objective is clean and reasonable.
* The paper presents several ablation studies, which help understand the importance of the individual components of FB-CPR.

**Weaknesses:**

* One weakness is its limited novelty. The method is largely built upon the previous FB framework, and the only difference between the original FB paper and this work is the additional discriminator term for behavioral regularization. Having a data-regularization term in data-driven RL (i.e., BC, offline RL, motion priors, etc.) is a standard, well-established technique. While FB-CPR's conditioning of the discriminator on inferred $z$ is distinct from standard offline RL regularization techniques, I believe this alone doesn't constitute significant novelty, given how other skill-based offline unsupervised RL works (e.g., HILP (Park et al., 2024), FRE (Frans et al., 2024), etc.) employ similar $z$-inference techniques when applying behavioral regularization, though they don't use explicit discriminators.
* Another weakness is that the effectiveness of FB-CPR is only shown on Humanoid. While Humanoid control is indeed an important problem, it is unclear whether FB-CPR can generally be applicable to other environments, or if it is only effective for Humanoid.
* It seems the benefits of FB-CPR mostly come from behavioral regularization, and the contribution of the FB objective seems relatively marginal (Fig. 4, top right). While the authors show that the FB objective helps to some degree, given the significant complexity of the FB algorithm and its marginal effect on performance, I'm not entirely convinced that having the FB component is worthwhile.

Despite the above weaknesses, I believe the contributions of this work outweigh its weaknesses, and thus recommend weak acceptance.

**Questions:**

I'm curious why FB-CPR is only shown on Humanoid control. For example, could the same method be applied to manipulation environments (e.g., D4RL Kitchen or similar human demonstration-based manipulation datasets)?

---

> ### Author Response · Authors · 2024-11-25
> **Rebuttal**
>
> Thanks for your review! We have submitted a revised version with additional experiments (see general answer) and we address your questions below.
>
> > One weakness is its limited novelty.
>
> While we agree with the reviewer that combining a policy optimization loss with some imitation regularization is a fairly common principle in the RL literature, its instantiation into unsupervised RL is non-trivial and the specific algorithmic solution we propose with FB-CPR is novel and, more importantly, it is crucial to obtain the significant improvements reported in the paper compared to a wide range of baselines.
>
> More in detail, the FB method requires access to dataset with good coverage, it is trained fully offline, and it optimizes policies in a completely unsupervised way. While working well in small domains, it does not scale to more challenging problems, such as humanoid. FB-CPR resolves the limitations of FB through the imitation learning regularization as well as an online training approach. Compared to other regularized unsupervised RL approaches (e.g., ASE, CALM), FB-CPR leverages the FB components for trajectory encoding (unlike CALM that requires training a dedicated encoder) and to preserve policy optimality (unlike ASE, which builds on diversity principles). As demonstrated in Table 1, these differences are crucial to make FB-CPR perform significantly better than other baselines. FRE (similar to HILP) is an offline unsupervised RL algorithm and it proceeds through a two-step process to first learn a task encoding and then optimize policies accordingly using a standard offline regularized RL algorithm (IQL), which requires access to actions in the offline data. On the contrary, FB-CPR has access to observation-only datasets and it trains representations and policies end-to-end in an online fashion and the regularization is based on a conditional discriminator, with a significantly different objective than IQL. Finally, we have performed ablations and compared FB-CPR with several different variations of the regularization: 1) FB-AW trained offline on the action-labeled AMASS dataset (Fig. 4-bottom right); 2) FB-CPR trained online with BC regularization using the action-labeled AMASS dataset (Fig. 6-bottom right); 3) FB-CPR but with an unconditional discriminator (Fig.4-top left). In all cases, the specific combination of FB training and conditional-policy regularization is the crucial ingredient to achieve the best performance across all problems we considered.
>
> > I'm curious why FB-CPR is only shown on Humanoid control.
>
> We primarily focused on the humanoid problem due to its dynamical complexity, high dimensionality, availability of human data, and the possibility of defining a large set of “natural” tasks. Unfortunately no other existing RL benchmark have all these properties at the same time. Nonetheless, we have included in the revised version of the paper additional experiments in the AntMaze domain recently introduced in the OGBench benchmark, which provides a meaningful dataset of short trajectories designed to test stitching capabilities of unsupervised RL algorithm as well as a few downstream tasks to evaluate generalization performance. Please refer to the general answer for more details.

---

> > ### Comment · Reviewer_whMZ · 2024-11-26
> >
> > Thanks for the response. I appreciate the additional results on new benchmarks beyond Humanoid control as well as the new ablations. My initial concerns are mostly resolved. While I still think the algorithmic contribution (i.e., novelty) is relatively less prominent, I believe the strengths clearly outweigh the weaknesses. I also appreciate the authors' effort in providing an exceptionally detailed Appendix. I would like to give a rating of 7, but since this option is not available, I've instead increased the confidence score to 5.

---

> > > ### Author Response · Authors · 2024-12-01
> > >
> > > Thanks for your message! We are glad our response helped in clarifying your concerns and we are grateful for your support to the paper!

---

### Official Review · Reviewer_D81m · 2024-11-05

**Soundness:** 3
**Presentation:** 3
**Contribution:** 3
**Rating:** 6
**Confidence:** 3

**Summary:**

The paper proposes FB-CPR, a regularizer for unsupervised RL that improves its pre-training performance given a state-only trajectory dataset. FB-CPR is built on prior works of forward-backward (FB) representation in RL & successor measures of state. An FB approximation is trained with bellman update and can used to approximate successor measure, which is used as a zero-shot policy evaluator. Furthermore, using tricks from prior works, the authors further uses a latent vector z to extend the core components (FB & successor measure) to multiple policies. At pre-training time, FB-CPR is used as an regularizer, with a discrimination loss added to original unsupervised RL objective to make sure learned behavior stay close to the distribution in the dataset. FInally, the authors evaluated the merit of the proposed method with one natural application of simulated humanoid control, where mocap data (state-only trajectory dataset) is available. The resulting method outperforms baselines by some metrics, with outstanding performance in staying human-like.

**Strengths:**

The method is sound towards its goal. Intuitively, the discriminator encourages FB_CPR to learn policies that makes it roll out stay close to that in the dataset M, while the Q function itself is approximated by Fz.

The paper's presentation is clear considering its technical depth.

The paper is solid with extensive details for reproduction, evaluations, and sufficient mathematical details. The experiments considered sufficient baselines and there are a good amount of ablations for more insights

**Weaknesses:**

I think the authors explained the necessary details very well in their writing. However, given that the technical depth of FB heavily depends on prior works, I think the authors should definitely provide more intuitions along the way. When reading the prelim section, there are many times that I need to stop and ask myself why is this math transformation okay. A paper shall be relatively self-contained, and shall still allow readers to understand the high-level intuition of each equation without consulting the details of prior works.

One core assumption of the work is that one can reparameterize the policy dependence with the introduction of a policy embedding z (Eqn 4). I wonder how grounded is this, especially when z has to live in a continuous space. I list a few questions about z in my Questions section.

It's pretty clear that the paper heavily depends on the prior line of work of FB & state measure. I skimmed the mentioned works in the prelim section, and it seems that most of these works are evaluated on relative toy datasets only e.g. maze. Therefore, it's a big step for the authors to make claims about an application like humanoid control while skipping evidence of FB & state measure methods being general enough for RL benchmarks. This makes readers wonder whether there are hidden limitations of the proposed method. It seems that standard RL benchmarks are out of scope for this project given its unique setting of having some state dataset, but alternative evidence to address my above concern would be appreciated.

Figure 2 is very helpful to one's understanding of the method. However, I believe two simple changes could significantly enhance it: 1. link $F(\dot,z)^\top z$ back to $Q^\pi$ to remind readers of this connection back in prelim, especially because $Q^\pi=F(\dot,z)^\top z$ wasn't even highlighted with its own equation number. 2. Add a post-training / adaptation box that emphasizes all prior training is happening without reward, and is a separate part from downstream abilities.

Algorithm 1 is very helpful too, and I believe many readers would be looking for an algorithm box like this in main paper. The authors could highlight it's in the appendix more, or put a abbreviated algo box in main paper called "Algorithm 1 (informal)", and link readers to the full version in the appendix in the caption.

I wonder whether TD3 is a competitive baseline at all for "Naturalness" - it seems like sequence-based methods like diffuser serve as a stronger baseline here, as RL methods are purely optimized for reward maximization.

**Questions:**

What's the dimension of z? Usually, a policy has to be represented by a neural network. I wonder whether a compact latent variable z is expressive enough to represent policies that have combinatorial complexity. In that case, you must be sacrificing something. Could you provide more intuitions and conduct more ablations on the dimension of z?

---

> ### Author Response · Authors · 2024-11-25
> **Rebuttal**
>
> Thanks for your review! We addressed your questions in the general answer and in the following replies.
>
> > One core assumption of the work is that one can reparameterize the policy dependence with the introduction of a policy embedding z (Eqn 4). I wonder how grounded is this, especially when z has to live in a continuous space.
>
> In our experiments, the policy $pi_z(s)$ is a z-conditioned network with two initial “embedding layers”, one processing (s,z), and the other processing the state s alone. The second embedding layer has half the hidden units of the first layer, and their outputs are concatenated and fed into the main MLP. In FB-CPR the overall network is then optimized through the loss in (11), which allows us to learn a continuously parameterized policy that can generalize across states as well as the embeddings z.
>
> > What's the dimension of z? Usually, a policy has to be represented by a neural network. I wonder whether a compact latent variable z is expressive enough to represent policies that have combinatorial complexity. In that case, you must be sacrificing something. Could you provide more intuitions and conduct more ablations on the dimension of z?
>
> The state and action space of the humanoid agent are 358-dimensional and 69-dimensional respectively, while in our experiments we use a 256-dimensional Z space, which clearly does not allow expressing the whole combinatorial set of policies. Nonetheless, our model exploits two inductive biases that allow using its capacity to express policies that more relevant to the problem: 1) the conditional-policy regularization helps focusing the model on the “manifold” of human-like behaviors that is much smaller than all possible policies; 2) the low-rank decomposition in the FB models favors representations that capture variables with slower dynamics, hence inducing tasks whose optimal policies tend to generate more steady behaviors. Please refer to App. D.2 Fig. 7 for an ablation on the dimension of the embedding.
>
> > I wonder whether TD3 is a competitive baseline at all for "Naturalness"
>
> We prioritized TD3 over other models because the human evaluation was intended to primarily investigate whether the performance gap between FB-CPR and TD3 in reward-based and goal-based tasks could be partially explained by TD3 exploiting the physics of the humanoid model to optimize performance at the cost of “human-like” behaviors. While it is difficult to design a top-line with the best performance under a “human-like” constraint, we believe this evaluation provides a first qualitative assessment that FB-CPR may trade off performance and qualitative behavior and it is able to carry over the human-like regularization across reward-based and goal-based tasks.
>
> > It seems that most of these works are evaluated on relative toy datasets only e.g. maze. Therefore, it's a big step for the authors to make claims about an application like humanoid control while skipping evidence of FB & state measure methods being general enough for RL benchmarks.
>
> While still relatively recent, the FB model is covered in a variety of previous works (e.g., [Touati and Ollivier, 2021]) and it has been tested in several RL benchmarks including discrete and continuous mazes, FetchReach, Ms.Pacman and most of the environments in the URLB benchmark, and it is included as baseline in other unsupervised RL works (e.g., [Park et al., 2024b]). In particular, [Touati et al., 2023], [Pirotta et al, 2024] performed an extensive comparison between FB-based models and several reward-based and imitation-learning baselines, showing its effectiveness in mid-scale problems whenever good coverage offline datasets are provided. Due to space constraints, in the main paper we have focused on the core aspects of the FB model and its losses to allow the reader to understand the key differences we introduced in the FB-CPR model. We will  expand on more algorithmic aspects and theoretical properties in the supplementary material to make our contribution more self-contained.
>
> Please refer to the general answer for additional experiments in AntMaze (and walker). In general, we found that many existing benchmarks for unsupervised RL are defined on environment with simple low-dimensional dynamics, or do not have datasets of ``meaningful'' behaviors, or do not support more than a handful of hand-picked tasks. We believe that the humanoid benchmark we introduced in the paper provides a challenging and exhaustive evaluation that could help to advance research in this domain.
>
> > Figure 2 and Algorithm 1
>
> Thanks for the suggestions! We will update Fig.2 clarifying the FB components and illustrating the inference part. We will also add a compact version of Alg.1 if space permits it. Furthermore, we plan to release the training code together with the trained models and the Humanoid environment and benchmark in a few weeks.

---

> > ### Author Response · Authors · 2024-12-01
> >
> > Dear reviewer, we hope our rebuttal helped in resolving your concerns and we are wondering whether there is any additional point you would like us to clarify. Thanks!

---

### Author Response · Authors · 2024-11-25
**General answer: additional experiments**

Thanks to all reviewers for their detailed reviews and feedback! We have uploaded a revised version of the paper with additional experiments and ablations as you requested. We address common concerns and we list the main changes to the paper below, while we address specific comments in the individual rebuttal. We believe this further demonstrates the algorithmic novelty of FB-CPR and its generalization capabilities to a wide range of downstream tasks unseen at training.

**Additional experiments**

We included new experiments and baselines to address reviewers’ concerns about the fact that only the humanoid domain was used for empirical evaluation of FB-CPR.
* As requested by Reviewer D81m and Reviewer whMZ, in App. F we included new experiments in the AntMaze domain from the recent OGBench benchmark suite (https://seohong.me/projects/ogbench/). Notice that this is one of the few existing RL benchmarks suitable to test behavioral foundation models, as it provides useful datasets that can be used for offline training or demonstration regularization and it defines a variety of tasks to evaluate performance at test time. These new results not only confirm the advantage of the regularization in FB-CPR compared to other online and offline variants of FB, but they also show that it outperforms existing offline unsupervised RL algorithms even in the specific case of goal-based RL.
* We would like to point out that App. E also contains experiments in the DMC bipedal walker environment where we constructed demonstration data and parametrized tasks for evaluation (e.g., walk, run, spin, crawl at different speeds and directions). Also in this domain FB-CPR outperforms other variants of FB except for the one using only a non-conditional discriminator, which achieves slightly better performance in reward-based and imitation tasks, while being worse for tracking. This is due to the fact that, unlike in Humanoid and AntMaze, the behavior dataset contains a very well curated and balanced set of demonstrations that are coming from optimal policies that already cover quite well the space of tasks of interest.
* As suggested by Reviewer Gh4R, in App. H, we included additional comparison to METRA and some of its variants. Interestingly, METRA completely fails at learning any useful behavior and performs poorly across all types of tasks. Upon investigation, we observed that the agent simply learned to fall on the floor and to remain still in different positions. This happens despite all the loss functions, and in particular the ``diversity loss'' for representation learning, are well optimized during pre-training. This is due to the fact that, from the agent perspective, lying still on the floor in different positions can be regarded as displaying diverse behaviors, and no extra inductive bias would push the agent to learn more complicated skills (e.g., locomotion ones). We then tested other variants. First, we introduced prior knowledge by limiting features to (x,y) coordinates. While this avoids behavior collapse, it does not lead to significant performance improvements. Second, we combined METRA with the ASE regularization using the AMASS dataset. Overall, this reached comparable results as the DIAYN version of ASE we originally reported in the paper.
* We would like to stress that we primarily focused our evaluation on humanoid control due to its dynamical complexity, high dimensionality, availability of human data, and the possibility of defining a large set of “natural” tasks. We believe that the definition of a new humanoid benchmark with 45 reward tasks, over 900 test motions, and 50 goal poses, is a contribution in itself to support the advancement of the research in unsupervised RL. For comparison with previous RL humanoid literature, (Jiang et al., 2024) used only 15 motions and 6 rewards, (Park et al., 2024c) consider only x-y goal-based tasks, (Luo et al., 2024b) consider only 138 test motions and a few reward-based tasks that require hierarchical training on the top of their model. Beyond humanoid, most existing RL benchmarks are not suitable to test behavioral foundation models as they have simple dynamics, do not have datasets, or do not support more than a handful of hand-picked tasks. We plan to publicly release the SMPL-humanoid-based environment, the data processing tools, and all the tasks used in the paper in a few weeks.

---

### Author Response · Authors · 2024-11-25
**General answer: Understanding the embedding space**

**Understanding the embedding space**
Reviewer D81m requested ablations on the size of the embedding space and asked for better intuition on its structure.
* As requested by the reviewer, in App. D.2 Fig. 7 we included a new ablation showing how performance changes with the size of the embedding, while keeping all other parameters constant. The results show that the performance steadily improves until saturating at around d=128/256. On the other hand, at d=512 we start observing training stability and overfitting issues. While we believe that increasing batch size and correcting losses by dimension could mitigate these issues, we leave this investigation for future work.
* In App. G.1, we investigate how the embedding z correlates with the behaviors expressed by different policies. We have first computed the embedding of about 100 motions for each of five different categories (crawl, walk, jump, run, cartwheel) and we used UMAP dimensionality reduction technique to visualize them. We have also included the embedding of some of the reward-based tasks. This illustration reveals that 1) the latent space provides a meaningful clustering of the motions and 2) that motion and reward representations are well aligned in the latent space. We believe this shows how using the same representation to encode trajectories in the regularization term and in the low-rank decomposition of FB is crucial to successfully align motions and rewards at training and guarantee good inference performance in both types of tasks at test time.
* In App. G.2, we investigate how interpolation works in the embedding space. We selected a few pairs of “composable” reward-based tasks, such as “move” and “spin”, “move” and “left hand up”, … We first perform inference for the two tasks separately (i.e., z_1 = inference(reward_1) and z_2 = inference(reward_2)) and then we interpolate between the two as z_alpha = (1-alpha)*z_1 + alpha*z_2 for alpha in [0,1]. We have included the videos of pi(z_alpha) for different pairs of tasks and values of alpha in the supplementary material in the folder task_interpolation. Interestingly, for the large majority of the combinations not only the behavior changes quite smoothly with alpha but the model is able to effectively compose different tasks and generate complex behaviors such as “move while spinning” and “move with the left hand up”.
* Finally, we recall that in App. D we report an extensive qualitative evaluation of the behaviors learned by FB-CPR compared to other models. In particular, this analysis showed that 1) FB-CPR has the most extensive coverage of motions in the dataset (Fig. 9); 2) it retains a higher degree of diversity (Fig. 7-8); and 3) it is able to produce policies that are farther from the training set, which allows it to solve more diverse tasks at test time (Fig. 12).

---

### Author Response · Authors · 2024-11-25
**General answer: The importance of the conditional-policy regularization and the FB loss**

**The importance of the conditional-policy regularization and the FB loss**
Reviewer whMZ considered the contribution of the FB loss to be marginal, while on the contrary, Reviewer Gh4R asked to investigate further the actual value of the regularization of FB-CPR compared to standard FB.
* We would like to stress that the FB part of the algorithm is not limited to the F^t z term in the policy optimization loss, but it crucially learns the representation B used to both encode motions and perform inference for reward-based and goal-based tasks. Completely removing FB fundamentally reduces FB-CPR to CALM, which performs representation learning for encoding only driven by the imitation loss. As illustrated in Table 1, not only this prevents CALM from being applied to reward-based tasks, but it also across goal-based and tracking tasks. In the ablation in Fig.4 top right, we keep the representation learning B and we only remove the F^t z term in loss (11). We respectfully disagree with reviewer whMZ that the contribution is only marginal, since reward and tracking performance are improved by 13% and 12% respectively. While the improvement in reward is expected since maximizing F^t z corresponds to a standard policy improvement step, the advantage in tracking is less obvious and it illustrates that the synergy between the FB loss and the regularization term leads overall to better representations, better inference, and ultimately better policies.
* At the bottom of App. D.2, we included experiments of FB without any regularization and trained directly on online data (i.e., samples collected by executing policies from randomly selected zs) and unfortunately performance is very poor across all the evaluation metrics. This is due to the fact that FB itself does not have any effective exploration strategy to collect useful samples and it does not have any guidance on which policies to favor.
* In the original submission, we have already included several ablations to understand the role of different components of the algorithm: 1) FB-AW trained offline on the action-labeled AMASS dataset (Fig. 4-bottom right); 2) FB-CPR trained online with BC regularization using the action-labeled AMASS dataset (Fig. 6-bottom right); 3) FB-CPR but with an unconditional discriminator (Fig.4-top left). We believe these ablations provide a good coverage of the different regularization options and how the specific choices in FB-CPR (conditional discriminator on zs embedded through ER_FB) are critical to achieve the best overall performance.

---

### Meta-Review · Area_Chair_PrW9 · 2024-12-21

**Metareview:**

The paper proposes forward-backward representations with conditional policy regularization (FB-CPR), a novel regularization method for unsupervised reinforcement learning (RL) that uses an FB representation and behavioral regularization to improve performance when pre-training on state-only trajectory datasets. The key idea is to introduce a discriminator to enforce that learned behaviors remain close to those in the dataset, which is enhanced with a latent vector that adapts the regularization to multiple policies. The method is evaluated on humanoid control tasks, showing superior performance in terms of human-like behavior when compared to other baselines.

Reasons to accept
- The paper is well-written, and the proposed method is well-explained.
- The paper presents thorough evaluations in the humanoid control domain, demonstrating that FB-CPR significantly outperforms baselines on various metrics. Scaling up unsupervised RL up to Humanoid control is rarely seen.
- The introduction of a conditional discriminator to regularize behaviors is a unique feature that distinguishes FB-CPR from other offline RL methods.
- The method shows promise in humanoid control tasks, addressing important challenges like generalization and human-like behavior.

Reasons to reject
- The method heavily builds on previous FB frameworks, with the main novelty being the added discriminator term. This makes the contribution feel incremental compared to other works in unsupervised RL and skill-based regularization.
- FB-CPR is only evaluated on humanoid control tasks, raising concerns about its generalizability to other domains like manipulation or uncurated video datasets.
- The paper’s reliance on complex FB representations might be hard to follow for readers unfamiliar with the foundational works. More intuitive explanations and clearer examples would improve understanding.

Despite some initial concerns and questions from the reviewers, after the author-reviewer discussion, all the reviewers unanimously recommend accepting this paper. Consequently, I recommend accepting the paper.

**Additional Comments On Reviewer Discussion:**

During the rebuttal period, three reviewers acknowledged the author's rebuttal, and two reviewers adjusted the score accordingly (and one reviewer adjusted the confidence score).

---

### Decision · Program_Chairs · 2025-01-22

Accept (Poster)